# From Uniform to Learned Knots: A Study of Spline-Based Numerical Encodings for Tabular Deep Learning

## Abstract

Numerical preprocessing remains a critical component of tabular deep learning, where the representation of continuous features can strongly affect downstream performance. Although this is well understood for classical statistical and machine learning models, the extent to which explicit numerical preprocessing systematically benefits tabular deep learning remains less well understood. In this work, we study this question with a particular focus on spline-based numerical encodings. We investigate three spline families for encoding numerical features, namely B-splines, M-splines, and integrated splines (I-splines), under uniform, quantile-based, target-aware, and learnable-knot (gradient-based) placement. For the learnable-knot variants, we adopt a differentiable knot parameterization that enables stable end-to-end optimization of knot locations jointly with the backbone. We evaluate these numerical encodings on a diverse collection of public regression and classification datasets using MLP, ResNet, and FT-Transformer backbones, and compare them against common numerical preprocessing baselines. Our results show that the effect of numerical encodings depends strongly on the task, the output size of the encoding, and the backbone. For classification, piecewise-linear encoding (PLE) is the most robust choice overall, while spline-based encodings remain competitive. For regression, no single encoding dominates uniformly. Instead, performance depends on the spline family, knot-placement strategy, and the output size of the encoding, with larger gains typically observed for MLP and ResNet than for FT-Transformer. We further find that learnable-knot variants can be optimized stably under the proposed parameterization, but may substantially increase training cost, especially for M-spline and I-spline expansions. Overall, the results show that numerical encodings should be assessed not only in terms of predictive performance, but also in terms of computational overhead. An anonymized implementation is publicly available at `https://anonymous.4open.science/r/tdl-numerical-encodings-881C/`.

## 1 Introduction

Most tabular datasets contain numerical columns whose effects are often non-uniform. A feature may matter only over specific value ranges, exhibit threshold behavior, or relate to the target through localized changes (Hastie et al., 2009; Breiman et al., 2017). However, a common deep learning pipeline represents each numerical feature as a single scaled scalar, for example, through normalization or min-max scaling, and relies on the backbone to learn nonlinear structure from these inputs (Gorishniy et al., 2021; Borisov et al., 2024). This induces a strong bias toward global smooth transformations and can be mismatched with tabular problems in which predictive structure is tied to specific value ranges. In such cases, localized or threshold-based effects must be recovered indirectly by the backbone from scalar inputs alone.

Prior work shows that the representation of numerical features can substantially affect tabular deep learning performance. In particular, explicit encodings such as piecewise-linear encoding (PLE), and periodic mappings can improve results across several backbones (Gorishniy et al., 2022). Surveys also note that numerical encodings remain less systematically explored than architectural modifications, despite their practical importance (Borisov et al., 2024; Somvanshi et al., 2024). These observations motivate alternative numerical encodings that provide localized flexibility while remaining compatible with standard tabular backbones.

In this work, we study spline-based feature expansions as numerical encodings for tabular deep learning. We consider B-splines (de Boor, 1972), M-splines (Ramsay, 1988), and integrated splines (I-splines) (Meyer, 2008), and evaluate multiple knot placement strategies, including uniform and quantile-based placement, target-aware knots derived from CART and LightGBM split points (Breiman et al., 1984; Ke et al., 2017), and learnable-knot placement. For the learnable-knot variants, we use a differentiable parameterization based on ordered spacings, implemented through a softmax followed by cumulative summation, which preserves knot ordering while remaining fully differentiable (Durkan et al., 2019; Suh et al., 2024). To isolate the effect of numerical encodings, we keep the downstream models unchanged and evaluate MLP, ResNet, and FT-Transformer backbones (Gorishniy et al., 2021).

We summarize our main contributions as follows:

1. We present a systematic benchmark of numerical encodings for tabular deep learning, comparing standard scaling, min-max scaling, PLE, and spline-based encodings across regression and classification tasks.
2. We study spline-based encodings within a unified framework, covering B-splines, M-splines, and I-splines under uniform, quantile-based, target-aware, and learnable-knot placement. For the learnable-knot variants, we use a differentiable parameterization that enables stable end-to-end optimization of knot locations.
3. We show empirically that the effect of numerical encodings depends on the task, output size, and backbone. PLE is the most consistent choice for classification, whereas for regression the strongest results depend on the spline family and knot-placement strategy, with the preferred output size varying across settings and spline-based encodings often among the best-performing methods. We also show that learnable-knot variants can introduce substantial training overhead, especially for M-spline and I-spline expansions.

## 2 Related Work

**Tabular deep learning and tree ensembles.** Tabular deep learning has been studied with a range of architectures, including decision-tree-inspired models such as TabNet and NODE, attention-based models such as TabTransformer and SAINT, sequential state-space models such as Mambular, and strong MLP-based baselines such as ResNet-style MLPs and RealMLP (Arik & Pfister, 2019; Popov et al., 2019; Huang et al., 2020; Somepalli et al., 2021; Gorishniy et al., 2021; Thielmann et al., 2024; Holzmüller et al., 2025). At the same time, large empirical studies show that relatively simple backbones such as MLP, ResNet, and FT-Transformer remain strong and reproducible baselines on standard benchmarks (Gorishniy et al., 2021; 2022; Holzmüller et al., 2025). In parallel, gradient-boosted decision trees remain widely used and often serve as the reference level of performance on tabular benchmarks, with common implementations including XGBoost, LightGBM, and CatBoost (Chen & Guestrin, 2016; Ke et al., 2017; Prokhorenkova et al., 2018; Shwartz-Ziv & Armon, 2021).

**Tabular foundation models and PFN-based approaches.** Another line of work studies tabular foundation models based on in-context learning. TabPFN introduced the PFN paradigm for tabular classification, where a transformer is pretrained on synthetic tabular tasks and applied without task-specific gradient-based training (Hollmann et al., 2022). Subsequent work extended this line to broader settings, including larger datasets, regression, categorical features, and missing values (Hollmann et al., 2025; Grinsztajn et al., 2025). Recent models such as TabICL, Mitra, and Orion-MSP further improve scalability or synthetic-pretraining design for tabular in-context learning (Qu et al., 2025; Zhang et al., 2025; Bouadi et al., 2025). These models follow a different evaluation paradigm from the one studied here. They are typically designed to consume tabular inputs directly, together with model-specific internal representations or tokenization schemes, rather than relying on external numerical encodings as the main object of comparison in standard benchmarking pipelines (Hollmann et al., 2022; Grinsztajn et al., 2025; Qu et al., 2025).

**Numerical preprocessing and numerical encodings.** Several surveys note that much of the tabular deep learning literature focuses on backbone design, while numerical preprocessing is often limited to standard scaling or simple transformations (Borisov et al., 2024; Somvanshi et al., 2024). A key exception is the

work of Gorishniy et al. (2022), which studies explicit numerical encodings for continuous features, including piecewise-linear encoding (PLE) and periodic mappings, and shows improvements across MLP, ResNet, and FT-Transformer backbones. A related line of work views numerical encoding through the lens of function evaluations. For example, Shtoff et al. (2025) propose Function Basis Encoding (FBE) for factorization machines, where each numerical feature is mapped to a vector of function values, including spline bases. These works support the view that the representation of numerical features can materially affect tabular prediction performance.

**Relation to positional encodings.** Numerical encodings are also related to positional and Fourier feature maps. In all cases, a scalar input is mapped to a vector before the main model, giving the network a more structured representation than the raw value alone (Vaswani et al., 2017; Rahimi & Recht, 2007; Tancik et al., 2020). The distinction lies in what the scalar represents. In sequence models, it usually denotes position, so the encoding is designed to expose order, distance, or relative position, as in sinusoidal positional encodings and RoPE (Vaswani et al., 2017; Su et al., 2024; Kazemnejad et al., 2023). In tabular learning, the scalar is a feature value whose scale, distribution, and meaning vary across columns. Numerical encodings therefore serve a different role, namely to provide feature-wise representations of continuous values rather than shared representations of sequence locations (Gorishniy et al., 2022). This makes positional encodings a useful conceptual analogue, while keeping the motivation and inductive bias of tabular numerical encodings distinct.

**Splines as bases and as neural components.** Splines are classical tools for representing nonlinear effects. B-splines provide a standard stable basis, P-splines combine B-spline bases with smoothness penalties, and thin-plate regression splines provide low-rank constructions that are widely used in practice (de Boor, 1972; Eilers & Marx, 1996; Wood, 2003; 2017). M-splines and their integrals, integrated splines (I-splines), are commonly used for nonnegative and monotone constructions (Ramsay, 1988; Meyer, 2008). In deep learning, splines have often appeared as trainable model components rather than as general-purpose preprocessing. Examples include spline-parameterized activations (Bohra et al., 2020), learnable spline-based input normalization for tabular representation learning (Suh et al., 2024), and spline-parameterized function modules in KAN-style architectures (Liu et al., 2025; Eslamian et al., 2025).

**Learnable-knot spline models and free-knot splines.** A natural extension of classical spline modeling is to treat knot locations as learnable parameters rather than fixing them in advance. In traditional spline regression, free-knot formulations can increase flexibility but lead to a difficult nonconvex optimization problem because of ordering constraints and possible knot degeneracies (Mohanty & Fahnestock, 2021; Thielmann et al., 2025). In deep learning, differentiable parameterizations have recently made gradient-based knot optimization practical at scale. A common strategy is to parameterize positive knot spacings with a softmax and recover ordered knots by cumulative summation, yielding a differentiable map from unconstrained parameters to strictly increasing knot vectors (Durkan et al., 2019). Variants of this idea appear in neural spline flows (Durkan et al., 2019), in learnable spline-based normalization for tabular data (Suh et al., 2024), and in spline-parameterized components within KAN-style models (Liu et al., 2025; Eslamian et al., 2025; Zheng et al., 2025). Closest to our setting, Suh et al. (2024) learn per-feature spline transforms end-to-end, but their focus is on input normalization rather than explicit basis expansions consumed by otherwise unchanged tabular backbones.

Overall, prior work shows that numerical encodings can affect tabular deep learning performance (Gorishniy et al., 2022; Shtoff et al., 2025), and that spline parameterizations can be trained end-to-end as neural components (Durkan et al., 2019; Bohra et al., 2020; Suh et al., 2024; Liu et al., 2025; Eslamian et al., 2025). However, most existing spline-based approaches place splines inside the model architecture, for example as activations, normalization layers, or KAN-style modules, rather than using them as explicit numerical preprocessing for standard tabular backbones. It therefore remains unclear how learnable-knot spline encodings behave when used as preprocessing and compared directly against standard scaling, min-max scaling, and PLE. We address this by evaluating B-spline, M-spline, and I-spline encodings under uniform, quantile-based, target-aware placement based on CART and LightGBM split points (Breiman et al., 1984; Ke et al., 2017), and learnable-knot placement within a unified benchmark on MLP, ResNet, and FT-Transformer backbones.

## 3 Methodology

In this section, we describe numerical encoding as a feature-wise representation problem and define the spline-based encodings used throughout the study. We then describe the knot-placement strategies used to construct or learn the corresponding knot sequences.

**Feature-wise numerical encoding.** We consider supervised learning on tabular data with numerical and categorical features. For sample $i$, let

$$\mathbf{x}^{(i)} = \left(\mathbf{x}_{\text{num}}^{(i)}, \mathbf{x}_{\text{cat}}^{(i)}\right)$$

denote the input, where $\mathbf{x}_{\text{num}}^{(i)} \in \mathbb{R}^d$ contains $d$ numerical features and $\mathbf{x}_{\text{cat}}^{(i)}$ denotes the categorical variables. Let $y_i$ denote the corresponding target, and let $x_{i,j} \in \mathbb{R}$ denote the value of numerical feature $j$ for sample $i$. A numerical encoding maps each scalar value $x_{i,j}$ to a vector-valued representation

$$\boldsymbol{\phi}_j(x_{i,j}) = \left(\phi_{j,1}(x_{i,j}), \ldots, \phi_{j,m_j}(x_{i,j})\right) \in \mathbb{R}^{m_j},$$

where $m_j$ is the output size of the encoding for feature $j$. The encoded numerical input for sample $i$ is obtained by concatenating the feature-wise representations,

$$\boldsymbol{\Phi}(\mathbf{x}_{\text{num}}^{(i)}) = \left[\boldsymbol{\phi}_1(x_{i,1}) \mid \cdots \mid \boldsymbol{\phi}_d(x_{i,d})\right] \in \mathbb{R}^{\sum_{j=1}^{d} m_j}.$$

This notation describes a general feature-wise encoding and suppresses any encoding-specific parameters. For spline-based encodings, we make the dependence on the knot sequences explicit below. For convenience, the main notation used in the methodology is summarized in Appendix C.1.

Numerical encoding differs from categorical and positional encoding in the structure of the input being represented. Categorical encodings represent discrete identifiers, whereas positional encodings assign vector representations to positions or coordinates whose meaning is shared across the input, such as order, distance, or relative location. Numerical features in tabular data are different because each column represents a specific measured quantity. Each feature can have its own scale, distribution, meaning, and relationship to the target. This makes numerical encoding more difficult than applying a common representation rule across columns, because the same scalar value can have different meanings depending on the feature. Therefore, numerical encoding is a feature-wise modeling choice, which motivates the definition of a separate map $\boldsymbol{\phi}_j$ for each numerical feature (Gorishniy et al., 2022).

Different encoding families reflect different assumptions about how a numerical value should be represented. Standard scaling and min-max scaling keep each feature as a single rescaled scalar. PLE replaces the scalar by a piecewise-linear representation determined by bin boundaries, so that the value is encoded through its position between neighboring bins (Gorishniy et al., 2022). Spline-based encodings represent the value by evaluations of basis functions defined over a feature-specific knot sequence $\boldsymbol{\tau}_j$ (Shtoff et al., 2025; de Boor, 1972). The construction of $\boldsymbol{\tau}_j$ determines how resolution is allocated over the feature domain and is described in Section 3.1.

Figure 1 gives a schematic view of these encoding schemes for a single numerical feature and one example value. Detailed definitions of PLE and the B-, M-, and I-spline bases are provided in Appendix D and Appendices C.3 to C.5, respectively.

**Spline-based numerical encodings.** For spline-based encodings, the general feature-wise map $\boldsymbol{\phi}_j$ depends on the knot sequence of feature $j$. We write this dependence explicitly as $\boldsymbol{\phi}_j(x_{i,j}; \boldsymbol{\tau}_j)$, where $\boldsymbol{\tau}_j$ denotes the knot sequence. For a chosen spline family and degree $p$, let $b_{j,\ell}^{(p)}(x; \boldsymbol{\tau}_j)$ denote the $\ell$-th basis function for feature $j$ evaluated at a value $x$, with $\ell = 1, \ldots, m_j$. For sample $i$, the feature-wise spline encoding is

$$\boldsymbol{\phi}_j(x_{i,j}; \boldsymbol{\tau}_j) = \left(b_{j,1}^{(p)}(x_{i,j}; \boldsymbol{\tau}_j), \ldots, b_{j,m_j}^{(p)}(x_{i,j}; \boldsymbol{\tau}_j)\right) \in \mathbb{R}^{m_j}.$$

The corresponding encoded numerical vector is

$$\boldsymbol{\Phi}(\mathbf{x}_{\text{num}}^{(i)}; \boldsymbol{\tau}) = \left[\boldsymbol{\phi}_1(x_{i,1}; \boldsymbol{\tau}_1) \mid \cdots \mid \boldsymbol{\phi}_d(x_{i,d}; \boldsymbol{\tau}_d)\right] \in \mathbb{R}^{\sum_{j=1}^{d} m_j},$$

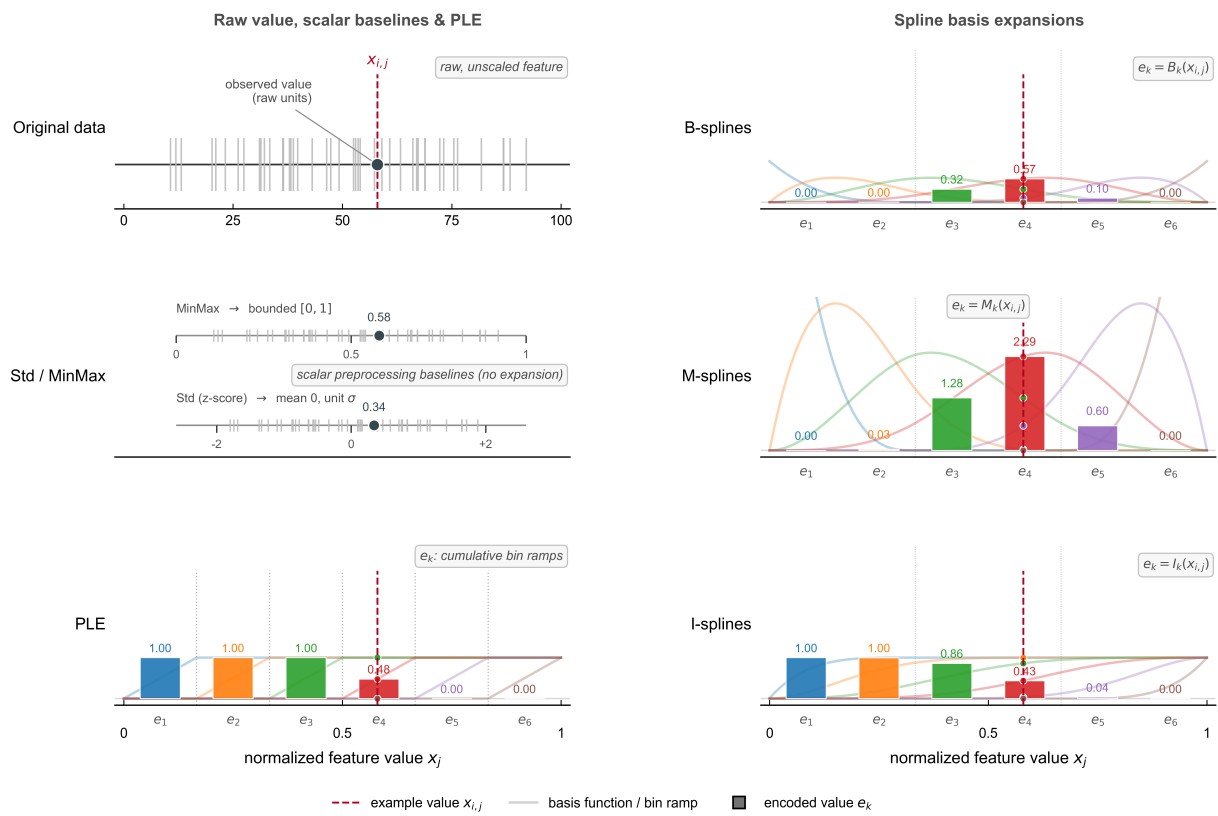

Figure 1: The figure illustrates how a single numerical value $x_{i,j}$, shown by the red dashed line, is represented by different schemes. Std and MinMax keep one scalar value per feature, whereas PLE and B-/M-/I-splines expand the value into coordinates $(e_1, \ldots, e_{m_j})$ by evaluating bin ramps or basis functions. The markers show the evaluated values and the bars show the resulting encoded coordinates. B-splines provide smooth local bases, M-splines nonnegative local bases, and I-splines cumulative monotone bases. For the PLE and spline panels, we set the output size to $m_j = 6$ for illustration. Details of the construction are provided in Appendix E.

where $\boldsymbol{\tau} = (\boldsymbol{\tau}_1, \ldots, \boldsymbol{\tau}_d)$ denotes the collection of feature-wise knot sequences.

Throughout, we use cubic splines with degree $p = 3$. For B-, M-, and I-splines, the number of basis functions is determined by the number of internal knots $K_j$ through

$$m_j = K_j + p + 1.$$

Therefore, knot indices range from 1 to $K_j$, while basis-function indices range from 1 to $m_j$.

B-splines provide compactly supported local polynomial bases (de Boor, 1972). M-splines are nonnegative normalized spline bases, and I-splines are obtained by integrating M-splines, which gives monotone basis functions (Ramsay, 1988; Meyer, 2008). We include these three spline families because they share the same knot-based construction while inducing different feature representations.

In all cases, the downstream model receives the encoded numerical vector $\boldsymbol{\Phi}(\mathbf{x}_{\text{num}}^{(i)}; \boldsymbol{\tau})$ together with the categorical features $\mathbf{x}_{\text{cat}}^{(i)}$, which are processed separately as described in Section 4. The backbone architecture is kept fixed in order to isolate the effect of the numerical encoding.

**Spline encodings in the pipeline.** For fixed-knot variants, each knot sequence $\boldsymbol{\tau}_j$ is constructed during preprocessing using only the training split of each fold and then kept fixed during backbone training. In these cases, we do not fit spline coefficients directly. That is, we do not define or learn a separate univariate spline function with coefficient vector $\boldsymbol{\alpha}_j = (\alpha_{j,1}, \dots, \alpha_{j,m_j})$ of the form

$$f_j(x) = \sum_{\ell=1}^{m_j} \alpha_{j,\ell} b_{j,\ell}^{(p)}(x; \boldsymbol{\tau}_j).$$

Instead, the downstream network learns weights on the encoded numerical representation $\boldsymbol{\Phi}(\mathbf{x}_{\text{num}}^{(i)}; \boldsymbol{\tau})$. For learnable-knot variants, we use the same feature-wise basis expansion, but update the entries of the internal-knot vector $\boldsymbol{\kappa}_j$ jointly with the downstream model during training rather than fixing them during preprocessing.

With this notation in place, we next describe how the internal-knot vector $\boldsymbol{\kappa}_j$ and the corresponding knot sequence $\boldsymbol{\tau}_j$ are constructed for the knot-placement strategies considered in our experiments.

### 3.1 Knot Placement Strategies

A central methodological component of our work is the treatment of knot placement. Throughout this subsection, let $\mathbf{x}_j = (x_{1,j}, \dots, x_{n,j})^\top$ denote the training values of numerical feature $j$ in the current fold. For each feature, we construct an ordered internal-knot vector

$$\boldsymbol{\kappa}_j = (\kappa_{j,1}, \dots, \kappa_{j,K_j}), \qquad \kappa_{j,1} < \cdots < \kappa_{j,K_j}. \tag{1}$$

These internal knots are then augmented with the usual boundary knots to form the full spline knot sequence $\boldsymbol{\tau}_j$ used by the basis functions. Except for the learnable-knot variant, the internal knots are determined during preprocessing using only the training split of each fold and remain fixed during downstream training. In the learnable-knot variant, the internal knots are treated as learnable parameters, and the full knot sequence $\boldsymbol{\tau}_j$ is constructed from them during training.

We consider uniform placement, quantile-based placement, target-aware placement, and learnable-knot placement. For target-aware placement, we consider two variants based on CART and LightGBM split points. The individual strategies are described below.

#### 3.1.1 Uniform knot placement

Uniform internal knots are equally spaced over the observed training range of feature $j$:

$$\kappa_{j,\ell} = \min(\mathbf{x}_j) + \frac{\ell}{K_j + 1}\big(\max(\mathbf{x}_j) - \min(\mathbf{x}_j)\big), \qquad \ell = 1, \dots, K_j. \tag{2}$$

#### 3.1.2 Quantile knot placement

Quantile internal knots place more knots in regions where samples are concentrated:

$$\kappa_{j,\ell} = Q_j\left(\frac{\ell}{K_j + 1}\right), \qquad \ell = 1, \dots, K_j, \tag{3}$$

where $Q_j(\cdot)$ is the empirical quantile function of the training values $\mathbf{x}_j$. This strategy is target-agnostic and adapts only to the marginal distribution of feature $j$.

#### 3.1.3 Target-aware knot placement

After min-max scaling the training values of feature $j$ to $[0,1]$, we construct a *univariate* target-aware internal-knot vector $\boldsymbol{\kappa}_j$ by fitting a predictive tree on the training split of each fold. In this subsection, $\mathbf{x}_j = (x_{1,j}, \dots, x_{n,j})^\top$ and $x_{i,j}$ refer to these scaled training values. We consider two variants, CART-based and LightGBM-based. Let

$$\{(x_{i,j}, y_i)\}_{i=1}^n \tag{4}$$

denote the resulting training pairs for feature $j$.

**CART-based knots.** We fit a depth- and sample-constrained univariate CART tree $T_j$, using a regressor for regression and a classifier for classification (Breiman et al., 1984). Let $\mathcal{S}_j$ be the multiset of split thresholds used by the internal nodes of $T_j$ for feature $j$. We first map thresholds to the observed training range:

$$\widetilde{\mathcal{S}}_j = \big\{ \mathrm{clip}(s; \min(\mathbf{x}_j), \max(\mathbf{x}_j)) : s \in \mathcal{S}_j \big\}, \tag{5}$$

and then deduplicate and sort them to obtain candidates $\{s_{j,(1)} < \cdots < s_{j,(r)}\}$.

For numerical stability, we enforce a minimum spacing constraint by pruning near-duplicates. We keep a subsequence $\mathcal{C}_j \subseteq \{s_{j,(1)}, \ldots, s_{j,(r)}\}$ such that

$$|s - s'| \geq \epsilon \quad \text{for all distinct } s, s' \in \mathcal{C}_j, \tag{6}$$

where $\epsilon$ is set as a small fraction of the normalized range (DiMatteo et al., 2001; Spiriti et al., 2013).

To match a desired spline complexity, we convert the target number of basis functions $m_j$ and the spline degree $p$ into a target number of internal knots:

$$K_j = m_j - p - 1. \tag{7}$$

If $|\mathcal{C}_j| > K_j$, we retain the $K_j$ most informative thresholds, ranked by the impurity reduction of their corresponding split. For a split at threshold $s$ occurring at node $v$ with children $L$ and $R$, we use

$$\Delta I_v(s) = I(v) - \frac{n_L}{n_v} I(L) - \frac{n_R}{n_v} I(R), \tag{8}$$

where $I(\cdot)$ denotes the node impurity and $n_v, n_L, n_R$ are sample counts. If $|\mathcal{C}_j| < K_j$, we supplement the remaining knots with quantiles of the training values $\mathbf{x}_j$ until reaching $K_j$. The selected thresholds and supplemental quantiles are then sorted to form the internal-knot vector $\boldsymbol{\kappa}_j$, and the full knot sequence $\boldsymbol{\tau}_j$ is constructed from $\boldsymbol{\kappa}_j$ using the standard boundary handling for the chosen spline family.

**LightGBM-based knots.** We follow the same construction, but replace $T_j$ with a univariate gradient-boosted tree ensemble (Friedman, 2001; Ke et al., 2017). Let $\mathcal{S}_j^{(t)}$ be the set of thresholds used by tree $t$, and let $g_t(s) \geq 0$ denote the split gain assigned by LightGBM to threshold $s$. We aggregate threshold importance across the ensemble through

$$G_j(s) = \sum_{t:\, s \in \mathcal{S}_j^{(t)}} g_t(s), \tag{9}$$

rank candidate thresholds by $G_j(s)$, and then apply the same spacing filter in equation 6 and the same target internal-knot budget in equation 7. If no valid splits are produced, for example because of sparsity or strong regularization, we fall back to quantile-based internal knots to ensure a usable basis. We include this variant because aggregating split thresholds across an ensemble can provide a more stable set of high-gain knot candidates than a single CART tree. The selected thresholds and supplemental quantiles, when needed, are then sorted to form the internal-knot vector $\boldsymbol{\kappa}_j$, and the full knot sequence $\boldsymbol{\tau}_j$ is constructed from $\boldsymbol{\kappa}_j$ using the standard boundary handling for the chosen spline family.

**Relation to target-aware binning in PLE.** Our procedure is target-aware in the sense that knot candidates are derived from supervised split thresholds. It differs from the target-aware preprocessing used for PLE in Gorishniy et al. (2022) in two respects. First, PLE uses supervised split thresholds for binning and encoding, whereas we use them to define internal spline knots for a continuous basis expansion. Second, our construction enforces spline-specific constraints, including the conversion from basis size to an internal-knot budget in equation 7, the minimum-spacing condition in equation 6, and the supplementation or pruning of candidate thresholds when too few or too many are available. These steps are specific to spline basis construction and are not required for PLE.

### 3.1.4 Learnable knot placement

In the learnable-knot variant, also referred to as *gradient-based knot placement*, we treat the internal-knot vector $\boldsymbol{\kappa}_j$ as learnable parameters and optimize them jointly with the downstream backbone by backpropagation. As in the target-aware setting, all numerical features are first min-max scaled to $[0, 1]$ using the

training split of each fold. This places all spline constructions on a common domain and ensures that the learned internal knots satisfy $\kappa_{j,\ell} \in (0, 1)$.

**Initialization.** For each numerical feature $j \in \{1, \ldots, d\}$, we fix the number of internal knots $K_j$ and initialize the internal-knot vector

$$\boldsymbol{\kappa}_j = (\kappa_{j,1}, \ldots, \kappa_{j,K_j})$$

from the same rule as the corresponding fixed-knot baseline, namely uniform placement. This provides a valid and well-spaced starting configuration.

**Ordered knots via spacing parameterization.** Direct optimization of $\boldsymbol{\kappa}_j$ is numerically fragile because the parameters must satisfy the strict ordering constraint

$$0 < \kappa_{j,1} < \cdots < \kappa_{j,K_j} < 1,$$

and neighboring knots may collide during training. We therefore optimize an unconstrained vector $\mathbf{a}_j \in \mathbb{R}^{K_j+1}$ and map it to a strictly increasing internal-knot vector through positive interval widths. We first define the vector of normalized allocation weights

$$\boldsymbol{\pi}_j = (\pi_{j,1}, \ldots, \pi_{j,K_j+1}),$$

with entries

$$\pi_{j,r} = \frac{\exp(a_{j,r})}{\sum_{s=1}^{K_j+1} \exp(a_{j,s})}, \qquad r = 1, \ldots, K_j + 1. \tag{10}$$

We then convert these allocation weights into the interval-width vector

$$\mathbf{w}_j = (w_{j,1}, \ldots, w_{j,K_j+1}),$$

with entries

$$w_{j,r} = \delta + \big(1 - (K_j + 1)\delta\big)\pi_{j,r}, \qquad r = 1, \ldots, K_j + 1. \tag{11}$$

Here, $\delta$ is chosen such that $0 < \delta < 1/(K_j+1)$. By construction, $w_{j,r} \geq \delta$ and $\sum_{r=1}^{K_j+1} w_{j,r} = 1$. The internal knots are then obtained by cumulative summation,

$$\kappa_{j,\ell} = \sum_{r=1}^{\ell} w_{j,r}, \qquad \ell = 1, \ldots, K_j, \tag{12}$$

which guarantees

$$0 < \kappa_{j,1} < \cdots < \kappa_{j,K_j} < 1.$$

The resulting internal-knot vector $\boldsymbol{\kappa}_j$ is combined with the standard boundary construction to form the full spline knot sequence $\boldsymbol{\tau}_j$ used by the basis functions. This softmax-cumsum parameterization follows standard constructions for ordered spline breakpoints in differentiable spline models (Durkan et al., 2019; Suh et al., 2024).

**Spline feature expansion and gradient flow.** Given $\boldsymbol{\kappa}_j$, we construct the full knot sequence $\boldsymbol{\tau}_j$ and compute the spline encoding $\phi_j(x_{i,j}; \boldsymbol{\tau}_j)$ in each forward pass. The expanded numerical representation for sample $i$ is then formed as $\boldsymbol{\Phi}(\mathbf{x}_{\mathrm{num}}^{(i)}; \boldsymbol{\tau}(\mathbf{a}))$ by concatenating the feature-wise encodings across numerical features. Since $\phi_j$ depends on $\boldsymbol{\tau}_j$, and $\boldsymbol{\tau}_j$ is a differentiable function of $\mathbf{a}_j$ through equation 10–equation 12, gradients from the task loss propagate to $\mathbf{a}_j$.

**Learning objective.** Let $\mathbf{a} = (\mathbf{a}_1, \ldots, \mathbf{a}_d)$ collect the knot parameters, let $\boldsymbol{\kappa}(\mathbf{a}) = \{\boldsymbol{\kappa}_j(\mathbf{a}_j)\}_{j=1}^d$ denote the induced internal-knot vectors, and let $\boldsymbol{\tau}(\mathbf{a}) = \{\boldsymbol{\tau}_j(\mathbf{a}_j)\}_{j=1}^d$ denote the corresponding full knot sequences. Let $f_\theta$ be the downstream backbone applied to the expanded numerical representation together with the categorical features. We minimize

$$\min_{\theta, \mathbf{a}} \frac{1}{n} \sum_{i=1}^{n} \mathcal{L}\Big(f_\theta\Big(\boldsymbol{\Phi}(\mathbf{x}_{\mathrm{num}}^{(i)}; \boldsymbol{\tau}(\mathbf{a})), \mathbf{x}_{\mathrm{cat}}^{(i)}\Big), y_i\Big) + \lambda \mathcal{R}_{\mathrm{space}}(\mathbf{a}), \tag{13}$$

where $\mathcal{L}$ is cross-entropy for classification and squared loss for regression.

**Spacing regularization.** Very small interval widths can lead to ill-conditioned basis evaluations. To discourage such configurations, we penalize the induced spacings using a reciprocal barrier,

$$\mathcal{R}_{\text{space}}(\mathbf{a}) \; = \; \frac{1}{d}\sum_{j=1}^{d} \frac{1}{K_j+1} \sum_{r=1}^{K_j+1} \frac{1}{w_{j,r}(\mathbf{a}_j)+\varepsilon}, \tag{14}$$

where $\varepsilon > 0$ and $w_{j,r}(\mathbf{a}_j)$ denotes the $r$-th entry of the interval-width vector $\mathbf{w}_j$ produced by equation 10 and equation 11. Such spacing penalties are common in free-knot spline optimization (DiMatteo et al., 2001; Spiriti et al., 2013; Thielmann et al., 2025) and are also consistent with stability heuristics used in differentiable spline models (Durkan et al., 2019; Suh et al., 2024). This design is also in line with recent work that incorporates structured smooth components into end-to-end trainable additive neural models (Luber et al., 2023). Detailed steps of the end-to-end preprocessing workflow and learnable-knot optimization are provided in Appendix H, Algorithms 1 and 2, respectively.

**Stability considerations.** In our implementation of gradient-based knot optimization, the number of internal knots $K_j$ is fixed in advance, and only their locations, represented by $\boldsymbol{\kappa}_j$, are updated during training through the unconstrained parameters $\mathbf{a}_j$, jointly with the backbone parameters $\theta$ as described in Algorithm 2. Stability is supported by the spacing parameterization in our formulation. The entries of $\boldsymbol{\pi}_j$ are mapped to interval widths through equation 11, and the ordered internal knots are then recovered by cumulative summation as in equation 12. This guarantees valid ordered knot configurations with minimum spacing controlled by $\delta$ and removes the need for sorting or post-hoc merging (Durkan et al., 2019; Suh et al., 2024). In contrast to merge-based free-knot approaches, which use a predefined merge threshold $\alpha$ for nearby knots, our formulation enforces valid knot configurations directly through the spacing parameterization and the minimum-spacing constant $\delta$. This avoids the need for an additional merge-threshold hyperparameter (Thielmann et al., 2025).

In practice, optimization was further supported by initialization from a well-spaced fixed-knot rule and, when used, by a warm-start phase in which $\mathbf{a}$ is frozen for the first $E_{\text{warm}}$ epochs before joint updates of $(\theta, \mathbf{a})$ are enabled. Empirically, we observed stable knot updates for learning rates $\eta_a$ comparable to, and in some cases up to twice, the backbone learning rate $\eta_\theta$. By contrast, too small values of $\eta_a$ often led to negligible knot movement. A qualitative illustration of knot relocation during training is provided in Appendix K.1.

## 4 Experimental Setup

### 4.1 Datasets and numerical encodings

**Datasets and basic preprocessing.** We evaluate our methods on 25 tabular datasets covering regression and classification tasks, collected from the UCI Machine Learning Repository and OpenML. Dataset statistics and abbreviations are reported in Table 4. We use 5-fold cross-validation for all experiments. In total, this yields $25 \times 5 \times 3 \times 14 = 5250$ training runs across 25 datasets, 5 folds, 3 backbones, and 14 numerical encoding methods. We apply a minimal preprocessing pipeline. Rows with missing values are removed and no explicit outlier treatment is performed. Numerical features are scaled to $[0, 1]$ before applying feature-expansion methods, which ensures comparable basis construction across heterogeneous feature scales. Categorical features are label-encoded as integer identifiers, without one-hot or target encoding. Unseen categories at evaluation time are assigned the identifier $-1$. For MLP and ResNet, these identifiers are used directly as scalar inputs. For FT-Transformer, numerical and categorical features are processed by separate tokenizers, using a linear tokenizer for numerical features and an embedding tokenizer for categorical features (Gorishniy et al., 2021).

**Numerical encoding methods.** We study spline-based numerical encodings and PLE under a capacity-controlled protocol; see Table 3. In the main benchmark, we evaluate Std, MinMax, PLE, B-splines (BS), I-splines (IS), and the learnable-knot M-spline variant. Fixed-knot M-spline variants are excluded from the main benchmark and are reported only in the ablation study. For BS and IS, we consider uniform, quantile-based, learnable-knot, and target-aware knot placement. For target-aware placement, we use two variants

based on CART and LightGBM, as described in Section 3. The configuration details for the target-aware variants are provided in Table 8.

**Output size and matched PLE baseline.** To isolate the effect of knot or bin placement from representation size, we fix the per-feature output size to $m \in \{7, 15, 30\}$ for all features, for both spline encodings and PLE. For cubic splines ($p = 3$), $m = 7$ corresponds to three internal knots through $m = K + p + 1$, making it the smallest non-trivial spline resolution. We then increase the output size to $m = 15$ and $m = 30$ to examine how higher resolution affects predictive performance. Together, these settings cover low, medium, and relatively high output sizes while keeping the full benchmark computationally manageable. For PLE, the matched output size is implemented through the number of bins. An adaptive PLE variant is used only in the ablation study, where a tree-guided procedure selects the effective number of bins from $[5, 50]$

## 4.2 Models and evaluation protocol

**Backbones.** We evaluated three tabular backbones, MLP, ResNet, and FT-Transformer, to test whether the effect of numerical encodings is consistent across different model classes. MLP serves as a simple baseline with limited inductive bias, so improvements can be attributed more directly to the input representation. ResNet follows the tabular ResNet design of Gorishniy et al. (2021) and adds residual connections and normalization, providing a stronger MLP-based backbone. FT-Transformer uses feature tokenization and self-attention to model feature interactions (Gorishniy et al., 2021). Complete architectural hyperparameters are provided in Table 7.

**Training and evaluation.** Because the main focus of this study is the effect of numerical encodings, we adopt a shared training protocol across backbones. This provides a controlled comparison in which differences can be attributed more directly to the preprocessing method rather than to backbone-specific tuning. All backbones are trained with AdamW using learning rate $10^{-4}$, weight decay $10^{-5}$, batch size 512, and at most 200 epochs. We use early stopping on the validation metric with patience 15, together with a ReduceLROnPlateau scheduler with patience 10 and factor 0.1. We use 5-fold cross-validation for all experiments, with stratification for classification tasks, and hold out 10% of each training fold as a validation split for early stopping. To prevent information leakage, all preprocessing, including feature scaling, numerical encoding, and target standardization for regression, is fit using only the training portion of each fold and then applied to the corresponding validation and test partitions. For regression, targets are z-score normalized using training-fold statistics. For reproducibility, we use fold-specific seeds given by `seed + fold_id` and seed all random number generators consistently. We evaluate all methods using 5-fold cross-validation. For regression, we report NRMSE ($\downarrow$), while for classification we report AUC ($\uparrow$). On multiclass datasets, AUC is computed as weighted one-vs-rest AUC. Reported results are summarized as mean $\pm$ standard deviation across folds.

To preserve the intrinsic geometry of B-spline and I-spline encodings, such as partition-of-unity and cumulative structure, we do not apply additional feature-wise normalization to these encodings; see Appendices C.3 and C.5. For learnable-knot M-splines, the normalization term in the M-spline basis depends on the knot locations in $\boldsymbol{\tau}_j$ and changes during training. In particular, it contains the inverse knot-span term $(\tau_{j,\ell+p+1} - \tau_{j,\ell})^{-1}$, where $\tau_{j,\ell}$ and $\tau_{j,\ell+p+1}$ are entries of the knot sequence $\boldsymbol{\tau}_j$; see Appendix C.4. When adjacent knot spans shrink, this term can become large and lead to large basis values in practice. We therefore apply LayerNorm to each learnable-knot M-spline feature block as a numerical stabilization step (Ba et al., 2016).

## 4.3 Results and Analysis

We compare preprocessing methods from two complementary perspectives. First, we summarize performance using critical difference (CD) diagrams based on average ranks, following the Friedman/Nemenyi protocol commonly used in multi-dataset benchmarking (Demšar, 2006; Feuer et al., 2024; Kadra et al., 2024; Thielmann et al., 2024). The regression and classification CD diagrams are shown in Figures 2 and 3, respectively. In these diagrams, lower average ranks indicate better performance, and methods connected by a horizontal bar are not significantly different under the post-hoc test. We keep the CD diagrams because they compactly show both the rank ordering of methods and groups of methods that are statistically

indistinguishable. To make the rank values easier to inspect, the corresponding average-rank tables across all backbones are provided in Appendix I.1 for regression and Appendix I.2 for classification.

Second, we report backbone-specific heatmaps of the average test metric across datasets for each output size $m \in \{7, 15, 30\}$ in Figures 4 and 5. Each heatmap cell is the average NRMSE or AUC over all datasets of the corresponding task for a fixed backbone, preprocessing method, and output size. The CD diagrams provide an aggregate rank-based comparison, whereas the heatmaps show overall performance patterns across backbones, preprocessing methods, and output sizes. For Std and MinMax, feature expansion is not applicable, and their values therefore remain the same across output sizes in the heatmaps. These methods are included as baseline reference points for comparison. Implementation details, significance tests, and average-rank tables for the CD diagrams are provided in Appendix I. Detailed per-dataset results are reported in Tables 11, 12, and 13 for regression, and in Tables 14, 15, and 16 for classification.

Across all six CD settings, corresponding to regression and classification for $m \in \{7, 15, 30\}$, the Friedman test rejects the null hypothesis of equal performance. This suggests that the choice of preprocessing method has a statistically significant overall effect. At the same time, the Nemenyi cliques indicate that several top-ranked methods are often not significantly different from one another. The CD diagrams should therefore be read as identifying clusters of strong methods rather than a single universally dominant winner.

**Regression.** The regression results are clearly output-size dependent. At $m = 7$, the CD diagram in Figure 2 is led by B-spline variants with fixed or target-aware knot placement, with BS-LGBM, BS-Q, and BS-CART occupying the top ranks. At $m = 15$ and $m = 30$, the ranking shifts toward I-spline and learnable-knot variants. In particular, IS-Q and IS-LGBM remain among the strongest methods at both larger output sizes, and IS-Grad-U becomes competitive at $m = 30$. PLE is not among the strongest methods at low output size, but becomes more competitive at $m = 30$. By contrast, Std and MinMax remain near the bottom across all regression settings.

The regression heatmaps in Figure 4 highlight the dependence on the backbone. For MLP and ResNet, increasing the output size often improves the average NRMSE of spline-based methods, especially for target-aware and learnable-knot variants. On MLP, for example, BS-Grad-U improves from 0.2491 at $m = 7$ to 0.2322 at $m = 15$ and 0.2278 at $m = 30$, while IS-Grad-U attains the lowest average NRMSE at $m = 30$ with 0.2273. On ResNet, the strongest methods likewise shift toward larger output sizes, with IS-LGBM achieving the lowest average NRMSE at $m = 30$ with 0.2246. However, FT-Transformer behaves differently. Several B-spline variants worsen as $m$ increases, for example BS-Q changes from 0.2451 to 0.2666 to 0.3034 across $m = 7, 15, 30$. Std, with NRMSE 0.2465 remains competitive and in fact outperforms PLE at all three output sizes. At $m = 15$ and $m = 30$, it also outperforms many spline-based encodings, including all B-spline variants, IS-Grad-U, and MS-Grad-U. Overall, larger output sizes are often useful for regression with MLP and ResNet, but can become counterproductive for FT-Transformer.

**Classification.** The classification results are more stable than the regression results. In all three CD diagrams in Figure 3, PLE is the top-ranked method, and its average rank improves from 5.0 at $m = 7$ to 4.0 at $m = 15$ and 3.3 at $m = 30$. At $m = 7$, B-spline variants with target-aware or learnable-knot placement remain competitive. As the output size increases, I-spline variants form the closest competing group to PLE. As in regression, Std and MinMax remain near the bottom across all settings.

The classification heatmaps in Figure 5 show a more stable pattern than in regression. Across all three backbones, PLE achieves the highest average AUC at every output size, with small but consistent gains as $m$ increases. For example, its average AUC rises from 0.9194 to 0.9234 on MLP, from 0.9298 to 0.9312 on ResNet, and from 0.9319 to 0.9331 on FT-Transformer when moving from $m = 7$ to $m = 30$. More generally, many encoding methods improve with larger $m$, but the gain from $m = 15$ to $m = 30$ is usually smaller than the gain from $m = 7$ to $m = 15$. This pattern is visible for both B-spline and I-spline variants. For MLP and ResNet, several spline-based methods improve clearly over MinMax and often over Std, but they still do not consistently match PLE. In particular, several B-spline variants attain their strongest average AUC around $m = 15$, after which gains level off or slightly reverse at $m = 30$. FT-Transformer shows a different pattern. Std, with AUC 0.9256, remains competitive with most spline-based encodings. Among the spline

variants, only BS-Q, BS-CART, and BS-LGBM at $m = 15$ surpass Std, while the remaining spline settings stay below it.

**Backbone sensitivity and practical interpretation.** The results show that the effect of numerical preprocessing depends on both the task and the backbone. In regression, the strongest methods vary with the output size, with B-spline variants tending to perform best at smaller sizes and I-spline or learnable-knot variants becoming more competitive as the output size increases. In classification, PLE is the most robust choice across backbones and output sizes, while spline-based encodings remain competitive but do not consistently surpass it. The heatmaps also indicate that expressive preprocessing is more beneficial for MLP and ResNet than for FT-Transformer, which often shows smaller or less consistent gains, especially in regression. These findings suggest that preprocessing should be selected jointly with the task and backbone rather than treated as an independent design choice.

**Main takeaways:**

- The CD diagrams show statistically significant differences among preprocessing methods across all output sizes for both regression and classification.
- In regression, the aggregate ranking changes with output size. At $m = 7$, the strongest ranks are typically obtained by B-spline variants such as BS-LGBM, BS-Q, and BS-CART, whereas at $m = 15$ and $m = 30$ the ranking shifts toward I-spline and learnable-knot methods, especially IS-Q, IS-LGBM, and IS-Grad-U.
- In classification, PLE is the strongest overall baseline. It is top-ranked in the CD diagrams for all three output sizes and yields the highest average AUC across MLP, ResNet, and FT-Transformer.
- Larger and more expressive preprocessing tends to benefit MLP and ResNet more than FT-Transformer. For FT-Transformer, the gains are generally smaller and less consistent.
- Std and MinMax are generally weaker than explicit numerical encodings, especially for MLP and ResNet. For stronger backbones such as FT-Transformer, however, Std often remains competitive and can be a reasonable choice when simplicity and computational budget are important.

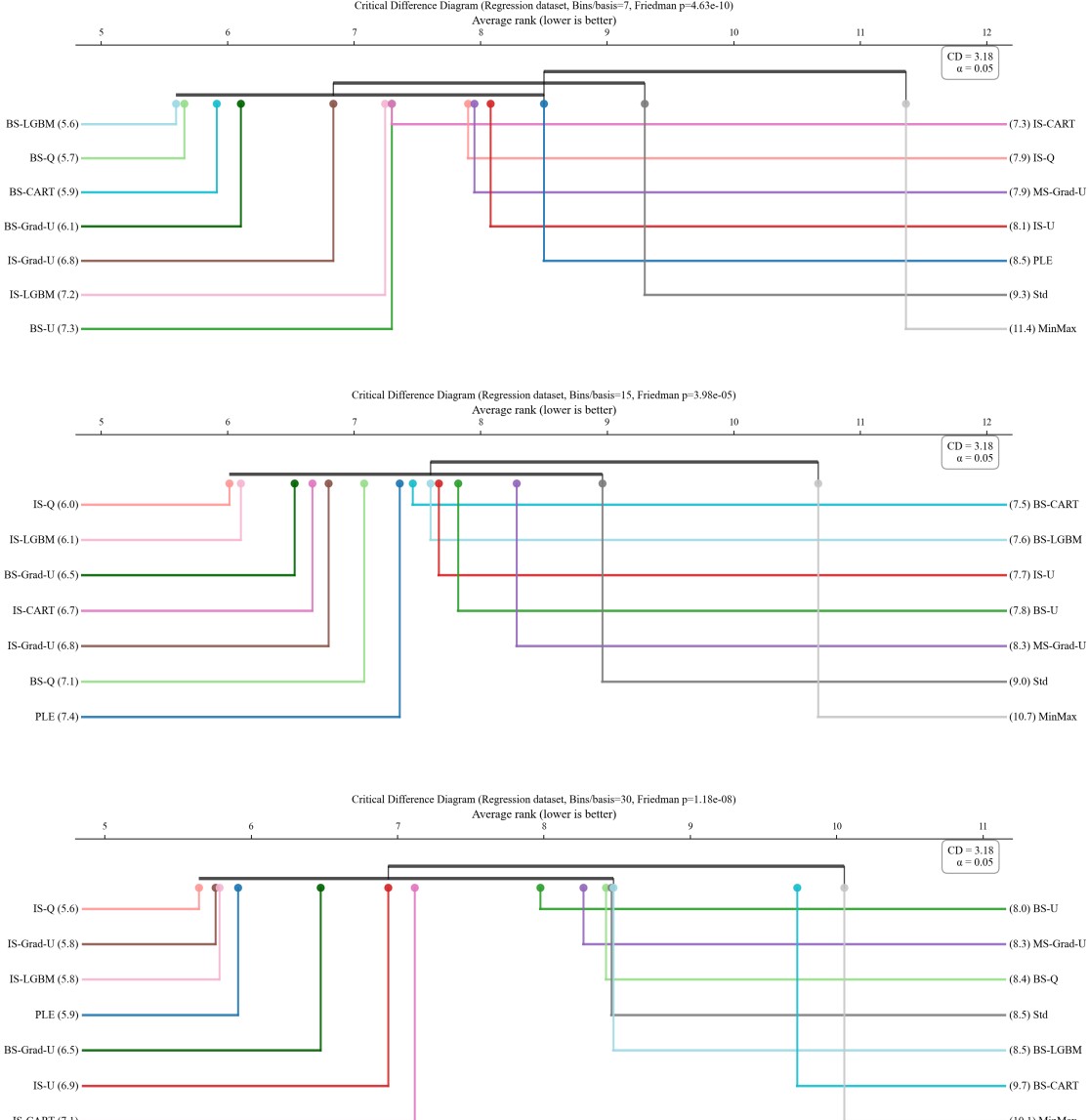

Figure 2: Regression critical difference diagrams. CD diagrams aggregated over all backbones for output sizes $m \in \{7, 15, 30\}$. Lower average rank indicates better overall performance. Methods connected by a horizontal bar are not significantly different under the Nemenyi test. Preprocessing abbreviations are given in Table 3, and detailed regression results for the corresponding output sizes are provided in Tables 11, 12, and 13.

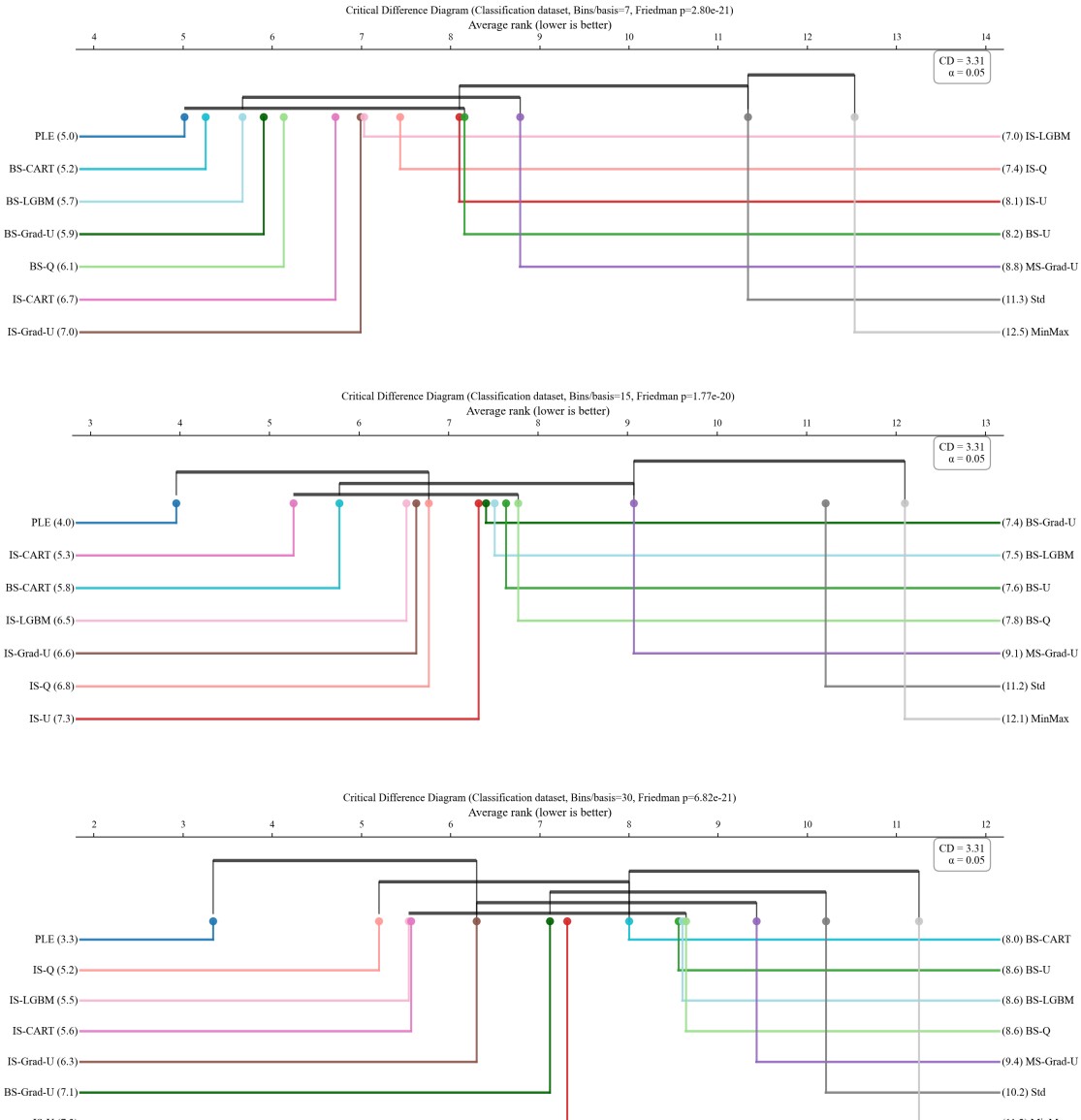

Figure 3: Critical difference diagrams for classification. The diagrams are aggregated over all backbones for output sizes $m \in \{7, 15, 30\}$. Lower average rank indicates better overall performance. Methods connected by a horizontal bar are not significantly different under the Nemenyi test. Preprocessing abbreviations are given in Table 3, and detailed classification results for the corresponding output sizes are reported in Tables 14, 15, and 16.

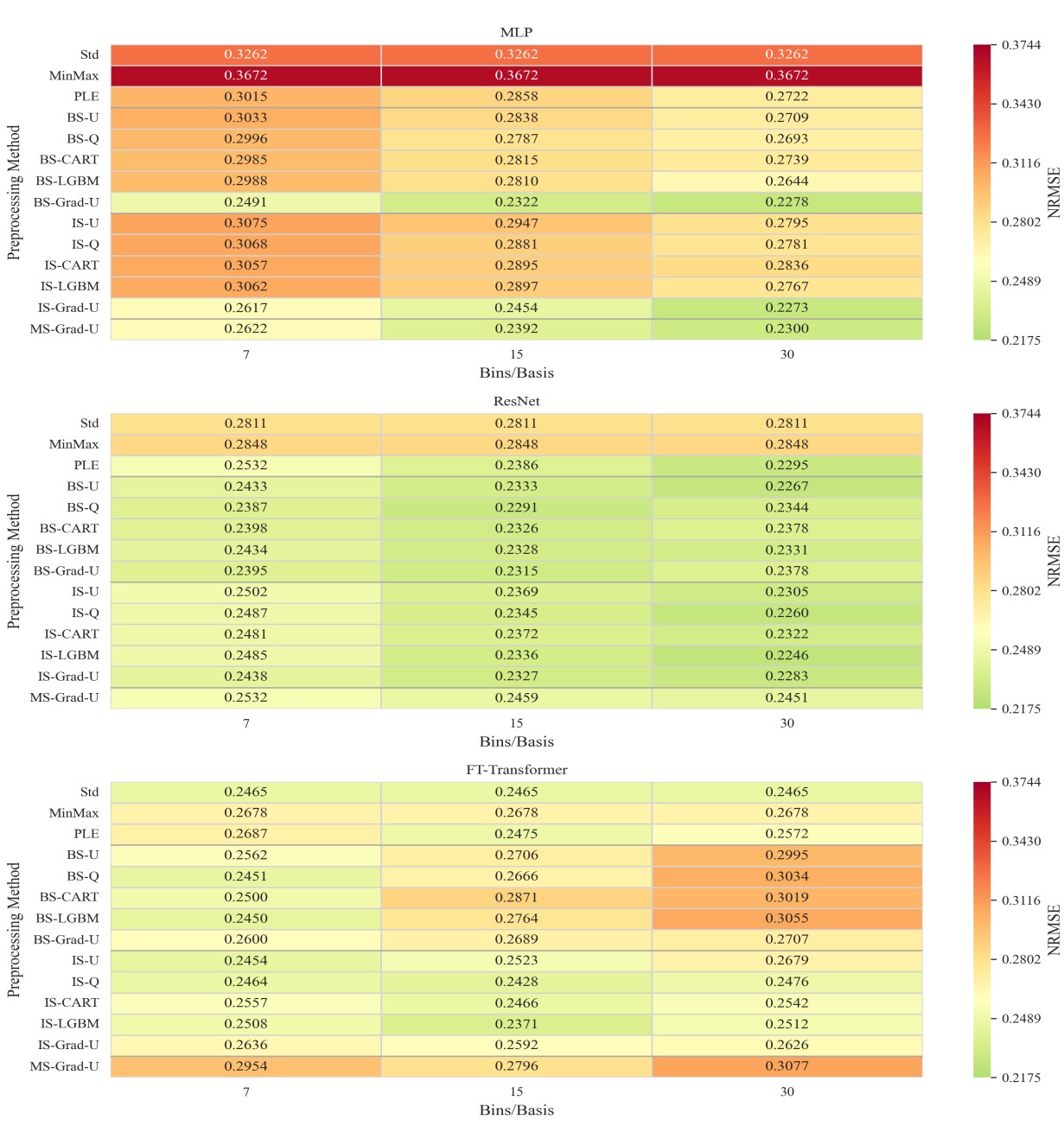

Figure 4: Average regression performance across backbones and output sizes. Each heatmap cell shows the mean test NRMSE ($\downarrow$) across all regression datasets for a given backbone, preprocessing method, and output size $m \in \{7, 15, 30\}$. Preprocessing abbreviations are given in Table 3, and detailed results are provided in Tables 11, 12, and 13.

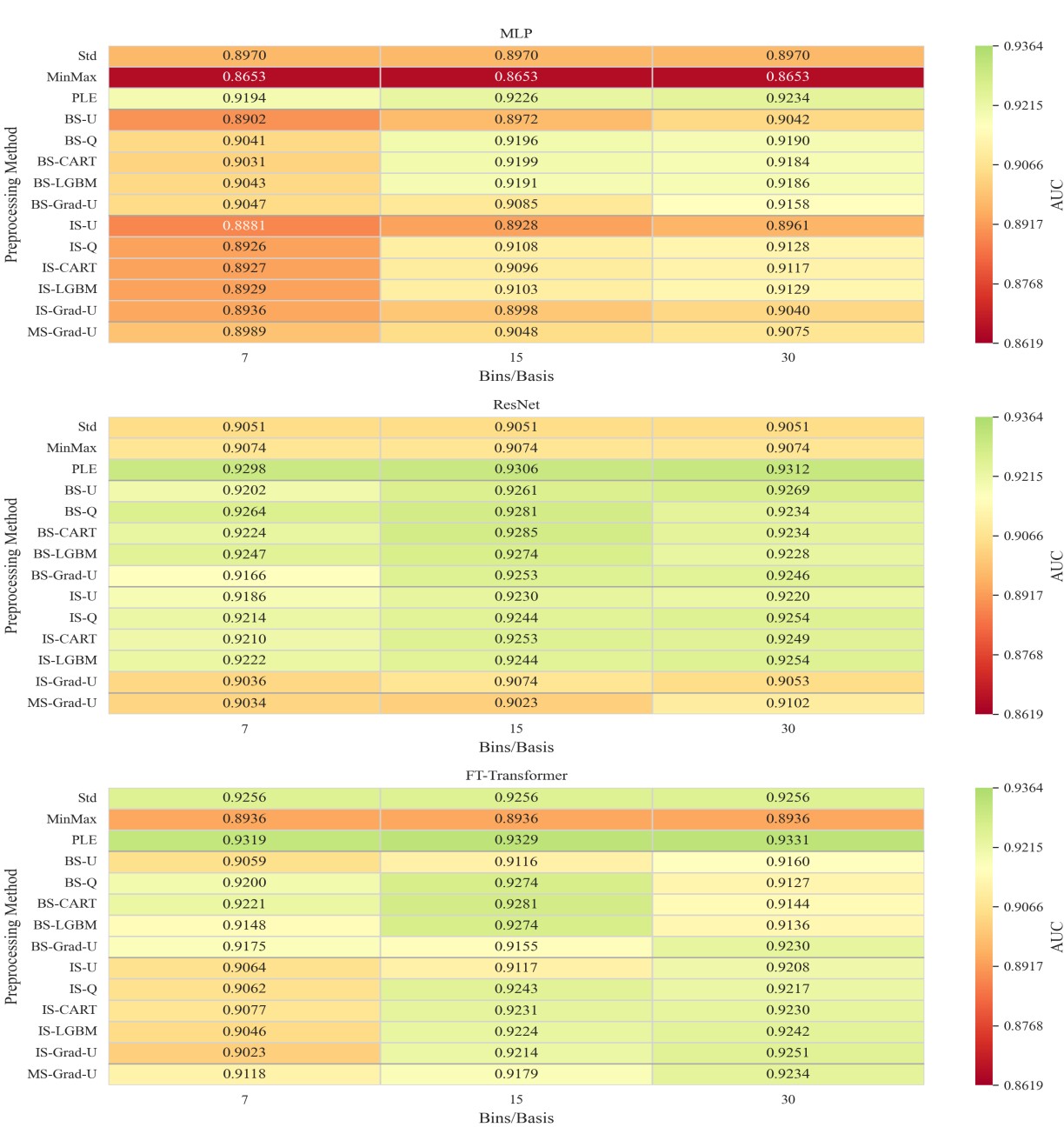

Figure 5: Average classification performance across backbones and output sizes. Each heatmap cell shows the mean test AUC (↑) across all classification datasets for a given backbone, preprocessing method, and output size $m \in \{7, 15, 30\}$. For multiclass datasets, AUC is computed as weighted one-vs-rest ROC-AUC. Higher values indicate better performance. Preprocessing abbreviations are given in Table 3, and detailed results are provided in Tables 14, 15, and 16.

### 4.4 Illustration of PLE and B-Spline Fits on Simple Synthetic Problems

To better understand the task-dependent behavior seen in the benchmark results, we compare PLE and cubic B-spline encodings on two simple one-dimensional synthetic problems. The goal is not to introduce another benchmark, but to provide a small controlled illustration of how the two encodings behave when the representation size is held fixed.

We consider one regression problem with a smooth nonlinear target and one classification problem with a class-probability function that contains flat regions and relatively sharp transitions. In both cases, we use the same encoding output size ($m = 10$), with uniform PLE bins and a clamped uniform cubic B-spline basis. Since this experiment is intended to illustrate the behavior of the encodings themselves rather than to reproduce the full benchmark setting, we use simple downstream models, namely Ridge regression for the regression task and logistic regression for the classification task. This keeps the comparison centered on the encoding and avoids additional effects from backbone expressiveness. Details of the synthetic data generation and preprocessing are provided in Appendix J.3.

Figure 6 shows the resulting fits. In the regression example (Fig. 6a), the B-spline basis gives a smoother fit and stays closer to the target curve, while the PLE fit shows more visible piecewise-linear changes at the bin boundaries. In the classification example (Fig. 6b), PLE follows the flat high-probability region and the sharper boundary changes more closely, whereas the B-spline fit changes more gradually across these regions. This difference is consistent with the structure induced by the two encodings. With logistic regression, PLE produces a piecewise-linear score function, and its cumulative bin construction can make it a natural match for threshold-like probability structure. By contrast, a cubic B-spline basis encourages a smoother local polynomial fit, which is often better aligned with smoothly varying regression targets.

Although these examples are intentionally simple, they are consistent with the broader pattern in the benchmark results. PLE is the most robust choice for classification, while spline-based variants are often more competitive on regression. This suggests that part of the difference may come from the kind of fitted function encouraged by the encoding itself. These plots are intended only as an illustration and should be read as a qualitative complement to the main benchmark rather than as a separate evaluation.

### 4.5 Efficiency Case Study: Learnable-Knot Overheads on SGEMM

We conduct an efficiency case study on SGEMM GPU, one of the 25 benchmark datasets, to examine the computational cost of learnable-knot spline encodings. SGEMM is a regression dataset with $d = 14$ numerical features. Dataset details are provided in Table 4. The analysis has three parts. We first summarize the asymptotic per-batch complexity of the preprocessing methods. We then quantify the additional parameter count introduced by learnable knots. Finally, using timestamps logged during training, we measure total GPU wall-clock time over 5-fold cross-validation.

As reference methods, we include Std, MinMax, and PLE together with selected spline-based encodings. The main comparison is between the learnable-knot variants BS-Grad-U and IS-Grad-U and their fixed-knot counterparts BS-U and IS-U. We also include MS-Grad-U to compare computational cost across learnable-knot spline families. This setup lets us separate asymptotic cost, parameter overhead, and observed runtime, and study how learnable-knot preprocessing scales with output size relative to fixed-knot baselines and standard reference methods.

#### 4.5.1 Asymptotic complexity

Table 1 summarizes the asymptotic per-batch complexity of the preprocessing methods. Let $d$ denote the number of numerical features, $B$ the batch size, $m$ the number of output bins or basis functions per feature, $p$ the spline degree, and $K = m - p - 1$ the number of internal knots; see Appendix C.2. For fixed preprocessing methods such as Std, MinMax, and PLE, the cost is given by applying the corresponding feature transformation. For fixed-knot spline expansions, the dominant cost is basis evaluation, which scales as $O(d\,B\,m\,p)$. This applies to B-, M-, and I-spline bases, since all three use $m = K + p + 1$ basis functions per feature and share the same leading dependence on $(d, B, m, p)$.

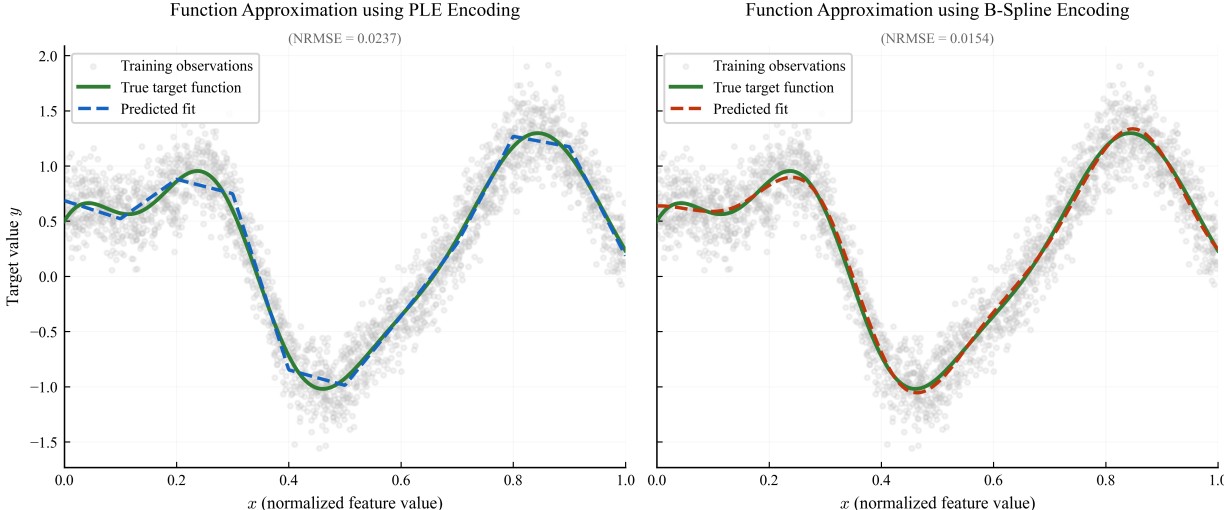

(a) Regression: smooth target approximation.

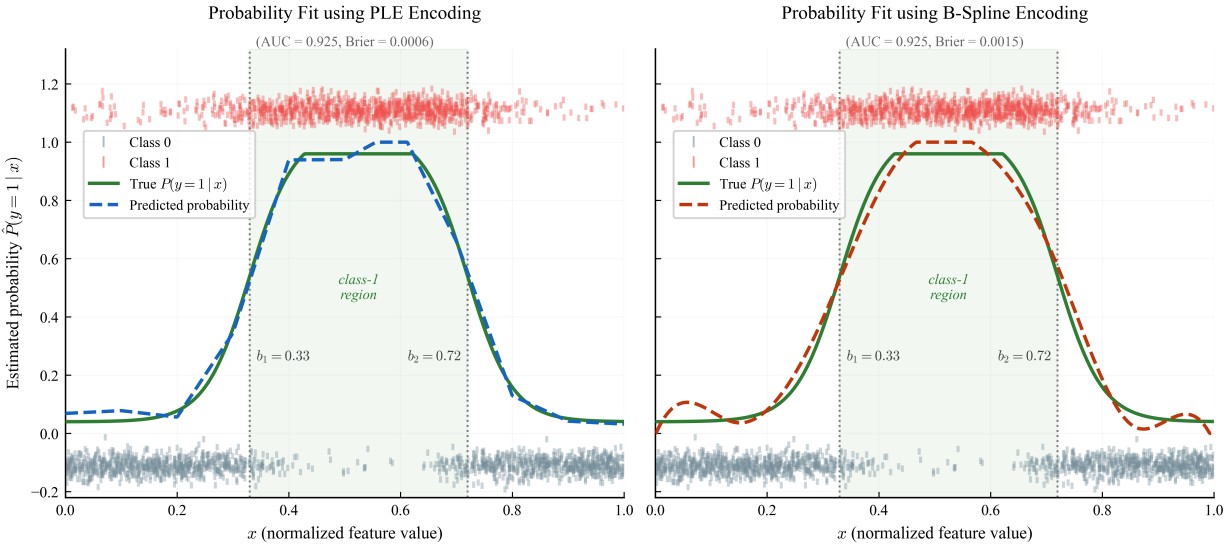

(b) Classification: threshold-like probability structure.

Figure 6: PLE and cubic B-spline fits on simple synthetic examples with the same basis budget ($m = 10$). (a) Regression example with a smooth nonlinear target, fitted using Ridge regression. The B-spline fit is smoother and tracks the target more closely, whereas the PLE fit shows piecewise-linear changes at the bin boundaries. (b) Classification example with a piecewise-constant class-probability function, fitted using logistic regression. The PLE fit better captures the sharp transitions and flat regions, whereas the B-spline fit changes more smoothly across the boundaries. This figure is intended as an illustration of the different behaviors of the two encodings rather than as a benchmark result.

For learnable-knot variants, the spline transform is part of the trainable computation graph, and additional overhead arises from differentiation with respect to knot parameters. Denoting the number of learnable internal knot parameters by $n_{\text{int}}$, this overhead appears in the forward and backward passes as summarized in Table 1. In our parameterization, $n_{\text{int}}$ is proportional to $K$; see equation 10 and equation 12. The table reports per-batch cost once knot optimization is active and excludes one-time initialization costs. In our training setup, knot updates are activated only after an initial warm-up phase, so the measured end-to-end

| Preprocessing variant | Transformation cost | Forward | Backward |
|---|:---:|:---:|:---:|
| Std | $O(d\,B)$ | – | – |
| MinMax | $O(d\,B)$ | – | – |
| PLE | $O(d\,B\,m)$ | – | – |
| Fixed knots | $O(d\,B\,m\,p)$ | – | – |
| Learnable knots | – | $O(d\,B\,m\,p) + O(d\,n_{\text{int}})$ | $O(d\,B\,m\,p) + O(d\,B\,n_{\text{int}})$ |

Table 1: Asymptotic time complexity per batch. Here, $d$ denotes the number of numerical features, $B$ the batch size, $m$ the per-feature output size, $p$ the spline degree, and $n_{\text{int}}$ the number of learnable internal knot parameters. Fixed knots refer to the B-, M-, and I-spline variants with uniform, quantile, and target-aware knot placement, while learnable knots refer to the gradient-based variants. Fixed preprocessing methods incur only transformation cost, whereas learnable-knot variants add forward and backward overhead during joint training with the backbone. Preprocessing abbreviations are given in Table 3.

runtime is lower than it would be if learnable-knot optimization were active from the first epoch. Although the three spline families share the same asymptotic order in our formulation, M-splines and I-splines incur larger constant factors due to normalization and cumulative or integral structure, which is reflected in the wall-clock measurements.

### 4.5.2 Parameter overhead of learnable knots

For learnable-knot variants, the additional parameters arise solely from making knot locations trainable and are independent of the downstream backbone. Under the softmax–cumsum parameterization in equation 10 and equation 12, we learn one scalar per interval width, giving $K + 1$ learnable parameters per numerical feature and therefore $d(K + 1) = d(m - p)$ additional parameters in total. For SGEMM, with $d = 14$ and $p = 3$, this corresponds to 56 extra parameters at $m = 7$, 168 at $m = 15$, and 378 at $m = 30$. This overhead is negligible relative to the backbone sizes, which range from approximately 66K to 1.13M parameters. Thus, learnable-knot variants primarily increase optimization cost rather than model capacity.

### 4.5.3 Wall-clock training time

Figure 7 reports total GPU wall-clock time over all five folds. Two effects drive the overall runtime. First, increasing the per-feature output size from $m \in \{7, 15, 30\}$ expands the numerical representation from $d = 14$ raw inputs to $dm \in \{98, 210, 420\}$ basis coordinates. This increases the computational load of the downstream backbone even when knots are fixed. On SGEMM, the effect is modest for MLP and ResNet, but clearly visible for FT-Transformer, where PLE and fixed-knot spline variants also become slower at larger $m$. Second, learnable-knot variants add backward-pass overhead through the knot parameters. Since BS-Grad-U, MS-Grad-U, and IS-Grad-U introduce the same number of additional knot parameters at a given $m$, their runtime differences are not explained by parameter count alone. They are more consistent with differences in basis-specific computation and the structure of the knot gradients.

Among the learnable-knot methods, BS-Grad-U is consistently the cheapest. Its runtime stays relatively stable for MLP and ResNet and increases only moderately for FT-Transformer. By contrast, MS-Grad-U and especially IS-Grad-U become much slower as $m$ increases, with the largest gaps appearing for the MLP and, at $m = 30$, also for FT-Transformer. This suggests that the dominant overhead comes from the computational structure of the spline family rather than from the number of learnable knot parameters.

**Why BS-Grad-U is cheaper than MS-Grad-U and IS-Grad-U.** The separation in wall-clock time is consistent with the definitions of the three spline families and becomes more pronounced as $m$ increases. B-splines have the most local computation. Under the Cox de Boor recursion, a degree-$p$ basis function depends only on a local subset of knots and is nonzero on at most $(p + 1)$ consecutive knot intervals. When knots are learned, a perturbation therefore affects only a limited neighborhood of basis functions, which

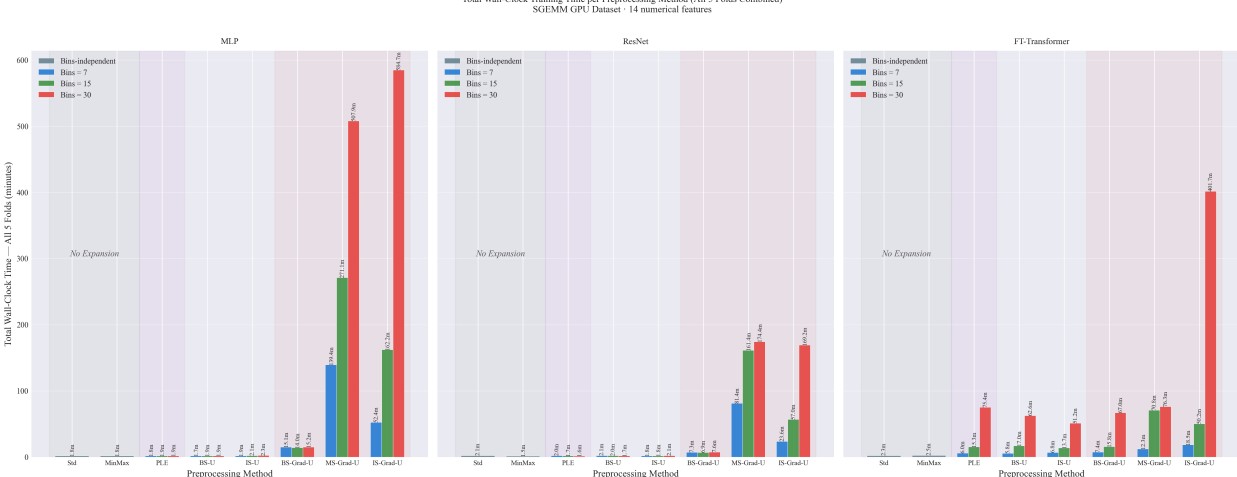

Figure 7: SGEMM efficiency case study: total GPU wall-clock time over 5-fold cross-validation. Comparison of Std, MinMax, PLE, fixed-knot spline variants (BS-U and IS-U), and learnable-knot spline variants (BS-Grad-U, MS-Grad-U, and IS-Grad-U) across basis budgets $m \in \{7, 15, 30\}$ for MLP, ResNet, and FT-Transformer. Values denote total end-to-end training time across all five folds. For learnable-knot variants, timings include the initial warm-up phase followed by joint optimization of knot parameters and backbone weights.

keeps the backward pass comparatively cheap. M-splines add a knot-dependent normalization factor,

$$M_{j,\ell}^{(p)}(x_j) = \frac{p+1}{\tau_{j,\ell+p+1} - \tau_{j,\ell}} \, B_{j,\ell}^{(p)}(x_j),$$

so gradients must propagate not only through the B-spline recursion but also through the knot-dependent denominator. I-splines inherit this normalization and additionally introduce cumulative dependence through the integral structure,

$$I_{j,\ell}^{(p)}(x_j) = \int_{-\infty}^{x_j} M_{j,\ell}^{(p)}(t) \, dt.$$

As a result, their knot gradients pass through a less local computation graph with higher backward-pass cost.

These differences become more visible at larger $m$. Increasing $m$ enlarges both the expanded representation and, through $K = m-p-1$, the number of internal knots. For BS-Grad-U, the added work remains relatively local. For MS-Grad-U and IS-Grad-U, the normalization and cumulative structure make this growth more expensive. This matches the stronger runtime increase observed for MS-Grad-U and IS-Grad-U in Fig. 7.

**Backbone-dependent overhead.** The downstream backbone also affects how these preprocessing costs appear in wall-clock time. With the expanded representation, the MLP consumes the full $d \times m$ input through a dense layer, so gradients from all expanded numerical features are mixed immediately before reaching the spline layer in the backward pass. This can make the more expensive knot-gradient computations of MS-Grad-U and IS-Grad-U more visible. ResNet may moderate this effect through its residual structure, while FT-Transformer processes features more independently before mixing them through attention. We do not attribute the backbone-specific differences to a single mechanism, since wall-clock time also depends on implementation details and optimization dynamics. Still, the larger overhead observed for the MLP is consistent with this interpretation.

**Practical takeaway.** The results suggest that numerical preprocessing should be selected jointly with the task, backbone, and available compute budget rather than treated as a fixed preprocessing step. Explicit encodings are most useful for representation-limited backbones such as MLPs and ResNet-style models

Table 2: Practical recommendations for selecting numerical preprocessing methods based on task, backbone, and computational budget.

| Factor | Setting | Start with | Guideline |
|---|---|---|---|
| Backbone | MLP or ResNet-style models | Explicit encodings | These models benefit most from numerical encodings because their input representation is relatively simple. Use PLE for classification and B-splines for regression, starting with small output sizes such as $m \approx 7$–10. |
| Backbone | Feature-tokenizing or high-capacity models | Std or MinMax | For models such as FT-Transformer, TabM, TabTransformer, or SAINT, simple preprocessing is a strong first choice. Add PLE or fixed-knot splines only if the simple baseline is insufficient. |
| Task | Classification | PLE | PLE is the strongest default for classification, with robust rankings across output sizes and backbones. It should usually be tried before more expensive spline variants. |
| Task | Regression | B-splines | Start with B-splines using uniform or quantile knot placement. With larger budgets, include target-aware knots and I-splines in the sweep. |
| Budget | Learnable-knot variants | BS-Grad-U | Among learnable-knot variants, BS-Grad-U provides the most favorable tradeoff between efficiency and performance. MS-Grad-U and IS-Grad-U should be treated as high-cost options. |
| Caution | M-splines | Not first choice | M-splines are not recommended as the default. In our experiments, both fixed and learnable M-spline variants were less attractive due to stability and efficiency concerns. |

(Gorishniy et al., 2021), while their marginal benefit is often smaller for architectures with stronger internal feature representations or tokenization mechanisms, such as FT-Transformer, TabM, TabTransformer, or SAINT (Gorishniy et al., 2021; 2025; Huang et al., 2020; Somepalli et al., 2021). As a practical starting point, PLE is the most reliable default for classification, whereas B-splines with uniform or quantile knots provide a strong and efficient default for regression. More expensive variants, including I-splines and learnable-knot splines, are better viewed as second-stage options once a stable model and training configuration have been established. A compact summary of these recommendations is given in Table 2.

## 5   Ablation study

To complement the main benchmark, we study how predictive performance changes with encoding resolution in a controlled synthetic regression setting. This allows us to isolate the effect of numerical feature encodings under a known input distribution and target structure.

**Synthetic regression setup.** We use a synthetic regression task to examine how performance changes with numerical encoding resolution in a controlled setting. The informative feature follows a skewed, non-uniform distribution, and the target combines smooth nonlinear variation, a threshold effect, and a localized peak. Detailed data generation and a visualization of the dataset are provided in Appendix K. We use the same MLP architecture and training setup as in the main experiments and vary only the encoding resolution, with $m \in \{5, 10, 15, 20, 25, 30, 35, 40, 45, 50\}$.

**Compared methods and reporting.** We compare Std, MinMax, and PLE with spline-based encodings using different knot-placement strategies. The sweep includes three reference methods without an output-size grid, namely Std, MinMax, and $\text{PLE}_{\text{adp}}^{50}$, together with 16 methods evaluated over $m \in \{5, 10, 15, 20, 25, 30, 35, 40, 45, 50\}$. This yields 163 method-resolution configurations in total. Each configuration is run with five random seeds, resulting in 815 training runs overall. Results are reported as

mean test NRMSE, with shaded bands indicating $\pm$ one standard deviation across seeds. For Std and Min-Max, output size is not applicable, while for $\text{PLE}^{50}_{\text{adp}}$ the maximum number of bins is capped at 50 and the effective discretization is determined adaptively by tree-guided splits. All remaining optimization settings follow the main experiments. The resulting trends are shown in Fig. 8, with the corresponding numerical results reported in Table 17.

**Main observations.** Figure 8 shows that, for most methods, test NRMSE improves as $m$ increases from 5 to roughly 15–35, after which the curves mostly plateau. This pattern is clearest for B-spline and I-spline variants, while PLE shows a similar but slightly flatter trend. The choice of knot-placement strategy mainly shifts the performance level within a spline family rather than changing the overall shape of the resolution curve. In this synthetic setting, CART-based and uniform placement give the strongest results.

Among all configurations, B-spline variants are the strongest overall. The best result is obtained by BS-CART at $m = 30$ with NRMSE $0.0456 \pm 0.0014$, and all top five settings are B-spline based, specifically BS-CART and BS-U. Several I-spline variants remain competitive, but they remain slightly above the best B-spline results within the same knot-placement group. By contrast, M-spline variants are generally weaker and often deteriorate at larger output sizes, with visibly wider uncertainty bands. One possible reason is the knot-dependent normalization factor $(\tau_{j,\ell+p+1} - \tau_{j,\ell})^{-1}$ in the M-spline definition in Appendix C.4, which may increase numerical sensitivity when adjacent knots become close. This larger variance is not apparent for MS-Grad-U, likely because the learnable-knot variant uses the LayerNorm-based stabilization described in Section 4. We do not investigate this effect further here.

**Scope of the main benchmark.** This ablation is consistent with the design choices made in the main benchmark. In particular, fixed-knot M-spline variants are not included there because, in this synthetic study, they are less stable and less competitive than the corresponding B-spline and I-spline variants, especially at larger output sizes. We nevertheless retain the learnable-knot M-spline variant, MS-Grad-U, as a reference point for end-to-end knot optimization, together with the numerical stabilization described in Section 4.

# 6 Conclusion

In this work, we showed that numerical encoding is an important modeling choice in tabular deep learning, not merely a preprocessing detail. Across regression and classification tasks, explicit numerical encodings often outperform standard scaling, with spline-based methods providing a strong and flexible alternative. The results also show that the best encoding depends on the task, backbone, output size, knot-placement strategy, and computational budget. In classification, PLE emerges as a robust default, while in regression B-spline variants provide an efficient and competitive starting point, with I-spline and learnable-knot variants becoming useful in selected higher-budget settings. Our ablation study further shows that increasing the encoding resolution is beneficial up to a moderate range, after which gains tend to plateau, and that B-spline and I-spline variants are generally more stable than M-spline variants. Overall, the findings suggest that numerical preprocessing should be treated as part of the model design space and selected deliberately rather than fixed by convention.

# 7 Limitations and Future Work

**Limitations.** Our study covers only part of the design space of numerical preprocessing for tabular deep learning. We focus on a selected set of encoding families, knot-placement strategies, output sizes, and standard trainable backbones, so the efficiency analysis should be read as a relative comparison within this setup rather than as a universal runtime benchmark. The synthetic ablation is intentionally controlled and therefore does not capture the full heterogeneity of real-world tabular data. In addition, all encodings considered here are feature-wise. Each numerical feature is transformed independently, and any cross-feature interaction must therefore be discovered later by the backbone. This keeps the benchmark tractable, but it excludes joint encodings of numerical interactions, such as tensor-product or grid-based constructions, whose dimensionality grows quickly with the number of encoded features and basis functions.

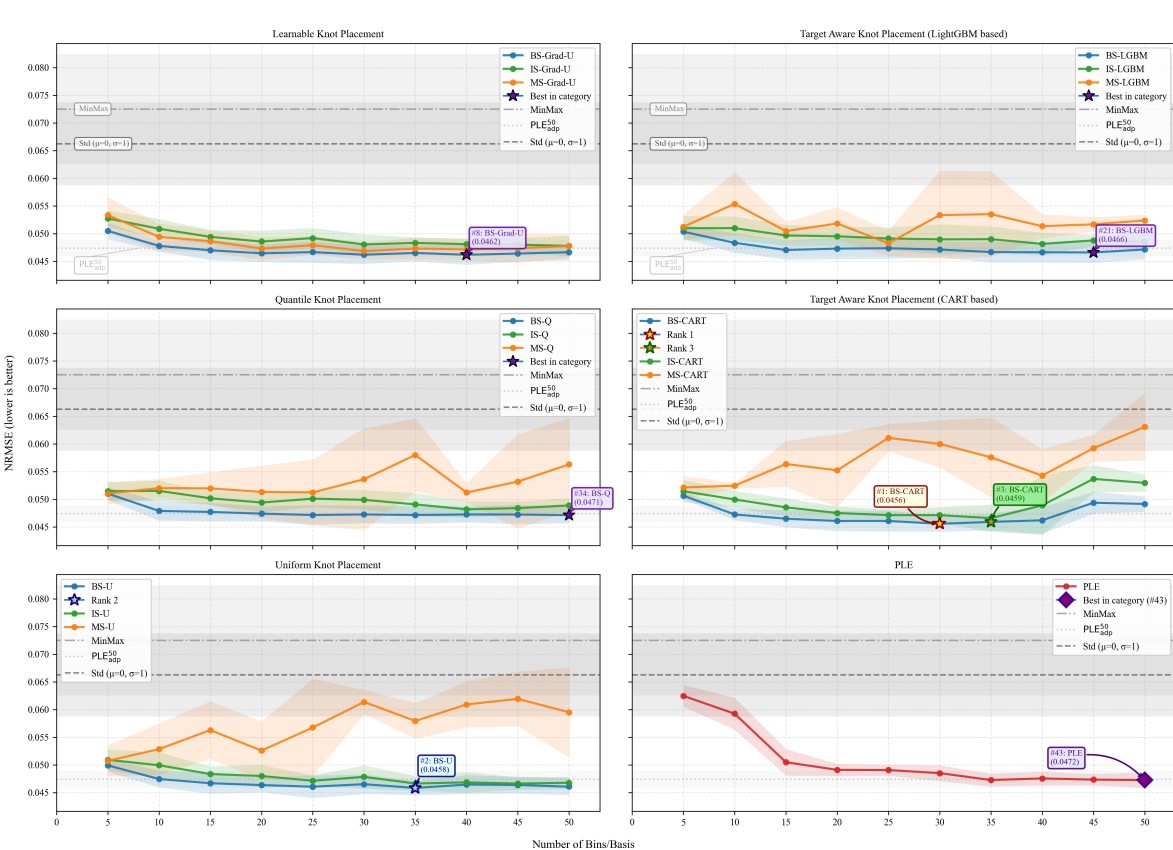

Figure 8: Sensitivity to basis resolution on a synthetic regression task. Test NRMSE (mean ± std over 5 seeds) for PLE and spline-based encodings as the number of bins or basis functions varies over $\{5, 10, 15, 20, 25, 30, 35, 40, 45, 50\}$. Results are grouped by knot-selection strategy. Dotted horizontal lines show the Std, MinMax, and $PLE_{adp}^{50}$ baselines. Preprocessing abbreviations are given in Table 3. All results use an MLP backbone.

**Future work.** Several extensions are natural. A first direction is to study multidimensional numerical encodings that model selected low-order interactions directly, for example tensor-product spline bases, ideally combined with sparse interaction selection or low-rank parameterizations to avoid the full combinatorial cost (Wood, 2017; Wang et al., 2021; Cheng et al., 2024). A second direction is to relax the current uniform per-feature encoding design and allow feature-specific choices of encoding family and output size, so that representation capacity can adapt to the role of each numerical column. It would also be valuable to compare a broader set of basis families and regularized learnable-knot parameterizations, including thin-plate splines and radial basis functions (Wood, 2003; Buhmann, 2000; Zheng et al., 2025). More broadly, the present findings should be re-examined on newer model regimes with stronger internal representation learning, where the marginal value of explicit preprocessing may differ.

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

## A    Preprocessing Abbreviations

For clarity and consistency, we refer to preprocessing methods throughout the paper using abbreviated names such as Std, MinMax, PLE, BS-*, IS-*, and MS-*. The complete mapping is provided in Table 3.

## B    Dataset Details

We benchmark on 25 tabular datasets, including 13 regression and 12 classification datasets, of which 3 are multiclass. The datasets are drawn from OpenML and the UCI Machine Learning Repository.[1][2] Table 4 reports per-dataset statistics, including the numbers of numerical and categorical features, split sizes, and class imbalance where applicable. Samples with missing values are removed. For classification datasets, we report the dominant-class ratio as a measure of class imbalance. Unless stated otherwise, numerical features are scaled to $[0, 1]$ before applying feature-expansion methods such as splines and PLE, while the baseline pipelines use standardization (Std) or min-max scaling (MinMax). For the Shuttle dataset, we randomly subsample 25K instances while preserving class proportions to control computational cost. Table 5 summarizes the overall scale and feature dimensionality of the benchmark suite.

---

[1] `https://www.openml.org`
[2] `https://archive.ics.uci.edu`

| Category | Method | Description | Target-aware | Learnable-knot |
|---|---|---|---|---|
| Baseline | Std | Standardization (z-score) | – | – |
| | MinMax | min-max scaling to $[0, 1]$ | – | – |
| | PLE | Piecewise Linear Encoding | ✓ | – |
| | $\text{PLE}_{\text{adp}}^{50}$ | Adaptive PLE with $n_{\text{bins}} \in [5, 50]$ selected by tree splits (Table 8) | ✓ | – |
| B-Spline | BS-U | Uniform knot placement | – | – |
| | BS-Q | Quantile-based knot placement | – | – |
| | BS-CART | CART-based target-aware knot placement | ✓ | – |
| | BS-LGBM | LightGBM-based target-aware knot placement | ✓ | – |
| | BS-Grad-U* | Uniform initialization with end-to-end knot optimization | – | ✓ |
| I-Spline | IS-U | Uniform knot placement | – | – |
| | IS-Q | Quantile-based knot placement | – | – |
| | IS-CART | CART-based target-aware knot placement | ✓ | – |
| | IS-LGBM | LightGBM-based target-aware knot placement | ✓ | – |
| | IS-Grad-U* | Uniform initialization with end-to-end knot optimization | – | ✓ |
| M-Spline | MS-U | Uniform knot placement | – | – |
| | MS-Q | Quantile-based knot placement | – | – |
| | MS-CART | CART-based target-aware knot placement | ✓ | – |
| | MS-LGBM | LightGBM-based target-aware knot placement | ✓ | – |
| | MS-Grad-U* | Uniform initialization with end-to-end knot optimization | – | ✓ |

**Note.** "–" indicates that the option is not applicable. "Target-aware" denotes knot placement based on target-dependent split points. "Learnable-knot" denotes variants in which internal knot locations are optimized jointly with the downstream model during training. Methods marked with * use uniform knot placement for initialization.

Table 3: Preprocessing abbreviations used throughout the paper. The table summarizes the naming convention for baseline, spline-based, target-aware, and learnable-knot variants.

| Dataset | Abbr. | #cat | #num | Train | Val | Test | Ratio | Reference / OpenML ID |
|---------|-------|------|------|-------|-----|------|-------|-----------------------|
| **Regression** | | | | | | | | |
| Abalone | AB | 1 | 7 | 3008 | 334 | 835 | – | Nash et al. (1994) |
| California Housing | CA | 1 | 8 | 14861 | 1651 | 4128 | – | OpenML: 45028 |
| CPU Small | CPU | 0 | 12 | 5899 | 655 | 1638 | – | OpenML: – |
| Diamonds | DI | 3 | 6 | 38837 | 4315 | 10788 | – | OpenML: 44979 |
| House Sales | HS | 0 | 18 | 15562 | 1729 | 4322 | – | OpenML: 42092 |
| Parkinsons | PA | 0 | 19 | 4230 | 470 | 1175 | – | Tsanas & Little (2009) |
| Wine Quality | WI | 0 | 11 | 4679 | 519 | 1299 | – | Cortez et al. (2009) |
| House8L | H8 | 0 | 8 | 16405 | 1822 | 4556 | – | OpenML: 218 |
| Pulsar | PU | 0 | 8 | 12888 | 1431 | 3579 | – | OpenML: 45558 |
| Sulphur | SU | 0 | 6 | 7259 | 806 | 2016 | – | OpenML: 44020 |
| FIFA Wage | FW | 0 | 5 | 13006 | 1445 | 3612 | – | OpenML: 44026 |
| SGEMM GPU | SG | 0 | 14 | 14400 | 1600 | 4000 | – | OpenML: 44961 |
| Protein | PR | 0 | 9 | 32926 | 3658 | 9146 | – | OpenML: 44963 |
| **Classification** | | | | | | | | |
| Adult | AD | 8 | 5 | 35167 | 3907 | 9768 | 76.1% | Becker & Kohavi (1996) |
| Bank | BA | 8 | 7 | 32553 | 3616 | 9042 | 88.3% | Moro et al. (2014) |
| Churn | CH | 2 | 8 | 7200 | 800 | 2000 | 79.6% | OpenML: 46911 |
| FICO | FI | 0 | 23 | 7532 | 836 | 2091 | 52.2% | OpenML: 45554 |
| Marketing | MA | 7 | 7 | 31100 | 3455 | 8638 | 88.4% | OpenML: – |
| EEG Eye State | EEG | 0 | 14 | 10786 | 1198 | 2996 | 55.1% | OpenML: 1471 |
| Gamma Telescope | GT | 1 | 9 | 9549 | 1060 | 2652 | 50.4% | OpenML: 44085 |
| IPUMS (LA 97) | IP | 1 | 19 | 3730 | 414 | 1036 | 50.1% | OpenML: 44084 |
| Loan Status | LS | 5 | 8 | 18905 | 2100 | 5251 | 77.7% | OpenML: 44556 |
| **Multiclass** | | | | | | | | |
| Air Quality (4-class) | AQ | 1 | 8 | 3600 | 400 | 1000 | 40.0% | OpenML: 46880 |
| Loan Type (7-class) | LT | 0 | 6 | 6154 | 683 | 1709 | 27.9% | OpenML: 46511 |
| Shuttle (7-class) | SH | 0 | 9 | 18000 | 2000 | 5000 | 78.6% | OpenML: 40685 |

Table 4: Benchmark datasets used in the experiments. For each dataset, we report the abbreviation, the numbers of categorical (#cat) and numerical (#num) features, and the average train, validation, and test split sizes over 5-fold cross-validation. *Ratio* denotes the dominant-class percentage for classification and multiclass datasets. The last column gives the OpenML dataset ID or the corresponding UCI citation.

| Metric | Regression | Classification | Total |
|--------|-----------|----------------|-------|
| Number of datasets | 13 | 12 | 25 |
| Total samples | 255,489 | 255,928 | 511,417 |
| Avg. samples per dataset | 19,653 | 21,327 | 20,456 |
| Avg. features per dataset | 10.5 | 13.0 | 11.7 |
| Min. features | 5 | 6 | 5 |
| Max. features | 19 | 23 | 23 |

Table 5: Benchmark dataset summary. Aggregate statistics of the benchmark suite, including the number of datasets, total samples, average samples per dataset, and feature counts, reported separately for regression and classification datasets and for the full collection.

## C  Spline Basis Definitions

### C.1  Notation Used in the Methodology

Table 6 summarizes the main notation used in Section 3 and the spline-basis definitions.

### C.2  Basis Indexing and Basis Function Counts

For numerical feature $j$, the spline expansion is defined by basis functions

$$\{b_{j,\ell}^{(p)}(x; \boldsymbol{\tau}_j)\}_{\ell=1}^{m_j},$$

where $x$ denotes a scalar feature value, $\boldsymbol{\tau}_j$ denotes the knot sequence for feature $j$, $\ell$ indexes the basis functions, and $m_j$ is the resulting expansion dimension. Throughout the study, we use cubic splines with degree $p = 3$.

**B-, M-, and I-splines.**  For B-, M-, and I-splines, the number of basis functions is determined by the spline degree and the knot configuration. Let $K_j$ denote the number of internal knots for feature $j$. Under the standard open, clamped knot construction,

$$m_j = K_j + p + 1, \qquad K_j = m_j - p - 1.$$

These relations apply to all three spline families. Here, $K_j$ counts internal knots, whereas $m_j$ counts the resulting basis functions. Thus, knot indices range from 1 to $K_j$, while basis-function indices range from 1 to $m_j$.

### C.3  B-spline Basis Definition

We follow the basis indexing convention in Appendix C.2. For numerical feature $j$, let $x$ denote a scalar feature value. We use a nondecreasing knot sequence

$$\boldsymbol{\tau}_j = (\tau_{j,1}, \ldots, \tau_{j,K_j+2p+2}),$$

obtained by augmenting the $K_j$ internal knots with boundary knots repeated $p + 1$ times at each end. The B-spline basis functions are defined by the Cox–de Boor recursion.

**Zero-degree basis**

$$B_{j,\ell}^{(0)}(x) = \begin{cases} 1, & \tau_{j,\ell} \leq x < \tau_{j,\ell+1}, \\ 0, & \text{otherwise.} \end{cases}$$

**Cox–de Boor recursion** For $q \geq 1$,

$$B_{j,\ell}^{(q)}(x) = \frac{x - \tau_{j,\ell}}{\tau_{j,\ell+q} - \tau_{j,\ell}} B_{j,\ell}^{(q-1)}(x) + \frac{\tau_{j,\ell+q+1} - x}{\tau_{j,\ell+q+1} - \tau_{j,\ell+1}} B_{j,\ell+1}^{(q-1)}(x),$$

with each fraction defined as zero when its denominator is zero. The degree-$p$ B-spline basis for feature $j$ is given by

$$\{B_{j,\ell}^{(p)}(\cdot)\}_{\ell=1}^{m_j}, \qquad m_j = K_j + p + 1.$$

**Feature-wise encoding** The B-spline encoding for feature $j$ is

$$\phi_j^B(x; \boldsymbol{\tau}_j) = \left( B_{j,1}^{(p)}(x), \ldots, B_{j,m_j}^{(p)}(x) \right) \in [0,1]^{m_j}.$$

Table 6: Main notation used in the methodology.

| Symbol | Meaning |
| --- | --- |
| $n$ | Number of training samples in the current fold |
| $d$ | Number of numerical features |
| $\mathbf{x}^{(i)}$ | Input vector for sample $i$ |
| $\mathbf{x}_{\text{num}}^{(i)}$ | Numerical feature vector for sample $i$ |
| $\mathbf{x}_{\text{cat}}^{(i)}$ | Categorical feature vector for sample $i$ |
| $x_{i,j}$ | Scalar value of numerical feature $j$ for sample $i$ |
| $\mathbf{x}_j$ | Training values of numerical feature $j$ in the current fold |
| $y_i$ | Target value or class label for sample $i$ |
| $\phi_j$ | Feature-wise numerical encoding map for feature $j$ |
| $\mathbf{\Phi}$ | Concatenated encoded numerical representation |
| $m_j$ | Output size of the encoding for feature $j$ |
| $p$ | Spline degree, with $p = 3$ in this study |
| $K_j$ | Number of internal knots for feature $j$ |
| $\boldsymbol{\kappa}_j$ | Internal-knot vector for feature $j$ |
| $\kappa_{j,\ell}$ | $\ell$-th internal knot of feature $j$ |
| $\boldsymbol{\tau}_j$ | Full knot sequence for feature $j$ |
| $\tau_{j,\ell}$ | $\ell$-th entry of the full knot sequence |
| $b_{j,\ell}^{(p)}$ | $\ell$-th degree-$p$ spline basis function for feature $j$ |
| $\boldsymbol{\alpha}_j$ | Coefficient vector of a univariate spline function, used only to distinguish spline fitting from spline feature expansion |
| $\mathcal{S}_j$ | Candidate split thresholds for feature $j$ in target-aware knot placement |
| $\mathcal{C}_j$ | Candidate thresholds after sorting, deduplication, and spacing filtering |
| $G_j(s)$ | Aggregated LightGBM split gain for threshold $s$ of feature $j$ |
| $\mathbf{a}_j$ | Unconstrained learnable knot parameters for feature $j$ |
| $\mathbf{a}$ | Collection of learnable knot parameters across all numerical features |
| $\boldsymbol{\pi}_j$ | Allocation-weight vector used in the learnable-knot parameterization |
| $\mathbf{w}_j$ | Interval-width vector used to construct ordered internal knots |
| $\delta$ | Minimum spacing constant for learnable knots |
| $\lambda$ | Weight of the spacing regularization term |
| $\mathcal{R}_{\text{space}}$ | Spacing regularization term for learnable knots |
| $f_\theta$ | Downstream backbone model with parameters $\theta$ |
| $\mathcal{L}$ | Task loss used for classification or regression |

## C.4 M-spline Basis Definition

M-splines are nonnegative, locally supported basis functions normalized to integrate to one. We follow the basis indexing convention in Appendix C.2. For numerical feature $j$, let $x$ denote a scalar feature value. We use a nondecreasing knot sequence

$$\boldsymbol{\tau}_j = (\tau_{j,1}, \ldots, \tau_{j,K_j+2p+2}),$$

obtained by augmenting the $K_j$ internal knots with boundary knots repeated $p+1$ times at each end.

**Definition as normalized B-splines** Let $B_{j,\ell}^{(p)}(x)$ denote the degree-$p$ B-spline basis function defined in Appendix C.3. The corresponding M-spline basis is

$$M_{j,\ell}^{(p)}(x) = \frac{p+1}{\tau_{j,\ell+p+1} - \tau_{j,\ell}} \, B_{j,\ell}^{(p)}(x), \qquad \ell = 1, \ldots, m_j,$$

with $M_{j,\ell}^{(p)}(x) = 0$ whenever $\tau_{j,\ell+p+1} = \tau_{j,\ell}$.

**Properties**

$$M_{j,\ell}^{(p)}(x) \geq 0, \qquad \int_{-\infty}^{\infty} M_{j,\ell}^{(p)}(t) \, dt = 1.$$

**Support** Each M-spline basis function has compact support on

$$\text{supp}\left(M_{j,\ell}^{(p)}\right) = [\tau_{j,\ell}, \tau_{j,\ell+p+1}).$$

**Feature-wise encoding** The M-spline encoding for feature $j$ is

$$\phi_j^M(x; \boldsymbol{\tau}_j) = \left(M_{j,1}^{(p)}(x), \ldots, M_{j,m_j}^{(p)}(x)\right) \in \mathbb{R}_{\geq 0}^{m_j}.$$

### C.5 I-spline Basis Definition

I-splines are integrated M-splines and give monotone nondecreasing basis functions. We follow the basis indexing convention in Appendix C.2. We use the same knot sequence $\boldsymbol{\tau}_j$ and M-spline basis $M_{j,\ell}^{(p)}$ as in Appendix C.4. For numerical feature $j$, let $x$ denote a scalar feature value.

**Definition as integrated M-splines**

$$I_{j,\ell}^{(p)}(x) = \int_{-\infty}^{x} M_{j,\ell}^{(p)}(t)\, dt, \qquad \ell = 1, \ldots, m_j.$$

**Monotonicity**

$$\frac{d}{dx} I_{j,\ell}^{(p)}(x) = M_{j,\ell}^{(p)}(x) \geq 0.$$

**Feature-wise encoding** The I-spline encoding for feature $j$ is

$$\phi_j^I(x; \boldsymbol{\tau}_j) = \left(I_{j,1}^{(p)}(x), \ldots, I_{j,m_j}^{(p)}(x)\right) \in [0,1]^{m_j}.$$

## D PLE Definition

**Piecewise Linear Encoding (PLE).**

For numerical feature $j$, let $x \in \mathbb{R}$ denote a scalar feature value and let

$$b_{j,0} < b_{j,1} < \cdots < b_{j,m_j}$$

denote the bin boundaries. The PLE representation of $x$ is defined as

$$\phi_j^{\text{PLE}}(x) = (e_{j,1}(x), \ldots, e_{j,m_j}(x)) \in [0,1]^{m_j},$$

where each component is given by

$$e_{j,t}(x) = \begin{cases} 0, & x < b_{j,t-1}, \\ 1, & x \geq b_{j,t}, \\ \dfrac{x - b_{j,t-1}}{b_{j,t} - b_{j,t-1}}, & b_{j,t-1} \leq x < b_{j,t}. \end{cases}$$

**Interpretation.** The encoding can be viewed as a cumulative piecewise-linear representation. Components corresponding to bins strictly to the left of $x$ are fully activated, components to the right are inactive, and the component associated with the interval containing $x$ is linearly interpolated.

**Properties.** For all $t = 1, \ldots, m_j$,

$$0 \leq e_{j,t}(x) \leq 1.$$

Moreover, at most one component is fractional:

$$\sum_{t=1}^{m_j} \mathbb{I}\big[0 < e_{j,t}(x) < 1\big] \leq 1.$$

Thus, PLE produces a cumulative representation with a single piecewise-linear transition around the interval containing $x$.

# E    Construction of the Numerical-Encoding Illustration

Figure 1 is a schematic illustration of how a single scalar value is transformed by different numerical encodings. It is constructed from a small synthetic one-dimensional sample rather than from a benchmark dataset, and is intended only to illustrate the representations produced by the encodings. No target variable, train/test split, or predictive model is used for this figure.

We generate 44 feature values on the normalized interval $[0,1]$ from a two-component Beta mixture with components Beta$(2.6, 4.0)$ and Beta$(5.0, 2.6)$, using equal mixture weights and a fixed random seed. The values are clipped to $[0.015, 0.985]$ and shown as the data rug in the panels. For the original-data panel, the same values are displayed on a raw scale $[0, 100]$. A single example value, corresponding to raw value 58 and normalized value $x_{i,j} = 0.58$, is marked by the red dashed line and evaluated under each encoding.

For Std and MinMax, the figure shows the corresponding scalar affine transformations. For PLE and the spline-based encodings, we use a common output size $m_j = 6$. PLE uses six uniformly spaced intervals on $[0,1]$ and is evaluated as a cumulative piecewise-linear encoding. No target-aware binning is used in this illustration. For the spline panels, we use cubic B-, M-, and I-spline bases with degree $p = 3$. Since $m_j = 6$, the number of internal knots is

$$K_j = m_j - p - 1 = 2.$$

The two internal knots are placed uniformly inside $[0,1]$, and the corresponding open, clamped knot sequence is used for basis evaluation.

Each expansion encoding is evaluated on a dense grid over $[0,1]$ to draw the bin ramps or basis functions. The bars show the encoded coordinates obtained by evaluating the corresponding encoding at $x_{i,j} = 0.58$. Thus, PLE and the spline panels use the same output dimensionality and differ only in how the coordinates of the encoded vector are formed.

# F    Model Architecture and Training Configuration

Table 7 summarizes the backbone hyperparameters and shared optimization settings used throughout the experiments.

## F.1    Hardware

All experiments were conducted on an Azure `Standard_NC48ads_A100_v4` virtual machine equipped with two NVIDIA A100 accelerators. Together, the two devices provided a total of $160\,\mathrm{GB}$ of GPU memory. Unless stated otherwise, all reported training and evaluation results were obtained on this hardware setup.

# G    Target-aware Knot Selection Configuration

**Adaptive vs. non-adaptive output size.** In *non-adaptive* mode, the per-feature output dimensionality is fixed in advance and shared across methods. We consider three output sizes, $m \in \{7, 15, 30\}$, corresponding to the number of basis functions for spline encodings and the number of bins for PLE. In *adaptive* mode, the output size is determined by the tree-guided procedure. This setting is used only for PLE in the ablation study, where the effective number of bins is selected from the range $[5, 50]$ subject to the regularization constraints reported in Table 8.

# H    Preprocessing Pipeline

For completeness, we provide the full algorithmic details of the spline preprocessing pipeline and the learnable-knot optimization procedure in Algorithm 1 and Algorithm 2, respectively. The preprocessing pipeline is shared by all spline variants listed in Table 3, whereas the second algorithm applies only to the learnable-knot variants.

| Model | Architecture | Configuration |
|---|---|---|
| MLP | 3-layer MLP | Hidden dims $[256, 128, 64]$; ReLU activations; dropout 0.3. |
| ResNet | Residual MLP blocks (Gorishniy et al. (2021)) | Block: Linear $\rightarrow$ BN $\rightarrow$ ReLU $\rightarrow$ Dropout $\rightarrow$ Linear $\rightarrow$ BN + skip; $d_{\mathrm{model}} = 256$; $n_{\mathrm{blocks}} = 3$; $d_{\mathrm{hidden\_factor}} = 2.0$; dropout 0.3; batch normalization. |
| FTT (FT-Transformer) | Feature Tokenizer + Transformer (Gorishniy et al. (2021)) | $d_{\mathrm{token}} = 192$; $n_{\mathrm{blocks}} = 3$; $n_{\mathrm{heads}} = 8$; attention dropout 0.2; FFN dropout 0.1; residual dropout 0.0; $ffn\_factor = 4/3$; ReGLU activations. |

**Shared training setup (all models)**

AdamW with backbone learning rate $\eta_\theta = 10^{-4}$ and weight decay $10^{-5}$, batch size 512, and a maximum of 200 epochs. Early stopping patience is set to 15, and ReduceLROnPlateau uses patience 10 with factor 0.1. For FT-Transformer, weight decay is excluded from feature token embeddings, layer normalization parameters, the [CLS] token, and bias terms.

**Additional setup for gradient-based knot optimization**

For learnable-knot spline variants, knot locations are optimized jointly with the backbone model. Knot updates are activated after a warm-up of $E_{\mathrm{warm}} = 50$ epochs. A separate learning rate is used for the knot parameters, with $\eta_a = 2\eta_\theta = 2 \times 10^{-4}$.

Table 7: Backbone architectures and training configuration. We report the hyperparameters for the MLP, ResNet, and FT-Transformer backbones, along with the shared optimization strategy. Additional settings specific to gradient-based knot optimization are listed separately.

| Method | Variant | Component | Non-adaptive (fixed) | Adaptive (range) |
|---|---|---|---|---|
| PLE | CART-based | Output size | $m = \{7, 15, 30\}$ | $min\_bins = 5$ $max\_bins = 50$ |
| | | Tree regularization | $min\_samples\_leaf = 1$ $min\_samples\_split = 2$ | $min\_samples\_leaf = 25$ $min\_samples\_split = 2$ |
| Splines | CART-based | Output size | $m = \{7, 15, 30\}$ | – |
| | | Tree / knot constraints | $max\_depth = 6$ $min\_knot\_spacing = 0.01$ | – |
| Splines | LightGBM-based | Output size | $m = \{7, 15, 30\}$ | – |
| | | GBDT hyperparameters | $n\_estimators = 100$ $max\_depth = 3$ $learning\_rate = 0.1$ | – |

Table 8: Target-aware configuration details and output-size settings. We report the configurations for target-aware PLE binning and target-aware spline knot placement using CART and LightGBM. Adaptive output size is used only for PLE in the ablation study; spline encodings use fixed output sizes $m \in \{7, 15, 30\}$.

---
**Algorithm 1:** Spline-Based Numerical Encoding Pipeline
---

**Input:** Numerical feature matrix $\mathbf{X}_{\text{num}} = (\mathbf{x}_1, \ldots, \mathbf{x}_d)$, with $\mathbf{x}_j = (x_{1,j}, \ldots, x_{n,j})^\top$; spline family $\mathcal{S} \in \{\text{B}, \text{M}, \text{I}\}$; knot strategy $\mathcal{K} \in \{\text{uniform}, \text{quantile-based}, \text{target-aware}, \text{learnable-knot}\}$; optional targets $\mathbf{y}$; number of internal knots $\{K_j\}_{j=1}^d$

**Output:** Expanded numerical encodings $\{\mathbf{\Phi}(\mathbf{x}_{\text{num}}^{(i)}; \boldsymbol{\tau})\}_{i=1}^n$

**Normalize** each numerical feature to $[0, 1]$ using training-split statistics

**for** $j \leftarrow 1$ **to** $d$ **do**

  **Knot placement** Construct an internal-knot vector $\boldsymbol{\kappa}_j = (\kappa_{j,1}, \ldots, \kappa_{j,K_j})$
  **if** $\mathcal{K}$ *is uniform* **then**

$$\kappa_{j,\ell} \leftarrow \frac{\ell}{K_j + 1}, \qquad \ell = 1, \ldots, K_j.$$

  **else if** $\mathcal{K}$ *is quantile-based* **then**

$$\kappa_{j,\ell} \leftarrow Q_j\left(\frac{\ell}{K_j + 1}\right), \qquad \ell = 1, \ldots, K_j,$$

  where $Q_j(\cdot)$ is the empirical quantile function of the normalized feature values $\mathbf{x}_j$
  **else if** $\mathcal{K}$ *is target-aware* **then**
    Fit a one-dimensional supervised splitter on $\{(x_{i,j}, y_i)\}_{i=1}^n$ and collect candidate split points
      Use either CART or LightGBM to obtain candidate thresholds on feature $j$
      Apply the spacing filter and retain up to $K_j$ thresholds ranked by split gain
      If fewer than $K_j$ valid thresholds remain, supplement with quantiles of $\mathbf{x}_j$
      Set $\boldsymbol{\kappa}_j$ to the sorted selected thresholds
    `// Applicable to all spline families` $\mathcal{S} \in \{\text{B}, \text{M}, \text{I}\}$`.`
  **else if** $\mathcal{K}$ *is learnable-knot* **then**
    `// Internal knots are optimized jointly with the downstream model.`
    Initialize $\boldsymbol{\kappa}_j$ from uniform placement
    Parameterize ordered internal knots via learnable spacings using the softmax-cumsum construction and update them by backpropagation during training, as in Algorithm 2
    `// Applicable to all spline families` $\mathcal{S} \in \{\text{B}, \text{M}, \text{I}\}$`.`

  **Full knot sequence** Construct $\boldsymbol{\tau}_j$ by augmenting $\boldsymbol{\kappa}_j$ with boundary knots using the standard boundary handling for spline family $\mathcal{S}$

  **Basis construction** Define basis functions $\{b_{j,\ell}^{(p)}(\cdot; \boldsymbol{\tau}_j)\}_{\ell=1}^{m_j}$ according to spline family $\mathcal{S}$, where $m_j = K_j + p + 1$
  **if** $\mathcal{S}$ *is B-splines* **then**
    Use the B-spline basis associated with $\boldsymbol{\tau}_j$ as defined in Appendix C.3
  **else if** $\mathcal{S}$ *is M-splines* **then**
    Use the corresponding nonnegative M-spline basis as defined in Appendix C.4
  **else if** $\mathcal{S}$ *is I-splines* **then**
    Use the integrated I-spline basis as defined in Appendix C.5

  **Basis evaluation** For each sample $i = 1, \ldots, n$, compute

$$\boldsymbol{\phi}_j(x_{i,j}; \boldsymbol{\tau}_j) \leftarrow \left(b_{j,1}^{(p)}(x_{i,j}; \boldsymbol{\tau}_j), \ldots, b_{j,m_j}^{(p)}(x_{i,j}; \boldsymbol{\tau}_j)\right).$$

**Concatenation** For each sample $i = 1, \ldots, n$, form

$$\mathbf{\Phi}(\mathbf{x}_{\text{num}}^{(i)}; \boldsymbol{\tau}) \leftarrow \left[\boldsymbol{\phi}_1(x_{i,1}; \boldsymbol{\tau}_1) \mid \cdots \mid \boldsymbol{\phi}_d(x_{i,d}; \boldsymbol{\tau}_d)\right].$$

---

---

**Algorithm 2:** Learnable-knot optimization for spline feature expansion

---

**Input:** Training data $\{(\mathbf{x}_{\text{num}}^{(i)}, \mathbf{x}_{\text{cat}}^{(i)}, y_i)\}_{i=1}^n$ with $d$ numerical features; numbers of internal knots $\{K_j\}_{j=1}^d$; minimum spacing $\delta > 0$; regularization weight $\lambda \geq 0$; stabilizer $\varepsilon > 0$; backbone $f_\theta$; learning rates $\eta_\theta, \eta_a$; warm-start epochs $E_{\text{warm}}$; total epochs $E$.

**Output:** Backbone parameters $\theta$ and knot parameters $\mathbf{a} = (\mathbf{a}_1, \ldots, \mathbf{a}_d)$.

**Normalize.** Map all numerical features to $[0, 1]$ using training-split statistics

**Initialize knot parameters. for** $j \leftarrow 1$ **to** $d$ **do**

    Choose an initial internal-knot vector

$$\boldsymbol{\kappa}_j^{(0)} = (\kappa_{j,1}^{(0)}, \ldots, \kappa_{j,K_j}^{(0)})$$

    using uniform placement

    Convert internal knots to interval widths:

$$w_{j,1}^{(0)} = \kappa_{j,1}^{(0)}, \quad w_{j,r}^{(0)} = \kappa_{j,r}^{(0)} - \kappa_{j,r-1}^{(0)} \quad (r = 2, \ldots, K_j), \quad w_{j,K_j+1}^{(0)} = 1 - \kappa_{j,K_j}^{(0)}.$$

    Invert the spacing map to initialize $\mathbf{a}_j \in \mathbb{R}^{K_j+1}$:

$$\pi_{j,r}^{(0)} = \frac{w_{j,r}^{(0)} - \delta}{1 - (K_j + 1)\delta} \quad (r = 1, \ldots, K_j + 1), \qquad a_{j,r} \leftarrow \log\Big(\max(\pi_{j,r}^{(0)}, 10^{-12})\Big).$$

Initialize backbone parameters $\theta$ using a standard initialization scheme

**for** $e \leftarrow 1$ **to** $E$ **do**

    **if** $e \leq E_{\text{warm}}$ **then**

        Freeze $\mathbf{a}$

    **else**

        Unfreeze $\mathbf{a}$

    **foreach** *minibatch* $\mathcal{B}$ **do**

        **Compute ordered internal knots. for** $j \leftarrow 1$ **to** $d$ **do**

            Compute the entries of the allocation-weight vector $\boldsymbol{\pi}_j$ and interval-width vector $\mathbf{w}_j$:

$$\pi_{j,r} \leftarrow \frac{\exp(a_{j,r})}{\sum_{s=1}^{K_j+1} \exp(a_{j,s})} \quad (r = 1, \ldots, K_j + 1),$$

$$w_{j,r} \leftarrow \delta + \Big(1 - (K_j + 1)\delta\Big)\pi_{j,r} \quad (r = 1, \ldots, K_j + 1).$$

        Recover the ordered internal knots by cumulative summation:

$$\kappa_{j,\ell} \leftarrow \sum_{r=1}^{\ell} w_{j,r} \quad (\ell = 1, \ldots, K_j).$$

        Construct the full knot sequence $\boldsymbol{\tau}_j$ from $\boldsymbol{\kappa}_j$ using boundary handling for the chosen spline family

        **Spline feature expansion.** For each sample $i \in \mathcal{B}$, compute $\boldsymbol{\Phi}(\mathbf{x}_{\text{num}}^{(i)}; \boldsymbol{\tau}(\mathbf{a}))$ by evaluating the chosen spline family using the full knot sequences $\{\boldsymbol{\tau}_j\}_{j=1}^d$

        **Forward and task loss.**

$$L_{\text{task}} \leftarrow \frac{1}{|\mathcal{B}|} \sum_{i \in \mathcal{B}} \mathcal{L}\Big(f_\theta\Big(\boldsymbol{\Phi}\big(\mathbf{x}_{\text{num}}^{(i)}; \boldsymbol{\tau}(\mathbf{a})\big), \mathbf{x}_{\text{cat}}^{(i)}\Big), y_i\Big).$$

        **Spacing regularization.**

$$\mathcal{R}_{\text{space}}(\mathbf{a}) \leftarrow \frac{1}{d} \sum_{j=1}^d \frac{1}{K_j + 1} \sum_{r=1}^{K_j+1} \frac{1}{w_{j,r} + \varepsilon}.$$

        **Total loss and update.**

$$L \leftarrow L_{\text{task}} + \lambda \, \mathcal{R}_{\text{space}}(\mathbf{a}).$$

        Take one optimizer step using $\nabla_\theta L$ and, if unfrozen, $\nabla_{\mathbf{a}} L$, for example

$$\theta \leftarrow \theta - \eta_\theta \nabla_\theta L, \qquad \mathbf{a} \leftarrow \mathbf{a} - \eta_a \nabla_{\mathbf{a}} L.$$

        The knot learning rate $\eta_a$ is chosen separately from $\eta_\theta$. In our experiments, we use $\eta_a = 2\eta_\theta$

---

# I Critical Difference (CD) Diagrams

We summarize comparisons of multiple preprocessing methods using *critical difference (CD) diagrams*, following the rank-based evaluation protocol for multi-dataset studies (Demšar, 2006). For each evaluation block $i$, here one *dataset × backbone* pair, we rank the $k$ preprocessing methods by performance, where rank 1 is best and ties receive the average rank. Let $r_{i,j}$ denote the rank of method $j \in \{1, \ldots, k\}$ on block $i \in \{1, \ldots, N\}$. The diagram reports the *average rank*

$$\bar{r}_j = \frac{1}{N} \sum_{i=1}^{N} r_{i,j},$$

where lower $\bar{r}_j$ indicates better overall performance.

To test whether rank differences are attributable to chance, we first apply the Friedman test for repeated-measures comparisons (Friedman, 1937; Iman & Davenport, 1980). When the global null is rejected, we use the Nemenyi post-hoc procedure to account for multiple pairwise comparisons (Nemenyi, 1963; Demšar, 2006). The corresponding *critical difference* at significance level $\alpha$ is

$$\text{CD} = q_\alpha \sqrt{\frac{k(k+1)}{6N}},$$

where $q_\alpha$ is the critical value of the Studentized range used by the Nemenyi test. Two methods are considered significantly different if $|\bar{r}_a - \bar{r}_b| > \text{CD}$. CD diagrams are widely used in modern ML and DL benchmarking to summarize average ranks and statistically indistinguishable groups across many datasets (Feuer et al., 2024; Kadra et al., 2024).

In our setting, we compare $k = 14$ preprocessing methods across three backbones, MLP, ResNet, and FT-Transformer, and report CD diagrams separately for each output size $m \in \{7, 15, 30\}$. The number of blocks is task-dependent. We have $N_{\text{reg}} = 13 \times 3 = 39$ for regression, $N_{\text{cls}} = 12 \times 3 = 36$ for classification, and $N_{\text{all}} = (13 + 12) \times 3 = 75$ when combining both tasks. In the reported CD diagrams, regression and classification are analyzed separately. For regression, methods are ranked by NRMSE, where lower values are better. For classification, methods are ranked by AUC, where higher values are better.

## I.1 Average-rank table for regression

Table 9 reports the average ranks used to generate the regression CD diagrams in Figure 2. The ranks are computed across dataset–backbone blocks, with lower values indicating better average performance. The table is provided as a numerical companion to the CD diagrams. The CD diagrams additionally show the post-hoc significance groups, while the table makes the average-rank values easier to inspect.

## I.2 Average-rank table for classification

Table 10 reports the average ranks used to generate the classification CD diagrams in Figure 3. The ranks are computed across dataset–backbone blocks, with lower values indicating better average performance. As in the regression setting, the table complements the CD diagrams by providing the numerical average ranks, while the diagrams show the statistically indistinguishable groups obtained from the post-hoc test.

Table 9: Average rank (lower is better) of each preprocessing method across all backbones on the regression benchmark, for basis resolutions of 7, 15 and 30 bins/basis functions. The best (lowest) rank in each column is in bold.

| Preprocessing Method | Avg. rank, $m = 7$ | Avg. rank, $m = 15$ | Avg. rank, $m = 30$ |
|---|---|---|---|
| Std | 9.29 | 8.96 | 8.46 |
| MinMax | 11.36 | 10.67 | 10.05 |
| PLE | 8.50 | 7.36 | 5.91 |
| BS-U | 7.29 | 7.82 | 7.97 |
| BS-Q | 5.65 | 7.08 | 8.42 |
| BS-CART | 5.91 | 7.46 | 9.73 |
| BS-LGBM | **5.59** | 7.60 | 8.47 |
| BS-Grad-U | 6.10 | 6.53 | 6.47 |
| IS-U | 8.08 | 7.67 | 6.94 |
| IS-Q | 7.90 | **6.01** | **5.64** |
| IS-CART | 7.29 | 6.67 | 7.12 |
| IS-LGBM | 7.24 | 6.10 | 5.78 |
| IS-Grad-U | 6.83 | 6.79 | 5.76 |
| MS-Grad-U | 7.95 | 8.28 | 8.27 |

Table 10: Average rank (lower is better) of each preprocessing method across all backbones on the classification benchmark, for basis resolutions of 7, 15 and 30 bins/basis functions. The best (lowest) rank in each column is in bold.

| Preprocessing Method | Avg. rank, $m = 7$ | Avg. rank, $m = 15$ | Avg. rank, $m = 30$ |
|---|---|---|---|
| Std | 11.33 | 11.21 | 10.21 |
| MinMax | 12.53 | 12.10 | 11.25 |
| PLE | **5.01** | **3.96** | **3.33** |
| BS-U | 8.15 | 7.64 | 8.56 |
| BS-Q | 6.12 | 7.78 | 8.64 |
| BS-CART | 5.25 | 5.78 | 8.00 |
| BS-LGBM | 5.67 | 7.51 | 8.60 |
| BS-Grad-U | 5.90 | 7.42 | 7.11 |
| IS-U | 8.10 | 7.33 | 7.31 |
| IS-Q | 7.43 | 6.78 | 5.19 |
| IS-CART | 6.71 | 5.26 | 5.56 |
| IS-LGBM | 7.03 | 6.53 | 5.53 |
| IS-Grad-U | 6.99 | 6.64 | 6.29 |
| MS-Grad-U | 8.78 | 9.07 | 9.43 |

## J  Experimental Results

We report detailed per-dataset results for both regression and classification in this appendix. For each task, we evaluate three fixed per-feature output sizes, corresponding to $m \in \{7, 15, 30\}$. For spline-based encodings (B-, I-, and M-splines), these values determine the number of basis functions. For PLE, the same values correspond to the number of bins. The baseline preprocessing methods, Std and MinMax, do not depend on output size and are therefore identical across all three settings. Within each backbone and dataset, the best-performing method is highlighted in bold.

### J.1 Regression Results

Regression tables report mean NRMSE ($\downarrow$) $\pm$ standard deviation over 5-fold cross-validation. Results are provided for $m = 7$, $m = 15$, and $m = 30$ in Tables 11, 12, and 13, respectively.

### J.2 Classification Results

Classification tables report mean AUC ($\uparrow$) $\pm$ standard deviation over 5-fold cross-validation. For binary datasets, this corresponds to standard ROC-AUC. For multiclass datasets, we report weighted one-vs-rest ROC-AUC. Results are provided for $m = 7$, $m = 15$, and $m = 30$ in Tables 14, 15, and 16, respectively.

### J.3 Synthetic Setup for the Illustrative Comparison of PLE and B-spline Encodings

This appendix provides the setup used for the illustrative comparison in Section 4.4. The experiment is intended only to visualize the different inductive biases of PLE and cubic B-spline encodings under the same basis budget. We generate two one-dimensional datasets on the interval $x \in [0, 1]$ using a fixed random seed. For regression, we sample $n = 2500$ inputs uniformly from $[0, 1]$ and define the target as

$$y = \sin(3\pi x) + 0.5\cos(7\pi x)e^{-2x} + 0.3x^2 + \varepsilon,$$

where $\varepsilon \sim \mathcal{N}(0, 0.04)$. This produces a smooth nonlinear target with varying local curvature.

For classification, we again sample $n = 2500$ inputs uniformly from $[0, 1]$ and draw labels from a Bernoulli distribution with class probability

$$p(x) = \text{clip}(\sigma(25(x - 0.33)) - \sigma(25(x - 0.72)) + 0.04,\ 0.04,\ 0.96),$$

where $\sigma(\cdot)$ denotes the logistic sigmoid. This creates a two-boundary band structure with sharp but noisy transitions.

**Encodings.** Both datasets are encoded with the same budget of $m = 10$ dimensions. For PLE, we use uniform bin boundaries on $[0, 1]$. For the spline representation, we use a clamped uniform cubic B-spline basis on $[0, 1]$. This keeps the basis budget fixed across the two encodings.

**Predictive models.** For the regression task, we fit a Ridge model on top of each encoding. For the classification task, we fit a logistic regression model on top of each encoding. The aim is to keep the downstream predictor simple so that the comparison mainly reflects the structure induced by the encoding.

**Evaluation.** For regression, we plot the fitted curve together with the noiseless target function and report NRMSE, computed against the noiseless target on a dense evaluation grid over $[0, 1]$. For classification, we plot the fitted class-probability curve together with the true probability function and report both AUC and Brier score. These figures are intended as qualitative illustrations rather than as a benchmark.

## K  Ablation Study Setup

This appendix gives the setup for the ablation study in Section 5. We use a synthetic regression dataset to isolate the effect of numerical encoding resolution under a controlled feature and target relationship. The dataset contains one informative feature and one nuisance feature. The informative feature $x_0 \in [0, 1]$ is sampled from the mixture distribution

$$x_0 \sim \begin{cases} \text{Beta}(2, 8), & \text{with probability } 0.70, \\ \text{Beta}(8, 2), & \text{with probability } 0.20, \\ \text{Uniform}(0, 1), & \text{with probability } 0.10, \end{cases}$$

which gives a non-uniform marginal distribution over the input domain. The nuisance feature is defined as

$$x_1 = 0.6\,x_0 + 0.4\,U(0, 1),$$

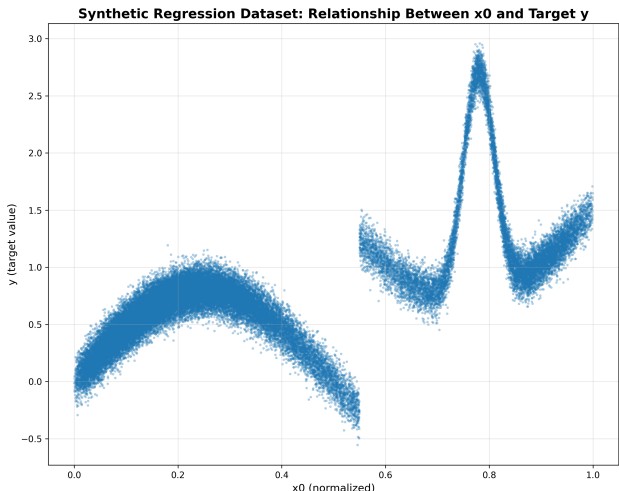

Figure 9: Synthetic regression dataset used in the ablation study. Scatter plot of the informative feature $x_0$ against the target variable $y$, illustrating the heterogeneous structure induced by the synthetic target function.

where $U(0,1)$ denotes a uniform random variable on $[0,1]$.

The target depends only on $x_0$ through

$$f(x_0) = 0.8 \sin(2\pi x_0) + 1.5\, \mathbf{1}[x_0 > 0.55] + 2.0 \exp\left(-\frac{(x_0 - 0.78)^2}{2(0.03)^2}\right),$$

and the final response is

$$y = f(x_0) + \varepsilon, \qquad \varepsilon \sim \mathcal{N}(0, 0.10^2).$$

This construction combines smooth nonlinear variation, a threshold effect, and a narrow localized peak. Figure 9 shows the relationship between $x_0$ and $y$. We study sensitivity to the number of bins or basis functions on this synthetic regression task by varying the encoding resolution $m \in \{5, 10, 15, 20, 25, 30, 35, 40, 45, 50\}$, while keeping all other hyperparameters fixed. This isolates the effect of encoding capacity from other architectural choices. Test NRMSE ($\downarrow$), averaged over 5 random seeds and reported with standard deviation, is given in Table 17.

### K.1 Knot Relocation During Training

To complement the ablation on encoding resolution, we include a small qualitative experiment on the same synthetic regression task to visualize how learnable knot locations evolve during training. Using the same MLP setup as in the main experiments, we fix the numerical encoding size to $m = 10$ basis functions per feature and optimize knot parameters jointly with the backbone. This gives six learnable internal knots per feature according to Appendix C.2. Unlike the encoding-resolution ablation, which reports results averaged over 5 random seeds, this experiment is shown for a single seed only and is intended as an illustration of knot movement during training rather than as a performance comparison.

Figure 10 shows the BS-Grad-U knot trajectories for the informative feature $x_0$ under uniform initialization and different knot learning rates. The learned knot movement depends on several factors, including the input distribution, the target structure, and the gradients induced during optimization. At the smallest learning rates, the knots move only slightly, whereas larger learning rates lead to more visible relocation during training. Across all settings shown, however, the trajectories remain well behaved and ordered, which is consistent with stable optimization under our parameterization. This figure should therefore be read as a qualitative illustration that knot relocation can remain stable during training, rather than as a complete analysis of the factors governing knot dynamics.

| Backbone | Method | AB | CA | CPU | DI | HS | PA | WI | FW | H8 | PR | PU | SG | SU |
|---|---|---|---|---|---|---|---|---|---|---|---|---|---|---|
| MLP | STD | 0.4610 ± 0.020 | 0.2751 ± 0.021 | 0.0456 ± 0.007 | 0.0581 ± 0.006 | 0.1638 ± 0.008 | 0.6641 ± 0.018 | 0.7010 ± 0.015 | 0.3536 ± 0.009 | 0.3754 ± 0.019 | 0.5976 ± 0.016 | 0.2126 ± 0.015 | 0.0583 ± 0.004 | 0.2740 ± 0.082 |
| | MinMax | 0.4837 ± 0.013 | 0.3223 ± 0.016 | 0.0569 ± 0.006 | 0.0644 ± 0.003 | 0.2177 ± 0.012 | 0.7792 ± 0.029 | 0.7105 ± 0.012 | 0.4085 ± 0.023 | 0.4292 ± 0.011 | 0.6411 ± 0.017 | 0.2254 ± 0.015 | 0.0826 ± 0.009 | 0.3525 ± 0.070 |
| | PLE | 0.4676 ± 0.017 | 0.2650 ± 0.020 | 0.0297 ± 0.005 | 0.0601 ± 0.004 | 0.1470 ± 0.014 | 0.5098 ± 0.024 | 0.6685 ± 0.018 | 0.3413 ± 0.013 | 0.3624 ± 0.029 | 0.5599 ± 0.018 | 0.2100 ± 0.018 | 0.0373 ± 0.006 | 0.2611 ± 0.076 |
| | BS-U | 0.4431 ± 0.015 | 0.2595 ± 0.019 | 0.0287 ± 0.004 | 0.0612 ± 0.004 | 0.1396 ± 0.013 | 0.5489 ± 0.015 | 0.6606 ± 0.011 | 0.3455 ± 0.009 | 0.3611 ± 0.023 | 0.5645 ± 0.013 | 0.2047 ± 0.018 | 0.0637 ± 0.018 | 0.2624 ± 0.077 |
| | BS-Q | 0.4438 ± 0.015 | 0.2569 ± 0.019 | 0.0290 ± 0.004 | 0.0627 ± 0.003 | 0.1342 ± 0.013 | 0.5415 ± 0.015 | 0.6548 ± 0.014 | 0.3422 ± 0.008 | 0.3536 ± 0.021 | 0.5407 ± 0.021 | 0.2090 ± 0.018 | 0.0655 ± 0.018 | 0.2604 ± 0.087 |
| | BS-CART | 0.4456 ± 0.019 | 0.2602 ± 0.018 | 0.0285 ± 0.004 | 0.0610 ± 0.003 | 0.1364 ± 0.015 | 0.5228 ± 0.018 | 0.6570 ± 0.015 | 0.3404 ± 0.009 | 0.3515 ± 0.026 | 0.5521 ± 0.017 | 0.2022 ± 0.018 | 0.0591 ± 0.016 | 0.2637 ± 0.100 |
| | BS-LGBM | 0.4457 ± 0.014 | 0.2558 ± 0.018 | 0.0283 ± 0.004 | 0.0610 ± 0.004 | 0.1347 ± 0.016 | 0.5270 ± 0.011 | 0.6575 ± 0.013 | 0.3392 ± 0.009 | 0.3510 ± 0.024 | 0.5566 ± 0.013 | 0.2045 ± 0.019 | 0.0644 ± 0.018 | 0.2592 ± 0.096 |
| | BS-Grad-U | **0.4372** ± 0.019 | **0.2190** ± 0.017 | **0.0273** ± 0.004 | **0.0244** ± 0.001 | **0.1033** ± 0.010 | **0.3187** ± 0.013 | **0.6075** ± 0.014 | **0.3888** ± 0.009 | **0.3182** ± 0.030 | 0.4306 ± 0.013 | **0.1939** ± 0.022 | **0.0112** ± 0.002 | **0.2079** ± 0.076 |
| | IS-U | 0.4449 ± 0.022 | 0.2618 ± 0.020 | 0.0296 ± 0.005 | 0.0614 ± 0.004 | 0.1498 ± 0.020 | 0.5726 ± 0.012 | 0.6671 ± 0.017 | 0.3464 ± 0.011 | 0.3616 ± 0.021 | 0.5791 ± 0.012 | 0.2105 ± 0.017 | 0.0511 ± 0.010 | 0.2610 ± 0.084 |
| | IS-Q | 0.4477 ± 0.022 | 0.2603 ± 0.018 | 0.0320 ± 0.004 | 0.0623 ± 0.004 | 0.1416 ± 0.017 | 0.5756 ± 0.015 | 0.6640 ± 0.015 | 0.3438 ± 0.007 | 0.3572 ± 0.023 | 0.5722 ± 0.011 | 0.2118 ± 0.017 | 0.0579 ± 0.016 | 0.2614 ± 0.086 |
| | IS-CART | 0.4434 ± 0.016 | 0.2635 ± 0.016 | 0.0299 ± 0.004 | 0.0612 ± 0.004 | 0.1479 ± 0.021 | 0.5687 ± 0.012 | 0.6658 ± 0.015 | 0.3424 ± 0.009 | 0.3534 ± 0.023 | 0.5741 ± 0.012 | 0.2078 ± 0.018 | 0.0564 ± 0.015 | 0.2590 ± 0.083 |
| | IS-LGBM | 0.4474 ± 0.022 | 0.2608 ± 0.021 | 0.0288 ± 0.005 | 0.0612 ± 0.004 | 0.1479 ± 0.021 | 0.5681 ± 0.010 | 0.6667 ± 0.018 | 0.3416 ± 0.009 | 0.3546 ± 0.023 | 0.5766 ± 0.011 | 0.2079 ± 0.018 | 0.0566 ± 0.014 | 0.2627 ± 0.086 |
| | IS-Grad-U | 0.4387 ± 0.018 | 0.2352 ± 0.019 | 0.0288 ± 0.004 | 0.0253 ± 0.001 | 0.1087 ± 0.013 | 0.3639 ± 0.010 | 0.6318 ± 0.013 | 0.3429 ± 0.009 | 0.3196 ± 0.027 | 0.4743 ± 0.014 | 0.1965 ± 0.021 | 0.0123 ± 0.002 | 0.2235 ± 0.076 |
| | MS-Grad-U | 0.4405 ± 0.022 | 0.2228 ± 0.013 | 0.0304 ± 0.004 | 0.0252 ± 0.002 | 0.1243 ± 0.018 | 0.4048 ± 0.010 | 0.6110 ± 0.018 | 0.3438 ± 0.009 | 0.3352 ± 0.029 | **0.4126** ± 0.011 | 0.1993 ± 0.022 | 0.0257 ± 0.004 | 0.2324 ± 0.068 |
| RESNET | STD | 0.4277 ± 0.027 | 0.2545 ± 0.017 | 0.0329 ± 0.007 | 0.0339 ± 0.012 | 0.1336 ± 0.015 | 0.4344 ± 0.023 | 0.6484 ± 0.011 | 0.3605 ± 0.011 | 0.3450 ± 0.029 | 0.5103 ± 0.013 | 0.1983 ± 0.019 | 0.0296 ± 0.004 | 0.2449 ± 0.097 |
| | MinMax | 0.4291 ± 0.025 | 0.2805 ± 0.025 | 0.0326 ± 0.005 | 0.0362 ± 0.007 | 0.1321 ± 0.014 | 0.4345 ± 0.018 | 0.6486 ± 0.013 | 0.3644 ± 0.013 | 0.3464 ± 0.030 | 0.5301 ± 0.022 | 0.1988 ± 0.020 | 0.0383 ± 0.009 | 0.2313 ± 0.077 |
| | PLE | 0.4548 ± 0.018 | 0.2385 ± 0.013 | 0.0266 ± 0.004 | 0.0271 ± 0.004 | 0.1118 ± 0.009 | 0.3018 ± 0.034 | 0.6180 ± 0.019 | 0.3388 ± 0.015 | 0.3172 ± 0.032 | 0.4075 ± 0.012 | 0.1992 ± 0.022 | 0.0190 ± 0.004 | 0.2318 ± 0.081 |
| | BS-U | 0.4292 ± 0.047 | 0.2248 ± 0.016 | 0.0262 ± 0.004 | 0.0301 ± 0.006 | 0.1091 ± 0.014 | 0.2500 ± 0.028 | 0.6149 ± 0.013 | 0.3460 ± 0.012 | 0.3185 ± 0.022 | 0.4198 ± 0.008 | 0.1974 ± 0.019 | 0.0181 ± 0.005 | **0.1785** ± 0.098 |
| | BS-Q | 0.4312 ± 0.016 | **0.2129** ± 0.013 | **0.0261** ± 0.003 | 0.0270 ± 0.003 | **0.1072** ± 0.016 | 0.2332 ± 0.015 | 0.6160 ± 0.016 | 0.3326 ± 0.006 | 0.3105 ± 0.031 | 0.4036 ± 0.019 | 0.1975 ± 0.018 | 0.0191 ± 0.004 | 0.1856 ± 0.113 |
| | BS-CART | 0.4327 ± 0.047 | 0.2240 ± 0.012 | 0.0267 ± 0.004 | 0.0280 ± 0.002 | 0.1078 ± 0.013 | 0.2236 ± 0.015 | 0.6183 ± 0.010 | 0.3330 ± 0.007 | 0.3114 ± 0.023 | 0.4097 ± 0.011 | **0.1955** ± 0.020 | 0.0194 ± 0.003 | 0.1874 ± 0.088 |
| | BS-LGBM | 0.4323 ± 0.015 | 0.2322 ± 0.019 | **0.0261** ± 0.003 | 0.0273 ± 0.003 | 0.1091 ± 0.013 | 0.2369 ± 0.014 | 0.6199 ± 0.004 | **0.3297** ± 0.009 | **0.3054** ± 0.026 | 0.4313 ± 0.019 | 0.1990 ± 0.018 | 0.0166 ± 0.002 | 0.1985 ± 0.109 |
| | BS-Grad-U | 0.4401 ± 0.021 | 0.2281 ± 0.036 | 0.0372 ± 0.006 | 0.0229 ± 0.001 | 0.1362 ± 0.022 | **0.1569** ± 0.008 | **0.6114** ± 0.013 | 0.3491 ± 0.008 | 0.3340 ± 0.038 | **0.3362** ± 0.010 | 0.2065 ± 0.020 | 0.0176 ± 0.006 | 0.2378 ± 0.092 |
| | IS-U | **0.4257** ± 0.021 | 0.2297 ± 0.016 | **0.0261** ± 0.004 | 0.0280 ± 0.002 | 0.1155 ± 0.017 | 0.2842 ± 0.019 | 0.6250 ± 0.015 | 0.3447 ± 0.010 | 0.3138 ± 0.028 | 0.4472 ± 0.021 | 0.1965 ± 0.019 | 0.0245 ± 0.004 | 0.1914 ± 0.101 |
| | IS-Q | 0.4294 ± 0.014 | 0.2379 ± 0.017 | 0.0264 ± 0.004 | 0.0271 ± 0.002 | 0.1120 ± 0.018 | 0.2724 ± 0.035 | 0.6257 ± 0.014 | 0.3363 ± 0.010 | 0.3112 ± 0.033 | 0.4436 ± 0.016 | 0.1983 ± 0.016 | 0.0199 ± 0.002 | 0.1926 ± 0.091 |
| | IS-CART | 0.4288 ± 0.017 | 0.2314 ± 0.011 | 0.0262 ± 0.004 | 0.0264 ± 0.002 | 0.1111 ± 0.016 | 0.2690 ± 0.031 | 0.6179 ± 0.023 | 0.3353 ± 0.011 | 0.3075 ± 0.028 | 0.4416 ± 0.017 | 0.1957 ± 0.017 | 0.0223 ± 0.004 | 0.2118 ± 0.084 |
| | IS-LGBM | 0.4293 ± 0.019 | 0.2345 ± 0.017 | **0.0261** ± 0.003 | 0.0279 ± 0.003 | 0.1086 ± 0.013 | 0.2660 ± 0.027 | 0.6160 ± 0.011 | 0.3343 ± 0.010 | 0.3126 ± 0.027 | 0.4520 ± 0.023 | 0.1971 ± 0.018 | 0.0211 ± 0.005 | 0.2051 ± 0.091 |
| | IS-Grad-U | 0.4298 ± 0.017 | 0.2270 ± 0.022 | 0.0312 ± 0.006 | 0.0235 ± 0.002 | 0.1304 ± 0.020 | 0.1993 ± 0.019 | 0.6121 ± 0.007 | 0.3762 ± 0.029 | 0.3335 ± 0.036 | 0.3654 ± 0.023 | 0.2046 ± 0.026 | **0.0162** ± 0.006 | 0.2203 ± 0.092 |
| | MS-Grad-U | 0.4464 ± 0.023 | 0.2215 ± 0.017 | 0.0338 ± 0.007 | **0.0220** ± 0.001 | 0.1307 ± 0.017 | 0.2999 ± 0.027 | 0.6222 ± 0.006 | 0.3565 ± 0.013 | 0.3427 ± 0.041 | 0.3427 ± 0.009 | 0.2081 ± 0.019 | 0.0279 ± 0.015 | 0.2375 ± 0.079 |
| FTT | STD | 0.4609 ± 0.028 | **0.2290** ± 0.010 | 0.0254 ± 0.003 | 0.0201 ± 0.002 | **0.1219** ± 0.013 | 0.1318 ± 0.024 | 0.6727 ± 0.043 | 0.3396 ± 0.013 | **0.3248** ± 0.030 | 0.3858 ± 0.016 | 0.2017 ± 0.029 | **0.0097** ± 0.002 | **0.2805** ± 0.065 |
| | MinMax | 0.4565 ± 0.040 | 0.2496 ± 0.018 | 0.0310 ± 0.005 | 0.0204 ± 0.001 | 0.1276 ± 0.019 | 0.3347 ± 0.088 | 0.6724 ± 0.038 | 0.3485 ± 0.010 | 0.3531 ± 0.026 | 0.4333 ± 0.031 | 0.2066 ± 0.022 | 0.0101 ± 0.006 | **0.2373** ± 0.078 |
| | PLE | 0.4871 ± 0.036 | 0.2395 ± 0.019 | 0.0330 ± 0.008 | 0.0198 ± 0.001 | 0.1409 ± 0.015 | 0.3246 ± 0.047 | **0.6520** ± 0.021 | 0.3409 ± 0.015 | 0.3547 ± 0.033 | 0.4144 ± 0.015 | 0.2117 ± 0.035 | 0.0160 ± 0.007 | 0.2580 ± 0.085 |
| | BS-U | 0.4683 ± 0.031 | 0.2477 ± 0.039 | 0.0381 ± 0.010 | 0.0208 ± 0.003 | 0.1324 ± 0.023 | 0.1893 ± 0.097 | 0.6582 ± 0.040 | 0.3414 ± 0.009 | 0.3553 ± 0.030 | 0.4045 ± 0.028 | 0.2050 ± 0.035 | 0.0252 ± 0.004 | 0.2438 ± 0.092 |
| | BS-Q | 0.4633 ± 0.031 | 0.2512 ± 0.039 | 0.0269 ± 0.005 | 0.0197 ± 0.001 | 0.1339 ± 0.022 | **0.0644** ± 0.059 | 0.6625 ± 0.039 | 0.3424 ± 0.014 | 0.3347 ± 0.039 | 0.4020 ± 0.019 | 0.2033 ± 0.023 | 0.0338 ± 0.006 | 0.2483 ± 0.080 |
| | BS-CART | 0.4717 ± 0.046 | 0.2428 ± 0.022 | 0.0301 ± 0.009 | 0.0197 ± 0.002 | 0.1368 ± 0.019 | 0.1357 ± 0.155 | 0.6666 ± 0.021 | 0.3398 ± 0.009 | 0.3334 ± 0.023 | 0.3863 ± 0.013 | 0.2065 ± 0.027 | 0.0395 ± 0.012 | 0.2407 ± 0.081 |
| | BS-LGBM | 0.4767 ± 0.040 | 0.2436 ± 0.017 | 0.0285 ± 0.004 | **0.0196** ± 0.001 | 0.1383 ± 0.019 | 0.0829 ± 0.076 | 0.6582 ± 0.030 | **0.3312** ± 0.009 | 0.3343 ± 0.033 | **0.3843** ± 0.020 | 0.2046 ± 0.025 | 0.0339 ± 0.012 | 0.2492 ± 0.097 |
| | BS-Grad-U | 0.4772 ± 0.047 | 0.2416 ± 0.036 | 0.0386 ± 0.003 | 0.0219 ± 0.001 | 0.1502 ± 0.025 | 0.1813 ± 0.177 | 0.6717 ± 0.033 | 0.3412 ± 0.012 | 0.3563 ± 0.052 | 0.3985 ± 0.019 | 0.2017 ± 0.029 | 0.0516 ± 0.035 | 0.2479 ± 0.086 |
| | IS-U | 0.4686 ± 0.021 | 0.2431 ± 0.022 | 0.0469 ± 0.011 | 0.0205 ± 0.003 | 0.1457 ± 0.036 | 0.0723 ± 0.040 | 0.6546 ± 0.021 | 0.3371 ± 0.008 | 0.3439 ± 0.027 | 0.3915 ± 0.018 | 0.2081 ± 0.019 | 0.0126 ± 0.004 | 0.2450 ± 0.072 |
| | IS-Q | 0.4669 ± 0.019 | 0.2315 ± 0.016 | 0.0351 ± 0.007 | 0.0201 ± 0.002 | 0.1460 ± 0.028 | 0.0942 ± 0.097 | 0.6656 ± 0.024 | 0.3344 ± 0.011 | 0.3442 ± 0.030 | 0.4017 ± 0.031 | 0.1989 ± 0.023 | 0.0177 ± 0.005 | 0.2472 ± 0.080 |
| | IS-CART | 0.4671 ± 0.024 | 0.2443 ± 0.024 | 0.0430 ± 0.017 | 0.0202 ± 0.001 | 0.1478 ± 0.013 | 0.2028 ± 0.235 | 0.6679 ± 0.031 | 0.3324 ± 0.007 | 0.3362 ± 0.026 | 0.3904 ± 0.013 | 0.2048 ± 0.033 | 0.0139 ± 0.005 | 0.2539 ± 0.076 |
| | IS-LGBM | **0.4551** ± 0.017 | 0.2551 ± 0.027 | 0.0335 ± 0.009 | 0.0202 ± 0.002 | 0.1437 ± 0.023 | 0.1607 ± 0.171 | 0.6627 ± 0.007 | 0.3322 ± 0.008 | 0.3320 ± 0.025 | 0.3879 ± 0.008 | 0.2039 ± 0.029 | 0.0171 ± 0.013 | 0.2568 ± 0.064 |
| | IS-Grad-U | 0.4988 ± 0.036 | 0.2749 ± 0.030 | 0.0462 ± 0.010 | 0.0222 ± 0.003 | 0.1652 ± 0.027 | 0.1585 ± 0.084 | 0.6799 ± 0.032 | 0.3451 ± 0.012 | 0.3672 ± 0.047 | 0.4109 ± 0.014 | **0.1976** ± 0.021 | 0.0192 ± 0.007 | 0.2414 ± 0.088 |
| | MS-Grad-U | 0.5020 ± 0.027 | 0.2538 ± 0.026 | 0.0286 ± 0.004 | 0.0219 ± 0.002 | 0.1374 ± 0.008 | 0.3694 ± 0.176 | 0.6739 ± 0.041 | 0.3526 ± 0.018 | 0.3935 ± 0.025 | 0.4171 ± 0.023 | 0.2025 ± 0.020 | 0.2494 ± 0.135 | 0.2384 ± 0.080 |

Table 11: Regression results for $m = 7$. Mean NRMSE ($\downarrow$) ± standard deviation over 5-fold cross-validation. Dataset and preprocessing abbreviations are given in Tables 4 and 3 respectively. Bold indicates the lowest NRMSE for each dataset within each backbone.

Table 12 (rotated). Columns: Backbone, Method, AB, CA, CPU, DI, HS, PA, WI, FW, H8, PR, PU, SG, SU

| Backbone | Method | AB | CA | CPU | DI | HS | PA | WI | FW | H8 | PR | PU | SG | SU |
|---|---|---|---|---|---|---|---|---|---|---|---|---|---|---|
| MLP | STD | 0.4610 ± 0.020 | 0.2751 ± 0.021 | 0.0456 ± 0.007 | 0.0581 ± 0.006 | 0.1638 ± 0.008 | 0.6641 ± 0.018 | 0.7010 ± 0.015 | 0.3536 ± 0.009 | 0.3754 ± 0.019 | 0.5976 ± 0.016 | 0.2126 ± 0.015 | 0.0583 ± 0.004 | 0.2740 ± 0.082 |
| | MinMax | 0.4837 ± 0.013 | 0.3223 ± 0.016 | 0.0569 ± 0.006 | 0.0644 ± 0.003 | 0.2177 ± 0.012 | 0.7792 ± 0.029 | 0.7105 ± 0.012 | 0.4085 ± 0.023 | 0.4292 ± 0.025 | 0.6411 ± 0.017 | 0.2254 ± 0.015 | 0.0826 ± 0.009 | 0.3525 ± 0.070 |
| | PLE | 0.4629 ± 0.011 | 0.2350 ± 0.012 | 0.0286 ± 0.004 | 0.0598 ± 0.004 | 0.1412 ± 0.018 | 0.4120 ± 0.014 | 0.6584 ± 0.017 | 0.3307 ± 0.008 | 0.3556 ± 0.025 | 0.5285 ± 0.013 | 0.2092 ± 0.017 | 0.0305 ± 0.007 | 0.2629 ± 0.082 |
| | BS-U | 0.4509 ± 0.014 | 0.2354 ± 0.013 | 0.0272 ± 0.004 | 0.0594 ± 0.005 | 0.1297 ± 0.011 | 0.4039 ± 0.014 | 0.6492 ± 0.013 | 0.3458 ± 0.008 | 0.3539 ± 0.023 | 0.5136 ± 0.013 | 0.2055 ± 0.019 | 0.0624 ± 0.019 | 0.2522 ± 0.094 |
| | BS-Q | 0.4588 ± 0.019 | 0.2273 ± 0.018 | 0.0268 ± 0.004 | 0.0621 ± 0.004 | 0.1289 ± 0.015 | 0.3585 ± 0.020 | 0.6627 ± 0.016 | **0.3304** ± 0.007 | 0.3445 ± 0.027 | 0.4847 ± 0.011 | 0.2039 ± 0.018 | 0.0819 ± 0.016 | 0.2522 ± 0.089 |
| | BS-CART | 0.4566 ± 0.014 | 0.2360 ± 0.017 | 0.0273 ± 0.004 | 0.0604 ± 0.005 | 0.1310 ± 0.014 | 0.3738 ± 0.008 | 0.6493 ± 0.013 | 0.3312 ± 0.008 | 0.3493 ± 0.025 | 0.4982 ± 0.013 | 0.2079 ± 0.020 | 0.0835 ± 0.016 | 0.2545 ± 0.097 |
| | BS-LGBM | 0.4570 ± 0.020 | 0.2255 ± 0.016 | 0.0275 ± 0.004 | 0.0629 ± 0.004 | 0.1308 ± 0.015 | 0.3826 ± 0.025 | 0.6615 ± 0.016 | 0.3321 ± 0.009 | 0.3495 ± 0.019 | 0.4907 ± 0.012 | 0.2025 ± 0.018 | 0.0794 ± 0.018 | 0.2507 ± 0.089 |
| | BS-Grad-U | 0.4414 ± 0.025 | **0.1971** ± 0.008 | **0.0246** ± 0.002 | **0.0226** ± 0.001 | **0.1020** ± 0.008 | **0.2099** ± 0.012 | **0.5910** ± 0.027 | 0.3389 ± 0.007 | 0.3294 ± 0.032 | 0.3772 ± 0.012 | 0.1975 ± 0.023 | **0.0115** ± 0.002 | **0.1757** ± 0.105 |
| | IS-U | 0.4436 ± 0.017 | 0.2446 ± 0.017 | 0.0279 ± 0.005 | 0.0605 ± 0.004 | 0.1435 ± 0.024 | 0.4950 ± 0.011 | 0.6570 ± 0.013 | 0.3441 ± 0.009 | 0.3547 ± 0.028 | 0.5458 ± 0.012 | 0.2063 ± 0.019 | 0.0436 ± 0.004 | 0.2646 ± 0.083 |
| | IS-Q | 0.4440 ± 0.018 | 0.2366 ± 0.017 | 0.0281 ± 0.005 | 0.0626 ± 0.004 | 0.1389 ± 0.018 | 0.4525 ± 0.012 | 0.6553 ± 0.012 | 0.3335 ± 0.009 | 0.3531 ± 0.026 | 0.5211 ± 0.012 | 0.2073 ± 0.015 | 0.0443 ± 0.009 | 0.2676 ± 0.084 |
| | IS-CART | 0.4485 ± 0.016 | 0.2456 ± 0.014 | 0.0297 ± 0.004 | 0.0602 ± 0.004 | 0.1394 ± 0.021 | 0.4657 ± 0.011 | 0.6494 ± 0.021 | 0.3327 ± 0.008 | 0.3510 ± 0.027 | 0.5303 ± 0.012 | 0.2085 ± 0.019 | 0.0444 ± 0.009 | 0.2579 ± 0.086 |
| | IS-LGBM | 0.4461 ± 0.016 | 0.2345 ± 0.016 | 0.0297 ± 0.004 | 0.0638 ± 0.004 | 0.1364 ± 0.016 | 0.4660 ± 0.022 | 0.6558 ± 0.010 | 0.3334 ± 0.010 | 0.3518 ± 0.025 | 0.5293 ± 0.017 | 0.2074 ± 0.016 | 0.0441 ± 0.009 | 0.2676 ± 0.086 |
| | IS-Grad-U | **0.4364** ± 0.013 | 0.2105 ± 0.014 | 0.0256 ± 0.005 | 0.0228 ± 0.001 | 0.1086 ± 0.011 | 0.2836 ± 0.020 | 0.6042 ± 0.013 | 0.3397 ± 0.009 | **0.3136** ± 0.029 | 0.4171 ± 0.013 | **0.1967** ± 0.022 | 0.0107 ± 0.002 | 0.2203 ± 0.078 |
| | MS-Grad-U | 0.4413 ± 0.048 | 0.2065 ± 0.004 | 0.0277 ± 0.003 | 0.0248 ± 0.001 | 0.1174 ± 0.009 | 0.2448 ± 0.007 | 0.5988 ± 0.012 | 0.3356 ± 0.009 | 0.3329 ± 0.032 | **0.3572** ± 0.012 | 0.2036 ± 0.026 | 0.0207 ± 0.004 | 0.1984 ± 0.083 |
| RESNET | STD | **0.4277** ± 0.027 | 0.2545 ± 0.017 | 0.0329 ± 0.007 | 0.0339 ± 0.012 | 0.1336 ± 0.015 | 0.4344 ± 0.023 | 0.6484 ± 0.011 | 0.3605 ± 0.011 | 0.3450 ± 0.029 | 0.5103 ± 0.013 | 0.1983 ± 0.019 | 0.0296 ± 0.004 | 0.2449 ± 0.097 |
| | MinMax | 0.4291 ± 0.025 | 0.2805 ± 0.025 | 0.0326 ± 0.005 | 0.0362 ± 0.007 | 0.1321 ± 0.013 | 0.4345 ± 0.018 | 0.6486 ± 0.013 | 0.3644 ± 0.013 | 0.3464 ± 0.030 | 0.5301 ± 0.022 | 0.1988 ± 0.020 | 0.0383 ± 0.009 | 0.2313 ± 0.077 |
| | PLE | 0.4558 ± 0.018 | 0.2003 ± 0.015 | 0.0260 ± 0.004 | 0.0270 ± 0.003 | 0.1067 ± 0.011 | 0.2079 ± 0.007 | 0.6091 ± 0.013 | 0.3288 ± 0.011 | 0.3168 ± 0.034 | 0.3758 ± 0.020 | 0.2007 ± 0.021 | 0.0213 ± 0.006 | 0.2258 ± 0.083 |
| | BS-U | 0.4460 ± 0.012 | 0.2125 ± 0.013 | 0.0256 ± 0.004 | 0.0264 ± 0.002 | 0.1129 ± 0.013 | 0.1563 ± 0.009 | 0.6080 ± 0.016 | 0.3419 ± 0.010 | 0.3242 ± 0.030 | 0.3800 ± 0.014 | 0.2020 ± 0.021 | 0.0164 ± 0.003 | 0.1810 ± 0.090 |
| | BS-Q | 0.4532 ± 0.011 | 0.2009 ± 0.013 | 0.0252 ± 0.004 | 0.0274 ± 0.003 | 0.1082 ± 0.012 | 0.1411 ± 0.014 | 0.6073 ± 0.017 | 0.3290 ± 0.011 | 0.3174 ± 0.030 | 0.3543 ± 0.010 | 0.2012 ± 0.021 | 0.0179 ± 0.002 | 0.1946 ± 0.107 |
| | BS-CART | 0.4522 ± 0.018 | 0.2075 ± 0.011 | 0.0258 ± 0.003 | 0.0291 ± 0.006 | 0.1138 ± 0.014 | 0.1539 ± 0.030 | 0.6101 ± 0.019 | **0.3259** ± 0.007 | 0.3159 ± 0.025 | 0.3716 ± 0.016 | 0.2003 ± 0.022 | 0.0192 ± 0.003 | 0.1983 ± 0.110 |
| | BS-LGBM | 0.4512 ± 0.018 | **0.1988** ± 0.017 | 0.0261 ± 0.003 | 0.0260 ± 0.003 | 0.1094 ± 0.011 | 0.1691 ± 0.015 | 0.6171 ± 0.023 | 0.3304 ± 0.010 | 0.3243 ± 0.026 | 0.3637 ± 0.009 | 0.2052 ± 0.020 | 0.0174 ± 0.003 | 0.1873 ± 0.083 |
| | BS-Grad-U | 0.4558 ± 0.036 | 0.2124 ± 0.017 | 0.0360 ± 0.007 | **0.0204** ± 0.002 | 0.1269 ± 0.015 | **0.1152** ± 0.012 | **0.5761** ± 0.026 | 0.3471 ± 0.007 | 0.3477 ± 0.032 | 0.3336 ± 0.009 | 0.2144 ± 0.021 | **0.0158** ± 0.004 | 0.2081 ± 0.059 |
| | IS-U | 0.4358 ± 0.018 | 0.2204 ± 0.014 | 0.0257 ± 0.003 | 0.0281 ± 0.002 | 0.1090 ± 0.017 | 0.1923 ± 0.013 | 0.6140 ± 0.015 | 0.3399 ± 0.009 | 0.3153 ± 0.028 | 0.4008 ± 0.014 | 0.1972 ± 0.020 | 0.0235 ± 0.003 | 0.1780 ± 0.090 |
| | IS-Q | 0.4372 ± 0.015 | 0.2102 ± 0.016 | 0.0252 ± 0.003 | 0.0258 ± 0.003 | **0.1064** ± 0.016 | 0.2093 ± 0.025 | 0.6166 ± 0.013 | 0.3300 ± 0.007 | **0.3098** ± 0.029 | 0.3856 ± 0.024 | 0.1987 ± 0.018 | 0.0195 ± 0.003 | **0.1746** ± 0.084 |
| | IS-CART | 0.4414 ± 0.018 | 0.2136 ± 0.014 | 0.0252 ± 0.004 | 0.0261 ± 0.001 | 0.1094 ± 0.014 | 0.2126 ± 0.033 | 0.6131 ± 0.023 | 0.3293 ± 0.009 | 0.3134 ± 0.023 | 0.3844 ± 0.019 | **0.1965** ± 0.022 | 0.0191 ± 0.001 | 0.2000 ± 0.074 |
| | IS-LGBM | 0.4428 ± 0.023 | 0.2121 ± 0.022 | 0.0252 ± 0.004 | 0.0263 ± 0.003 | 0.1067 ± 0.014 | 0.1762 ± 0.010 | 0.6123 ± 0.014 | 0.3299 ± 0.009 | 0.3135 ± 0.032 | 0.3930 ± 0.015 | 0.2001 ± 0.017 | 0.0175 ± 0.002 | 0.1810 ± 0.084 |
| | IS-Grad-U | 0.4411 ± 0.016 | 0.2282 ± 0.018 | 0.0349 ± 0.005 | 0.0220 ± 0.002 | 0.1217 ± 0.008 | 0.1472 ± 0.029 | 0.6031 ± 0.007 | 0.3583 ± 0.007 | 0.3284 ± 0.023 | 0.3330 ± 0.015 | 0.2006 ± 0.020 | 0.0207 ± 0.009 | 0.1863 ± 0.035 |
| | MS-Grad-U | 0.4580 ± 0.048 | 0.2144 ± 0.013 | 0.0379 ± 0.007 | 0.0212 ± 0.001 | 0.1377 ± 0.017 | 0.1846 ± 0.010 | 0.6225 ± 0.032 | 0.3476 ± 0.008 | 0.3574 ± 0.045 | **0.3320** ± 0.015 | 0.2057 ± 0.023 | 0.0298 ± 0.014 | 0.2473 ± 0.095 |
| FTT | STD | 0.4609 ± 0.028 | 0.2290 ± 0.010 | **0.0254** ± 0.003 | **0.0201** ± 0.002 | **0.1219** ± 0.013 | 0.1318 ± 0.024 | 0.6727 ± 0.043 | 0.3396 ± 0.013 | **0.3248** ± 0.030 | **0.3858** ± 0.016 | 0.2017 ± 0.029 | **0.0097** ± 0.002 | 0.2805 ± 0.065 |
| | MinMax | **0.4565** ± 0.040 | 0.2496 ± 0.018 | 0.0310 ± 0.005 | 0.0204 ± 0.001 | 0.1276 ± 0.019 | 0.3347 ± 0.088 | 0.6724 ± 0.038 | 0.3485 ± 0.010 | 0.3531 ± 0.026 | 0.4333 ± 0.031 | 0.2066 ± 0.022 | 0.0101 ± 0.006 | 0.2373 ± 0.078 |
| | PLE | 0.4957 ± 0.039 | **0.2138** ± 0.018 | 0.0311 ± 0.007 | 0.0224 ± 0.001 | 0.1339 ± 0.015 | 0.0890 ± 0.058 | 0.6491 ± 0.023 | 0.3346 ± 0.015 | 0.3512 ± 0.025 | 0.4163 ± 0.035 | 0.2088 ± 0.021 | 0.0163 ± 0.006 | 0.2558 ± 0.085 |
| | BS-U | 0.4940 ± 0.017 | 0.2699 ± 0.020 | 0.0335 ± 0.007 | 0.0228 ± 0.002 | 0.1466 ± 0.018 | 0.2731 ± 0.271 | 0.6782 ± 0.019 | 0.3443 ± 0.011 | 0.3696 ± 0.033 | 0.4095 ± 0.022 | 0.2073 ± 0.025 | 0.0299 ± 0.013 | 0.2395 ± 0.108 |
| | BS-Q | 0.5140 ± 0.017 | 0.2426 ± 0.021 | 0.0356 ± 0.005 | 0.0209 ± 0.002 | 0.1458 ± 0.025 | 0.1949 ± 0.230 | 0.6773 ± 0.011 | 0.3434 ± 0.011 | 0.3671 ± 0.022 | 0.3979 ± 0.011 | 0.2116 ± 0.026 | 0.0685 ± 0.025 | 0.2468 ± 0.073 |
| | BS-CART | 0.4872 ± 0.031 | 0.2530 ± 0.022 | 0.0309 ± 0.005 | 0.0232 ± 0.002 | 0.1360 ± 0.011 | 0.4762 ± 0.096 | 0.6667 ± 0.024 | 0.3392 ± 0.009 | 0.3693 ± 0.031 | 0.3950 ± 0.021 | 0.2109 ± 0.030 | 0.0917 ± 0.077 | 0.2534 ± 0.075 |
| | BS-LGBM | 0.5121 ± 0.043 | 0.2290 ± 0.020 | 0.0371 ± 0.009 | 0.0208 ± 0.002 | 0.1514 ± 0.030 | 0.3369 ± 0.191 | 0.6815 ± 0.028 | 0.3422 ± 0.006 | 0.3669 ± 0.015 | 0.4053 ± 0.009 | 0.2102 ± 0.033 | 0.0642 ± 0.006 | 0.2360 ± 0.097 |
| | BS-Grad-U | 0.5157 ± 0.030 | 0.2291 ± 0.033 | 0.0438 ± 0.006 | 0.0220 ± 0.001 | 0.1472 ± 0.036 | 0.1313 ± 0.061 | 0.6798 ± 0.027 | 0.3384 ± 0.006 | 0.3924 ± 0.029 | 0.4448 ± 0.043 | 0.2039 ± 0.030 | 0.0657 ± 0.028 | 0.2810 ± 0.063 |
| | IS-U | 0.4766 ± 0.033 | 0.2426 ± 0.025 | 0.0321 ± 0.004 | 0.0217 ± 0.001 | 0.1448 ± 0.008 | 0.1122 ± 0.196 | 0.6541 ± 0.017 | 0.3666 ± 0.010 | 0.3669 ± 0.019 | 0.3957 ± 0.011 | 0.2058 ± 0.025 | 0.0175 ± 0.003 | 0.2735 ± 0.071 |
| | IS-Q | 0.4732 ± 0.021 | 0.2287 ± 0.008 | 0.0312 ± 0.008 | 0.0202 ± 0.001 | 0.1292 ± 0.023 | 0.0771 ± 0.120 | 0.6459 ± 0.018 | 0.3393 ± 0.003 | 0.3366 ± 0.030 | 0.3958 ± 0.017 | 0.2044 ± 0.029 | 0.0156 ± 0.005 | 0.2586 ± 0.094 |
| | IS-CART | 0.4607 ± 0.028 | 0.2352 ± 0.016 | 0.0342 ± 0.012 | 0.0214 ± 0.001 | 0.1345 ± 0.016 | 0.1381 ± 0.164 | **0.6452** ± 0.037 | **0.3321** ± 0.009 | 0.3399 ± 0.028 | 0.3959 ± 0.016 | 0.2057 ± 0.026 | 0.0164 ± 0.003 | 0.2463 ± 0.078 |
| | IS-LGBM | 0.4632 ± 0.031 | 0.2145 ± 0.018 | 0.0337 ± 0.012 | 0.0221 ± 0.002 | 0.1334 ± 0.016 | **0.0400** ± 0.029 | 0.6481 ± 0.028 | 0.3362 ± 0.011 | 0.3365 ± 0.025 | 0.4008 ± 0.024 | **0.2005** ± 0.028 | 0.0153 ± 0.007 | 0.2383 ± 0.079 |
| | IS-Grad-U | 0.5169 ± 0.065 | 0.2555 ± 0.016 | 0.0468 ± 0.016 | 0.0244 ± 0.003 | 0.1734 ± 0.036 | 0.0719 ± 0.006 | 0.6826 ± 0.043 | 0.3463 ± 0.014 | 0.3809 ± 0.067 | 0.4100 ± 0.011 | 0.2090 ± 0.025 | 0.0254 ± 0.009 | **0.2270** ± 0.099 |
| | MS-Grad-U | 0.5012 ± 0.069 | 0.2786 ± 0.072 | 0.0347 ± 0.004 | 0.0270 ± 0.004 | 0.1571 ± 0.017 | 0.2290 ± 0.175 | 0.6698 ± 0.030 | 0.3486 ± 0.020 | 0.3892 ± 0.039 | 0.4254 ± 0.018 | 0.2067 ± 0.020 | 0.1353 ± 0.062 | 0.2320 ± 0.082 |

Table 12: Regression results for $m = 15$. Mean NRMSE ($\downarrow$) $\pm$ standard deviation over 5-fold cross-validation. Dataset and preprocessing abbreviations are given in Tables 4 and 3 respectively. Bold indicates the lowest NRMSE for each dataset within each backbone.

| Backbone | Method | AB | CA | CPU | DI | HS | PA | WI | FW | H8 | PR | PU | SG | SU |
|---|---|---|---|---|---|---|---|---|---|---|---|---|---|---|
| MLP | STD | 0.4610 ± 0.020 | 0.2751 ± 0.021 | 0.0456 ± 0.007 | 0.0581 ± 0.006 | 0.1638 ± 0.008 | 0.6641 ± 0.018 | 0.7010 ± 0.015 | 0.3536 ± 0.009 | 0.3754 ± 0.019 | 0.5976 ± 0.016 | 0.2126 ± 0.015 | 0.0583 ± 0.004 | 0.2740 ± 0.082 |
| | MinMax | 0.4837 ± 0.013 | 0.3223 ± 0.013 | 0.0569 ± 0.006 | 0.0644 ± 0.003 | 0.2177 ± 0.012 | 0.7792 ± 0.029 | 0.7105 ± 0.012 | 0.4085 ± 0.023 | 0.4292 ± 0.022 | 0.6411 ± 0.017 | 0.2254 ± 0.015 | 0.0826 ± 0.009 | 0.3525 ± 0.070 |
| | PLE | 0.4627 ± 0.040 | 0.2117 ± 0.009 | 0.0292 ± 0.002 | 0.0600 ± 0.002 | 0.1381 ± 0.016 | 0.3021 ± 0.033 | 0.6372 ± 0.027 | 0.3272 ± 0.007 | 0.3531 ± 0.022 | 0.5113 ± 0.013 | 0.2099 ± 0.018 | 0.0304 ± 0.011 | 0.2663 ± 0.088 |
| | BS-U | 0.4672 ± 0.055 | 0.2086 ± 0.006 | 0.0270 ± 0.003 | 0.0603 ± 0.001 | 0.1348 ± 0.014 | 0.2561 ± 0.024 | 0.6573 ± 0.024 | 0.3437 ± 0.008 | 0.3609 ± 0.022 | 0.4791 ± 0.014 | 0.2080 ± 0.020 | 0.0703 ± 0.011 | 0.2509 ± 0.089 |
| | BS-Q | 0.4737 ± 0.037 | 0.1951 ± 0.006 | 0.0266 ± 0.002 | 0.0601 ± 0.001 | 0.1269 ± 0.016 | 0.2737 ± 0.012 | 0.6572 ± 0.025 | **0.3268** ± 0.008 | 0.3590 ± 0.023 | 0.4597 ± 0.009 | 0.2069 ± 0.019 | 0.0872 ± 0.006 | 0.2475 ± 0.087 |
| | BS-CART | 0.4740 ± 0.039 | 0.2130 ± 0.006 | 0.0292 ± 0.003 | 0.0601 ± 0.002 | 0.1233 ± 0.012 | 0.2694 ± 0.034 | 0.6684 ± 0.025 | 0.3329 ± 0.009 | 0.3590 ± 0.022 | 0.4794 ± 0.009 | 0.2092 ± 0.019 | 0.0885 ± 0.007 | 0.2541 ± 0.093 |
| | BS-LGBM | 0.4725 ± 0.039 | 0.1974 ± 0.006 | 0.0269 ± 0.004 | 0.0606 ± 0.002 | 0.1252 ± 0.017 | 0.2036 ± 0.016 | 0.6557 ± 0.022 | 0.3272 ± 0.008 | 0.3572 ± 0.027 | 0.4642 ± 0.010 | 0.2104 ± 0.017 | 0.0876 ± 0.005 | 0.2486 ± 0.091 |
| | BS-Grad-U | 0.4656 ± 0.036 | **0.1603** ± 0.008 | **0.0246** ± 0.003 | **0.0221** ± 0.002 | 0.1027 ± 0.011 | 0.1307 ± 0.016 | 0.5977 ± 0.021 | 0.3350 ± 0.008 | 0.3337 ± 0.027 | 0.3654 ± 0.009 | 0.2028 ± 0.022 | **0.0101** ± 0.001 | 0.1902 ± 0.076 |
| | IS-U | **0.4475** ± 0.046 | 0.2205 ± 0.007 | 0.0265 ± 0.003 | 0.0593 ± 0.001 | 0.1412 ± 0.022 | 0.3577 ± 0.017 | 0.6417 ± 0.024 | 0.3406 ± 0.008 | 0.3576 ± 0.027 | 0.5196 ± 0.014 | 0.2078 ± 0.016 | 0.0572 ± 0.031 | 0.2558 ± 0.085 |
| | IS-Q | 0.4477 ± 0.037 | 0.2129 ± 0.006 | 0.0276 ± 0.002 | 0.0598 ± 0.002 | 0.1356 ± 0.016 | 0.3703 ± 0.014 | 0.6464 ± 0.023 | 0.3297 ± 0.008 | 0.3533 ± 0.023 | 0.5013 ± 0.011 | 0.2084 ± 0.019 | 0.0563 ± 0.031 | 0.2661 ± 0.080 |
| | IS-CART | 0.4581 ± 0.044 | 0.2245 ± 0.006 | 0.0290 ± 0.003 | 0.0607 ± 0.001 | 0.1401 ± 0.018 | 0.3837 ± 0.015 | 0.6472 ± 0.020 | 0.3353 ± 0.009 | 0.3577 ± 0.015 | 0.5232 ± 0.010 | 0.2077 ± 0.016 | 0.0564 ± 0.029 | 0.2634 ± 0.080 |
| | IS-LGBM | 0.4521 ± 0.037 | 0.2123 ± 0.004 | 0.0274 ± 0.002 | 0.0628 ± 0.002 | 0.1339 ± 0.016 | 0.3367 ± 0.013 | 0.6393 ± 0.019 | 0.3287 ± 0.007 | 0.3579 ± 0.021 | 0.5096 ± 0.010 | 0.2113 ± 0.016 | 0.0605 ± 0.034 | 0.2644 ± 0.085 |
| | IS-Grad-U | 0.4488 ± 0.043 | 0.1952 ± 0.007 | 0.0247 ± 0.002 | 0.0231 ± 0.002 | **0.1001** ± 0.016 | **0.1281** ± 0.015 | 0.5968 ± 0.026 | 0.3409 ± 0.008 | **0.3123** ± 0.031 | 0.3858 ± 0.011 | **0.1983** ± 0.023 | 0.0103 ± 0.004 | 0.1907 ± 0.079 |
| | MS-Grad-U | 0.4700 ± 0.042 | 0.1841 ± 0.005 | 0.0264 ± 0.002 | 0.0276 ± 0.002 | 0.1045 ± 0.011 | 0.1324 ± 0.015 | **0.5930** ± 0.019 | 0.3412 ± 0.007 | 0.3462 ± 0.039 | **0.3497** ± | 0.2054 ± 0.018 | 0.0185 ± 0.001 | **0.1906** ± 0.087 |
| RESNET | STD | **0.4277** ± 0.027 | 0.2545 ± 0.017 | 0.0329 ± 0.007 | 0.0339 ± 0.012 | 0.1336 ± 0.015 | 0.4344 ± 0.023 | 0.6484 ± 0.011 | 0.3605 ± 0.011 | 0.3450 ± 0.029 | 0.5103 ± 0.013 | **0.1983** ± 0.019 | 0.0296 ± 0.004 | 0.2449 ± 0.097 |
| | MinMax | 0.4291 ± 0.025 | 0.2805 ± 0.025 | 0.0326 ± 0.005 | 0.0362 ± 0.007 | 0.1321 ± 0.014 | 0.4345 ± 0.018 | 0.6486 ± 0.013 | 0.3644 ± 0.013 | 0.3464 ± 0.030 | 0.5301 ± 0.022 | 0.1988 ± 0.020 | 0.0383 ± 0.009 | 0.2313 ± 0.077 |
| | PLE | 0.4588 ± 0.033 | 0.1847 ± 0.007 | 0.0249 ± 0.002 | 0.0257 ± 0.002 | 0.1077 ± 0.006 | 0.1353 ± 0.035 | 0.5942 ± 0.019 | **0.3229** ± 0.009 | 0.3180 ± 0.029 | 0.3760 ± 0.012 | 0.2008 ± 0.021 | 0.0200 ± 0.004 | 0.2149 ± 0.076 |
| | BS-U | 0.4707 ± 0.046 | 0.1771 ± 0.007 | 0.0251 ± 0.002 | 0.0243 ± 0.001 | 0.1124 ± 0.007 | **0.0782** ± 0.006 | 0.5953 ± 0.021 | 0.3343 ± 0.006 | 0.3444 ± 0.030 | 0.3636 ± 0.016 | 0.2055 ± 0.024 | 0.0185 ± 0.003 | 0.1976 ± 0.102 |
| | BS-Q | 0.4792 ± 0.036 | 0.1776 ± 0.003 | 0.0255 ± 0.003 | 0.0255 ± 0.002 | 0.1141 ± 0.016 | 0.1293 ± 0.007 | 0.6264 ± 0.043 | 0.3273 ± 0.008 | 0.3529 ± 0.032 | 0.3588 ± 0.006 | 0.2109 ± 0.027 | 0.0164 ± 0.002 | 0.2038 ± 0.119 |
| | BS-CART | 0.4680 ± 0.047 | 0.1896 ± 0.006 | 0.0270 ± 0.004 | 0.0261 ± 0.002 | 0.1080 ± 0.013 | 0.1437 ± 0.011 | 0.6219 ± 0.021 | 0.3321 ± 0.012 | 0.3481 ± 0.030 | 0.3795 ± 0.011 | 0.2082 ± 0.021 | 0.0175 ± 0.003 | 0.2215 ± 0.085 |
| | BS-LGBM | 0.4838 ± 0.037 | **0.1763** ± 0.006 | 0.0254 ± 0.003 | 0.0270 ± 0.002 | 0.1100 ± 0.013 | 0.1176 ± 0.015 | 0.6105 ± 0.029 | 0.3249 ± 0.009 | 0.3470 ± 0.026 | 0.3637 ± 0.008 | 0.2091 ± 0.023 | **0.0157** ± 0.002 | 0.2197 ± 0.112 |
| | BS-Grad-U | 0.4965 ± 0.046 | 0.2015 ± 0.011 | 0.0348 ± 0.004 | 0.0222 ± 0.002 | 0.1226 ± 0.008 | 0.0913 ± 0.017 | **0.5823** ± 0.030 | 0.3470 ± 0.009 | 0.3636 ± 0.022 | 0.3652 ± 0.022 | 0.2214 ± 0.021 | 0.0176 ± 0.007 | 0.2258 ± 0.095 |
| | IS-U | 0.4395 ± 0.039 | 0.1939 ± 0.012 | 0.0238 ± 0.003 | 0.0254 ± 0.002 | 0.1096 ± 0.019 | 0.1603 ± 0.046 | 0.6020 ± 0.026 | 0.3402 ± 0.008 | 0.3198 ± 0.026 | 0.3756 ± 0.013 | 0.1998 ± 0.021 | 0.0221 ± 0.004 | 0.1843 ± 0.103 |
| | IS-Q | 0.4431 ± 0.039 | 0.1786 ± 0.007 | **0.0237** ± 0.002 | 0.0262 ± 0.002 | **0.1035** ± 0.011 | 0.1585 ± 0.019 | 0.6015 ± 0.030 | 0.3285 ± 0.009 | **0.3126** ± 0.031 | 0.3646 ± 0.009 | 0.1993 ± 0.020 | 0.0215 ± 0.004 | **0.1759** ± 0.059 |
| | IS-CART | 0.4483 ± 0.042 | 0.2012 ± 0.005 | 0.0258 ± 0.003 | 0.0258 ± 0.001 | 0.1078 ± 0.013 | 0.1463 ± 0.018 | 0.6043 ± 0.025 | 0.3332 ± 0.009 | 0.3215 ± 0.032 | 0.3920 ± 0.009 | 0.2040 ± 0.013 | 0.0182 ± 0.004 | 0.1908 ± 0.090 |
| | IS-LGBM | 0.4463 ± 0.037 | 0.1774 ± 0.008 | 0.0242 ± 0.002 | 0.0257 ± 0.003 | 0.1040 ± 0.012 | 0.1296 ± 0.023 | 0.6037 ± 0.028 | 0.3253 ± 0.008 | 0.3170 ± 0.032 | 0.3669 ± 0.012 | 0.2010 ± 0.020 | 0.0225 ± 0.007 | 0.1767 ± 0.070 |
| | IS-Grad-U | 0.4405 ± 0.045 | 0.2081 ± 0.016 | 0.0289 ± 0.005 | 0.0226 ± 0.003 | 0.1173 ± 0.011 | 0.0859 ± 0.025 | 0.5931 ± 0.020 | 0.3485 ± 0.008 | 0.3368 ± 0.027 | **0.3440** ± 0.015 | 0.2094 ± 0.025 | 0.0189 ± 0.002 | 0.2139 ± 0.013 |
| | MS-Grad-U | 0.4877 ± 0.033 | 0.2030 ± 0.010 | 0.0320 ± 0.005 | **0.0216** ± 0.001 | 0.1261 ± 0.011 | 0.0949 ± 0.025 | 0.6085 ± 0.015 | 0.3479 ± 0.008 | 0.3666 ± 0.026 | 0.3555 ± 0.014 | 0.2161 ± 0.019 | 0.0881 ± 0.106 | 0.2382 ± 0.068 |
| FTT | STD | 0.4609 ± 0.028 | 0.2290 ± 0.010 | **0.0254** ± 0.003 | **0.0201** ± 0.002 | **0.1219** ± 0.013 | **0.1318** ± 0.024 | 0.6727 ± 0.043 | 0.3396 ± 0.013 | **0.3248** ± 0.030 | **0.3858** ± 0.016 | **0.2017** ± 0.029 | **0.0097** ± 0.002 | 0.2805 ± 0.065 |
| | MinMax | **0.4565** ± 0.040 | 0.2496 ± 0.018 | 0.0310 ± 0.005 | 0.0204 ± 0.001 | 0.1276 ± 0.019 | 0.3347 ± 0.088 | 0.6724 ± 0.019 | 0.3485 ± 0.010 | 0.3531 ± 0.026 | 0.4333 ± 0.031 | 0.2066 ± 0.022 | 0.0101 ± 0.009 | **0.2373** ± 0.078 |
| | PLE | 0.4783 ± 0.051 | **0.2090** ± 0.011 | 0.0321 ± 0.007 | 0.0238 ± 0.003 | 0.1285 ± 0.018 | 0.2594 ± 0.084 | 0.6521 ± 0.024 | 0.3825 ± 0.012 | 0.3460 ± 0.043 | 0.4033 ± 0.016 | 0.2091 ± 0.018 | 0.0144 ± 0.009 | 0.2555 ± 0.073 |
| | BS-U | 0.5334 ± 0.055 | 0.2960 ± 0.021 | 0.0343 ± 0.007 | 0.0284 ± 0.006 | 0.1538 ± 0.032 | 0.4548 ± 0.140 | 0.7056 ± 0.042 | 0.3600 ± 0.015 | 0.3865 ± 0.031 | 0.4265 ± 0.010 | 0.2071 ± 0.023 | 0.0399 ± 0.011 | 0.2670 ± 0.079 |
| | BS-Q | 0.5773 ± 0.073 | 0.2558 ± 0.020 | 0.0373 ± 0.006 | 0.0260 ± 0.003 | 0.1425 ± 0.018 | 0.4200 ± 0.168 | 0.7167 ± 0.032 | 0.3536 ± 0.018 | 0.3823 ± 0.025 | 0.4339 ± 0.010 | 0.2184 ± 0.021 | 0.0887 ± 0.040 | 0.2912 ± 0.067 |
| | BS-CART | 0.5222 ± 0.043 | 0.2757 ± 0.037 | 0.0362 ± 0.005 | 0.0271 ± 0.005 | 0.1526 ± 0.026 | 0.4054 ± 0.094 | 0.7287 ± 0.046 | 0.3590 ± 0.028 | 0.3752 ± 0.025 | 0.4522 ± 0.012 | 0.2173 ± 0.034 | 0.0740 ± 0.015 | 0.2987 ± 0.056 |
| | BS-LGBM | 0.5655 ± 0.048 | 0.2533 ± 0.017 | 0.0345 ± 0.007 | 0.0260 ± 0.005 | 0.1430 ± 0.029 | 0.4390 ± 0.109 | 0.6885 ± 0.021 | 0.3598 ± 0.021 | 0.3817 ± 0.017 | 0.4837 ± 0.053 | 0.2191 ± 0.024 | 0.0889 ± 0.023 | 0.2891 ± 0.086 |
| | BS-Grad-U | 0.5173 ± 0.064 | 0.2166 ± 0.035 | 0.0393 ± 0.010 | 0.0290 ± 0.007 | 0.1393 ± 0.015 | 0.1377 ± 0.022 | **0.6146** ± 0.040 | 0.3495 ± 0.015 | 0.4861 ± 0.164 | 0.4667 ± 0.039 | 0.2049 ± 0.020 | 0.0369 ± 0.013 | 0.2817 ± 0.102 |
| | IS-U | 0.4607 ± 0.050 | 0.2305 ± 0.021 | 0.0341 ± 0.004 | 0.0242 ± 0.002 | 0.1401 ± 0.023 | 0.2945 ± 0.198 | 0.6707 ± 0.027 | 0.3407 ± 0.012 | 0.3501 ± 0.030 | 0.4164 ± 0.012 | 0.2125 ± 0.038 | 0.0148 ± 0.003 | 0.2938 ± 0.071 |
| | IS-Q | 0.4632 ± 0.054 | 0.2105 ± 0.009 | 0.0309 ± 0.007 | 0.0267 ± 0.003 | 0.1298 ± 0.010 | 0.1547 ± 0.102 | 0.6412 ± 0.018 | **0.3335** ± 0.010 | 0.3453 ± 0.020 | 0.3975 ± 0.012 | 0.2126 ± 0.031 | 0.0186 ± 0.006 | 0.2547 ± 0.094 |
| | IS-CART | 0.4769 ± 0.046 | 0.2286 ± 0.004 | 0.0357 ± 0.011 | 0.0221 ± 0.001 | 0.1396 ± 0.026 | 0.2034 ± 0.134 | 0.6485 ± 0.025 | 0.3384 ± 0.008 | 0.3393 ± 0.028 | 0.4050 ± 0.016 | 0.2094 ± 0.023 | 0.0161 ± 0.010 | 0.2417 ± 0.088 |
| | IS-LGBM | 0.4576 ± 0.046 | 0.2164 ± 0.008 | 0.0307 ± 0.007 | 0.0234 ± 0.003 | 0.1341 ± 0.034 | 0.1927 ± 0.067 | 0.6493 ± 0.025 | 0.3349 ± 0.009 | 0.3522 ± 0.025 | 0.4089 ± 0.010 | 0.2049 ± 0.028 | 0.0147 ± 0.010 | 0.2460 ± 0.077 |
| | IS-Grad-U | 0.4866 ± 0.045 | 0.2385 ± 0.031 | 0.0458 ± 0.009 | 0.0264 ± 0.005 | 0.1479 ± 0.023 | 0.1463 ± 0.033 | 0.6680 ± 0.042 | 0.3437 ± 0.009 | 0.3809 ± 0.026 | 0.4407 ± 0.033 | 0.2076 ± 0.029 | 0.0147 ± 0.006 | 0.2664 ± 0.080 |
| | MS-Grad-U | 0.5595 ± 0.051 | 0.3371 ± 0.096 | 0.0345 ± 0.004 | 0.0248 ± 0.001 | 0.1737 ± 0.023 | 0.1722 ± 0.033 | 0.7229 ± 0.024 | 0.3911 ± 0.027 | 0.4389 ± 0.087 | 0.4506 ± 0.025 | 0.2170 ± 0.029 | 0.1900 ± 0.033 | 0.2878 ± 0.082 |

Table 13: Regression results for $m = 30$. Mean NRMSE ($\downarrow$) $\pm$ standard deviation over 5-fold cross-validation. Dataset and preprocessing abbreviations are given in Tables 4 and 3 respectively. Bold indicates the lowest NRMSE for each dataset within each backbone.

| Backbone | Method | AD | BA | CH | FI | MA | AQ | EEG | GT | IP | LS | LT | SH |
|---|---|---|---|---|---|---|---|---|---|---|---|---|---|
| MLP | STD | 0.9017 ± 0.004 | 0.8907 ± 0.007 | 0.8425 ± 0.011 | 0.7871 ± 0.005 | 0.8857 ± 0.006 | 0.9934 ± 0.001 | 0.7758 ± 0.054 | 0.9125 ± 0.006 | 0.8999 ± 0.006 | 0.9548 ± 0.002 | 0.9199 ± 0.004 | **0.9994** ± 0.004 |
| | MinMax | 0.8910 ± 0.004 | 0.8799 ± 0.005 | 0.7566 ± 0.017 | 0.7840 ± 0.004 | 0.8714 ± 0.006 | 0.9931 ± 0.001 | 0.5800 ± 0.013 | 0.8916 ± 0.007 | 0.8993 ± 0.006 | 0.9475 ± 0.002 | 0.8945 ± 0.002 | 0.9945 ± 0.002 |
| | PLE | **0.9089** ± 0.003 | 0.9021 ± 0.004 | 0.8405 ± 0.010 | 0.7975 ± 0.006 | 0.8952 ± 0.002 | **0.9958** ± 0.001 | **0.9311** ± 0.004 | 0.9144 ± 0.004 | **0.9477** ± 0.004 | 0.9601 ± 0.002 | **0.9406** ± 0.003 | 0.9986 ± 0.000 |
| | BS-U | 0.9046 ± 0.004 | 0.8974 ± 0.004 | 0.8441 ± 0.011 | **0.7985** ± 0.007 | 0.8925 ± 0.004 | 0.9954 ± 0.001 | 0.6146 ± 0.024 | 0.9173 ± 0.003 | 0.9363 ± 0.005 | 0.9572 ± 0.001 | 0.9265 ± 0.002 | 0.9985 ± 0.001 |
| | BS-Q | 0.9037 ± 0.004 | 0.8988 ± 0.003 | 0.8451 ± 0.011 | 0.7981 ± 0.007 | 0.8901 ± 0.004 | 0.9957 ± 0.001 | 0.7612 ± 0.029 | 0.9208 ± 0.004 | 0.9432 ± 0.005 | 0.9576 ± 0.001 | 0.9355 ± 0.002 | 0.9987 ± 0.001 |
| | BS-CART | 0.9061 ± 0.004 | 0.8987 ± 0.004 | 0.8447 ± 0.010 | 0.7977 ± 0.006 | 0.8937 ± 0.005 | 0.9954 ± 0.001 | 0.7426 ± 0.031 | 0.9211 ± 0.004 | 0.9435 ± 0.004 | 0.9581 ± 0.001 | 0.9367 ± 0.003 | 0.9986 ± 0.001 |
| | BS-LGBM | 0.9064 ± 0.003 | 0.8986 ± 0.003 | 0.8461 ± 0.011 | 0.7979 ± 0.006 | 0.8937 ± 0.006 | 0.9955 ± 0.000 | 0.7618 ± 0.032 | 0.9176 ± 0.004 | 0.9448 ± 0.005 | 0.9582 ± 0.001 | 0.9332 ± 0.004 | 0.9981 ± 0.001 |
| | BS-Grad-U | 0.9078 ± 0.003 | 0.9148 ± 0.003 | **0.8598** ± 0.011 | 0.7981 ± 0.005 | **0.9089** ± 0.006 | 0.9955 ± 0.001 | 0.7177 ± 0.132 | 0.9300 ± 0.004 | 0.9437 ± 0.004 | **0.9618** ± 0.001 | 0.9193 ± 0.002 | 0.9987 ± 0.000 |
| | IS-U | 0.9040 ± 0.004 | 0.8971 ± 0.003 | 0.8425 ± 0.011 | 0.7972 ± 0.004 | 0.8898 ± 0.004 | 0.9955 ± 0.001 | 0.6029 ± 0.019 | 0.9167 ± 0.004 | 0.9325 ± 0.009 | 0.9572 ± 0.001 | 0.9244 ± 0.002 | 0.9975 ± 0.000 |
| | IS-Q | 0.9034 ± 0.004 | 0.8984 ± 0.004 | 0.8411 ± 0.013 | 0.7973 ± 0.004 | 0.8910 ± 0.003 | **0.9958** ± 0.000 | 0.6327 ± 0.020 | 0.9185 ± 0.003 | 0.9436 ± 0.004 | 0.9574 ± 0.001 | 0.9342 ± 0.002 | 0.9978 ± 0.001 |
| | IS-CART | 0.9054 ± 0.003 | 0.8968 ± 0.003 | 0.8425 ± 0.011 | 0.7954 ± 0.003 | 0.8918 ± 0.005 | 0.9957 ± 0.000 | 0.6342 ± 0.018 | 0.9179 ± 0.003 | 0.9442 ± 0.004 | 0.9578 ± 0.001 | 0.9326 ± 0.004 | 0.9983 ± 0.001 |
| | IS-LGBM | 0.9059 ± 0.003 | 0.8975 ± 0.003 | 0.8436 ± 0.011 | 0.7969 ± 0.004 | 0.8915 ± 0.006 | 0.9955 ± 0.001 | 0.6361 ± 0.016 | 0.9175 ± 0.003 | 0.9438 ± 0.003 | 0.9580 ± 0.001 | 0.9297 ± 0.003 | 0.9982 ± 0.001 |
| | IS-Grad-U | 0.9085 ± 0.004 | 0.9118 ± 0.005 | 0.8536 ± 0.007 | 0.7952 ± 0.005 | 0.9065 ± 0.006 | 0.9944 ± 0.001 | 0.6058 ± 0.074 | 0.9272 ± 0.006 | 0.9349 ± 0.003 | 0.9613 ± 0.000 | 0.9267 ± 0.004 | 0.9974 ± 0.001 |
| | MS-Grad-U | 0.9063 ± 0.004 | **0.9161** ± 0.004 | 0.8453 ± 0.015 | 0.7948 ± 0.007 | 0.9077 ± 0.004 | 0.9953 ± 0.000 | 0.6751 ± 0.134 | **0.9313** ± 0.005 | 0.9444 ± 0.002 | 0.9601 ± 0.000 | 0.9114 ± 0.005 | 0.9991 ± 0.001 |
| RESNET | STD | 0.9054 ± 0.004 | 0.9008 ± 0.004 | 0.8537 ± 0.010 | 0.7904 ± 0.005 | 0.8939 ± 0.004 | 0.9953 ± 0.001 | 0.7914 ± 0.107 | 0.9230 ± 0.005 | 0.9152 ± 0.009 | 0.9617 ± 0.001 | 0.9315 ± 0.004 | 0.9988 ± 0.001 |
| | MinMax | 0.9017 ± 0.004 | 0.8925 ± 0.003 | 0.8522 ± 0.015 | 0.7906 ± 0.004 | 0.8815 ± 0.007 | 0.9951 ± 0.001 | 0.8461 ± 0.057 | 0.9226 ± 0.004 | 0.9206 ± 0.014 | 0.9619 ± 0.001 | 0.9243 ± 0.005 | **0.9997** ± 0.000 |
| | PLE | **0.9125** ± 0.003 | 0.9155 ± 0.004 | 0.8562 ± 0.012 | 0.7939 ± 0.006 | 0.9040 ± 0.003 | 0.9962 ± 0.001 | **0.9903** ± 0.002 | 0.9293 ± 0.002 | **0.9473** ± 0.004 | **0.9634** ± 0.002 | **0.9498** ± 0.002 | 0.9993 ± 0.000 |
| | BS-U | 0.9078 ± 0.004 | 0.9094 ± 0.004 | **0.8621** ± 0.011 | 0.7958 ± 0.004 | 0.9029 ± 0.003 | 0.9961 ± 0.001 | 0.8884 ± 0.039 | 0.9361 ± 0.005 | 0.9412 ± 0.001 | 0.9622 ± 0.002 | 0.9419 ± 0.002 | 0.9991 ± 0.001 |
| | BS-Q | 0.9072 ± 0.004 | 0.9126 ± 0.002 | 0.8594 ± 0.011 | 0.7958 ± 0.008 | 0.9029 ± 0.003 | 0.9962 ± 0.001 | 0.9539 ± 0.018 | 0.9369 ± 0.005 | 0.9451 ± 0.003 | 0.9613 ± 0.001 | 0.9465 ± 0.003 | 0.9995 ± 0.001 |
| | BS-CART | 0.9095 ± 0.004 | 0.9125 ± 0.005 | 0.8591 ± 0.013 | **0.7984** ± 0.007 | 0.9030 ± 0.004 | 0.9962 ± 0.001 | 0.8965 ± 0.065 | 0.9360 ± 0.006 | 0.9470 ± 0.004 | 0.9620 ± 0.002 | 0.9492 ± 0.003 | 0.9995 ± 0.001 |
| | BS-LGBM | 0.9100 ± 0.003 | 0.9115 ± 0.004 | 0.8588 ± 0.010 | 0.7974 ± 0.007 | 0.9014 ± 0.006 | 0.9958 ± 0.001 | 0.9304 ± 0.033 | 0.9363 ± 0.005 | 0.9460 ± 0.003 | 0.9619 ± 0.002 | 0.9474 ± 0.002 | 0.9993 ± 0.001 |
| | BS-Grad-U | 0.9094 ± 0.004 | **0.9263** ± 0.002 | 0.8525 ± 0.011 | 0.7891 ± 0.008 | **0.9163** ± 0.006 | 0.9952 ± 0.001 | 0.8375 ± 0.080 | 0.9356 ± 0.005 | 0.9380 ± 0.004 | 0.9628 ± 0.002 | 0.9375 ± 0.003 | 0.9994 ± 0.001 |
| | IS-U | 0.9075 ± 0.004 | 0.9135 ± 0.004 | 0.8587 ± 0.015 | 0.7980 ± 0.006 | 0.9022 ± 0.004 | **0.9965** ± 0.001 | 0.8705 ± 0.030 | 0.9345 ± 0.004 | 0.9402 ± 0.004 | 0.9620 ± 0.002 | 0.9404 ± 0.002 | 0.9992 ± 0.001 |
| | IS-Q | 0.9069 ± 0.004 | 0.9139 ± 0.002 | 0.8561 ± 0.013 | **0.7984** ± 0.006 | 0.9017 ± 0.004 | 0.9964 ± 0.001 | 0.8970 ± 0.031 | 0.9370 ± 0.005 | 0.9433 ± 0.004 | 0.9627 ± 0.002 | 0.9435 ± 0.003 | 0.9994 ± 0.001 |
| | IS-CART | 0.9097 ± 0.004 | 0.9107 ± 0.002 | 0.8591 ± 0.013 | 0.7958 ± 0.004 | 0.9014 ± 0.004 | 0.9964 ± 0.001 | 0.8886 ± 0.041 | **0.9375** ± 0.004 | 0.9465 ± 0.004 | 0.9612 ± 0.002 | 0.9455 ± 0.001 | 0.9995 ± 0.001 |
| | IS-LGBM | 0.9098 ± 0.003 | 0.9131 ± 0.004 | 0.8585 ± 0.013 | 0.7967 ± 0.006 | 0.9036 ± 0.004 | 0.9963 ± 0.001 | 0.9018 ± 0.041 | 0.9353 ± 0.005 | 0.9458 ± 0.004 | 0.9617 ± 0.002 | 0.9441 ± 0.002 | 0.9994 ± 0.001 |
| | IS-Grad-U | 0.9089 ± 0.003 | 0.9253 ± 0.003 | 0.8528 ± 0.011 | 0.7900 ± 0.007 | 0.9152 ± 0.005 | 0.9948 ± 0.001 | 0.6896 ± 0.067 | 0.9359 ± 0.004 | 0.9314 ± 0.004 | 0.9626 ± 0.002 | 0.9372 ± 0.003 | 0.9995 ± 0.000 |
| | MS-Grad-U | 0.9081 ± 0.004 | 0.9243 ± 0.004 | 0.8476 ± 0.013 | 0.7872 ± 0.009 | 0.9147 ± 0.005 | 0.9951 ± 0.001 | 0.6952 ± 0.134 | 0.9327 ± 0.002 | 0.9433 ± 0.002 | 0.9605 ± 0.001 | 0.9328 ± 0.002 | 0.9992 ± 0.001 |
| FTT | STD | 0.9152 ± 0.003 | 0.9294 ± 0.003 | 0.8540 ± 0.011 | 0.7909 ± 0.006 | 0.9215 ± 0.004 | 0.9952 ± 0.001 | **0.9715** ± 0.011 | 0.9234 ± 0.005 | 0.9073 ± 0.013 | 0.9663 ± 0.001 | 0.9324 ± 0.008 | **1.0000** ± 0.000 |
| | MinMax | 0.9147 ± 0.004 | 0.9293 ± 0.004 | 0.8565 ± 0.011 | 0.7952 ± 0.007 | 0.9229 ± 0.003 | 0.9938 ± 0.002 | 0.5766 ± 0.035 | 0.9257 ± 0.004 | 0.9181 ± 0.020 | 0.9666 ± 0.002 | 0.9250 ± 0.006 | 0.9989 ± 0.001 |
| | PLE | **0.9233** ± 0.002 | 0.9324 ± 0.003 | 0.8561 ± 0.007 | 0.7973 ± 0.007 | 0.9244 ± 0.003 | 0.9956 ± 0.000 | 0.9675 ± 0.018 | 0.9238 ± 0.006 | **0.9474** ± 0.003 | 0.9668 ± 0.002 | **0.9488** ± 0.003 | 0.9994 ± 0.000 |
| | BS-U | 0.9155 ± 0.003 | 0.9320 ± 0.004 | 0.8579 ± 0.008 | 0.7962 ± 0.007 | 0.9255 ± 0.003 | 0.9955 ± 0.000 | 0.9675 ± 0.064 | 0.9299 ± 0.006 | 0.9436 ± 0.004 | 0.9666 ± 0.002 | 0.9412 ± 0.005 | 0.9992 ± 0.001 |
| | BS-Q | 0.9163 ± 0.003 | 0.9323 ± 0.003 | 0.8579 ± 0.004 | 0.7946 ± 0.008 | 0.9258 ± 0.004 | 0.9958 ± 0.000 | 0.6680 ± 0.064 | 0.9243 ± 0.007 | 0.9449 ± 0.004 | 0.9670 ± 0.003 | 0.9428 ± 0.004 | 0.9994 ± 0.001 |
| | BS-CART | 0.9168 ± 0.003 | 0.9306 ± 0.003 | 0.8557 ± 0.008 | 0.7974 ± 0.006 | 0.9252 ± 0.005 | **0.9959** ± 0.001 | 0.8585 ± 0.090 | 0.9263 ± 0.005 | 0.9461 ± 0.004 | 0.9669 ± 0.002 | 0.9457 ± 0.004 | **1.0000** ± 0.000 |
| | BS-LGBM | 0.9179 ± 0.004 | 0.9324 ± 0.004 | 0.8585 ± 0.008 | 0.7956 ± 0.006 | 0.9246 ± 0.004 | 0.9957 ± 0.000 | 0.7700 ± 0.062 | 0.9259 ± 0.004 | 0.9452 ± 0.006 | 0.9669 ± 0.002 | 0.9450 ± 0.003 | 0.9995 ± 0.000 |
| | BS-Grad-U | 0.9153 ± 0.003 | 0.9332 ± 0.003 | **0.8623** ± 0.008 | 0.7983 ± 0.007 | 0.9262 ± 0.004 | 0.9943 ± 0.001 | 0.8027 ± 0.050 | 0.9271 ± 0.008 | 0.9451 ± 0.008 | 0.9675 ± 0.001 | 0.9394 ± 0.004 | 0.9988 ± 0.001 |
| | IS-U | 0.9162 ± 0.003 | **0.9340** ± 0.002 | 0.8587 ± 0.008 | 0.7959 ± 0.007 | **0.9269** ± 0.004 | 0.9958 ± 0.000 | 0.6739 ± 0.080 | 0.9300 ± 0.005 | 0.9402 ± 0.010 | 0.9673 ± 0.002 | 0.9393 ± 0.006 | 0.9989 ± 0.001 |
| | IS-Q | 0.9158 ± 0.003 | 0.9323 ± 0.004 | 0.8561 ± 0.011 | 0.7979 ± 0.006 | 0.9259 ± 0.004 | 0.9955 ± 0.001 | 0.6793 ± 0.069 | 0.9238 ± 0.003 | 0.9413 ± 0.008 | 0.9670 ± 0.002 | 0.9403 ± 0.001 | 0.9990 ± 0.000 |
| | IS-CART | 0.9163 ± 0.004 | 0.9336 ± 0.004 | 0.8593 ± 0.011 | 0.7989 ± 0.008 | 0.9258 ± 0.003 | 0.9956 ± 0.001 | 0.6840 ± 0.077 | 0.9265 ± 0.003 | 0.9433 ± 0.005 | 0.9666 ± 0.002 | 0.9436 ± 0.005 | 0.9988 ± 0.001 |
| | IS-LGBM | 0.9166 ± 0.003 | 0.9332 ± 0.003 | 0.8594 ± 0.010 | 0.7972 ± 0.006 | 0.9267 ± 0.004 | 0.9954 ± 0.000 | 0.6516 ± 0.051 | 0.9232 ± 0.009 | 0.9444 ± 0.005 | 0.9669 ± 0.001 | 0.9421 ± 0.003 | 0.9989 ± 0.001 |
| | IS-Grad-U | 0.9171 ± 0.004 | 0.9333 ± 0.004 | 0.8606 ± 0.006 | **0.8006** ± 0.006 | 0.9261 ± 0.004 | 0.9953 ± 0.000 | 0.6190 ± 0.046 | **0.9306** ± 0.010 | 0.9366 ± 0.009 | **0.9680** ± 0.001 | 0.9427 ± 0.006 | 0.9977 ± 0.002 |
| | MS-Grad-U | 0.9159 ± 0.004 | 0.9300 ± 0.004 | 0.8553 ± 0.009 | 0.7924 ± 0.006 | 0.9231 ± 0.003 | 0.9941 ± 0.001 | 0.7681 ± 0.127 | 0.9246 ± 0.004 | 0.9444 ± 0.004 | 0.9659 ± 0.004 | 0.9291 ± 0.004 | 0.9992 ± 0.000 |

Table 14: Classification results for $m = 7$. Mean AUC ($\uparrow$) ± standard deviation over 5-fold cross-validation. For multiclass datasets, AUC corresponds to weighted one-vs-rest ROC-AUC. Dataset and preprocessing abbreviations are given in Tables 4 and 3 respectively. Bold indicates the highest AUC for each dataset within each backbone.

| Backbone | Method | AD | BA | CH | FI | MA | AQ | EEG | GT | IP | LS | LT | SH |
|---|---|---|---|---|---|---|---|---|---|---|---|---|---|
| MLP | STD | 0.9017 ± 0.004 | 0.8907 ± 0.007 | 0.8425 ± 0.011 | 0.7871 ± 0.005 | 0.8857 ± 0.006 | 0.9934 ± 0.001 | 0.7758 ± 0.054 | 0.9125 ± 0.006 | 0.8999 ± 0.006 | 0.9548 ± 0.002 | 0.9199 ± 0.004 | 0.9994 ± 0.001 |
| | MinMax | 0.8910 ± 0.004 | 0.8799 ± 0.005 | 0.7566 ± 0.017 | 0.7840 ± 0.004 | 0.8714 ± 0.006 | 0.9931 ± 0.001 | 0.5800 ± 0.013 | 0.8916 ± 0.007 | 0.8993 ± 0.007 | 0.9475 ± 0.002 | 0.8945 ± 0.004 | 0.9945 ± 0.002 |
| | PLE | **0.9125** ± 0.003 | 0.9069 ± 0.003 | 0.8438 ± 0.010 | 0.7959 ± 0.004 | 0.9000 ± 0.003 | **0.9961** ± 0.001 | **0.9402** ± 0.003 | 0.9192 ± 0.003 | **0.9474** ± 0.004 | 0.9602 ± 0.001 | **0.9492** ± 0.003 | **1.0000** ± 0.000 |
| | BS-U | 0.9066 ± 0.004 | 0.9042 ± 0.003 | 0.8449 ± 0.010 | 0.7965 ± 0.006 | 0.8986 ± 0.005 | 0.9959 ± 0.005 | 0.6683 ± 0.036 | 0.9164 ± 0.003 | 0.9415 ± 0.003 | 0.9585 ± 0.001 | 0.9362 ± 0.003 | 0.9983 ± 0.000 |
| | BS-Q | 0.9062 ± 0.004 | 0.9047 ± 0.003 | 0.8428 ± 0.011 | 0.7947 ± 0.007 | 0.8971 ± 0.004 | 0.9955 ± 0.001 | 0.9284 ± 0.004 | 0.9192 ± 0.004 | 0.9452 ± 0.004 | 0.9590 ± 0.001 | 0.9423 ± 0.002 | 0.9998 ± 0.000 |
| | BS-CART | 0.9097 ± 0.003 | 0.9054 ± 0.002 | 0.8442 ± 0.010 | 0.7950 ± 0.006 | 0.8999 ± 0.004 | 0.9958 ± 0.001 | 0.9218 ± 0.004 | 0.9179 ± 0.004 | 0.9458 ± 0.003 | 0.9592 ± 0.001 | 0.9437 ± 0.003 | **1.0000** ± 0.000 |
| | BS-LGBM | 0.9076 ± 0.003 | 0.9041 ± 0.003 | 0.8422 ± 0.011 | 0.7935 ± 0.008 | 0.8979 ± 0.004 | 0.9957 ± 0.001 | 0.9233 ± 0.006 | 0.9179 ± 0.004 | 0.9463 ± 0.004 | 0.9589 ± 0.001 | 0.9429 ± 0.002 | 0.9999 ± 0.000 |
| | BS-Grad-U | 0.9111 ± 0.003 | 0.9167 ± 0.003 | **0.8559** ± 0.013 | **0.7980** ± 0.014 | **0.9113** ± 0.003 | 0.9958 ± 0.001 | 0.7568 ± 0.100 | 0.9258 ± 0.005 | 0.9434 ± 0.005 | **0.9607** ± 0.001 | 0.9279 ± 0.003 | 0.9989 ± 0.000 |
| | IS-U | 0.9068 ± 0.003 | 0.9032 ± 0.003 | 0.8446 ± 0.011 | 0.7968 ± 0.004 | 0.8971 ± 0.004 | 0.9959 ± 0.001 | 0.6229 ± 0.030 | 0.9186 ± 0.004 | 0.9397 ± 0.005 | 0.9577 ± 0.001 | 0.9322 ± 0.003 | 0.9981 ± 0.001 |
| | IS-Q | 0.9071 ± 0.004 | 0.9048 ± 0.002 | 0.8424 ± 0.012 | 0.7962 ± 0.005 | 0.8986 ± 0.005 | 0.9959 ± 0.000 | 0.8211 ± 0.044 | 0.9237 ± 0.003 | 0.9433 ± 0.005 | 0.9580 ± 0.001 | 0.9389 ± 0.003 | 0.9993 ± 0.000 |
| | IS-CART | 0.9085 ± 0.003 | 0.9039 ± 0.003 | 0.8445 ± 0.011 | 0.7951 ± 0.005 | 0.8986 ± 0.004 | 0.9960 ± 0.001 | 0.8099 ± 0.045 | 0.9194 ± 0.005 | 0.9430 ± 0.003 | 0.9588 ± 0.001 | 0.9378 ± 0.003 | 0.9998 ± 0.000 |
| | IS-LGBM | 0.9075 ± 0.003 | 0.9043 ± 0.003 | 0.8422 ± 0.013 | 0.7956 ± 0.005 | 0.8993 ± 0.005 | **0.9961** ± 0.000 | 0.8159 ± 0.045 | 0.9226 ± 0.004 | 0.9437 ± 0.004 | 0.9579 ± 0.001 | 0.9391 ± 0.002 | 0.9997 ± 0.000 |
| | IS-Grad-U | 0.9107 ± 0.003 | 0.9163 ± 0.002 | 0.8551 ± 0.010 | 0.7960 ± 0.004 | 0.9100 ± 0.006 | 0.9951 ± 0.001 | 0.6561 ± 0.087 | **0.9285** ± 0.005 | 0.9386 ± 0.001 | 0.9600 ± 0.001 | 0.9329 ± 0.003 | 0.9988 ± 0.000 |
| | MS-Grad-U | 0.9101 ± 0.004 | **0.9173** ± 0.003 | 0.8442 ± 0.012 | 0.7927 ± 0.009 | 0.9099 ± 0.004 | 0.9955 ± 0.000 | 0.7291 ± 0.106 | 0.9284 ± 0.005 | 0.9468 ± 0.003 | 0.9579 ± 0.001 | 0.9261 ± 0.004 | 0.9992 ± 0.001 |
| RESNET | STD | 0.9054 ± 0.004 | 0.9008 ± 0.004 | 0.8537 ± 0.010 | 0.7904 ± 0.005 | 0.8939 ± 0.004 | 0.9953 ± 0.001 | 0.7914 ± 0.107 | 0.9230 ± 0.005 | 0.9152 ± 0.009 | 0.9617 ± 0.001 | 0.9315 ± 0.004 | 0.9988 ± 0.001 |
| | MinMax | 0.9017 ± 0.004 | 0.8925 ± 0.003 | 0.8522 ± 0.015 | 0.7906 ± 0.007 | 0.8815 ± 0.007 | 0.9951 ± 0.001 | 0.8461 ± 0.057 | 0.9226 ± 0.001 | 0.9206 ± 0.014 | 0.9619 ± 0.001 | 0.9243 ± 0.005 | 0.9997 ± 0.001 |
| | PLE | **0.9154** ± 0.003 | 0.9159 ± 0.002 | 0.8485 ± 0.011 | 0.7957 ± 0.007 | 0.9083 ± 0.001 | 0.9961 ± 0.001 | **0.9886** ± 0.001 | 0.9302 ± 0.006 | 0.9462 ± 0.006 | **0.9630** ± 0.002 | **0.9590** ± 0.002 | **1.0000** ± 0.000 |
| | BS-U | 0.9098 ± 0.003 | 0.9158 ± 0.003 | 0.8571 ± 0.012 | 0.7964 ± 0.007 | 0.9088 ± 0.006 | 0.9962 ± 0.001 | 0.9467 ± 0.032 | 0.9293 ± 0.006 | 0.9428 ± 0.003 | 0.9613 ± 0.000 | 0.9499 ± 0.002 | 0.9995 ± 0.001 |
| | BS-Q | 0.9096 ± 0.003 | 0.9143 ± 0.004 | 0.8560 ± 0.012 | 0.7943 ± 0.008 | 0.9067 ± 0.005 | 0.9955 ± 0.001 | 0.9766 ± 0.004 | 0.9275 ± 0.005 | 0.9453 ± 0.004 | 0.9615 ± 0.001 | 0.9503 ± 0.001 | **1.0000** ± 0.000 |
| | BS-CART | 0.9121 ± 0.003 | 0.9160 ± 0.003 | **0.8573** ± 0.011 | 0.7943 ± 0.007 | 0.9072 ± 0.005 | 0.9962 ± 0.000 | 0.9711 ± 0.004 | 0.9279 ± 0.005 | 0.9455 ± 0.004 | 0.9619 ± 0.002 | 0.9527 ± 0.002 | **1.0000** ± 0.000 |
| | BS-LGBM | 0.9101 ± 0.003 | 0.9131 ± 0.003 | 0.8548 ± 0.011 | 0.7919 ± 0.009 | 0.9065 ± 0.004 | 0.9957 ± 0.001 | 0.9728 ± 0.006 | 0.9258 ± 0.004 | **0.9465** ± 0.004 | 0.9611 ± 0.002 | 0.9506 ± 0.003 | 0.9998 ± 0.000 |
| | BS-Grad-U | 0.9112 ± 0.003 | 0.9155 ± 0.003 | 0.8433 ± 0.010 | 0.7834 ± 0.010 | 0.9095 ± 0.005 | 0.9955 ± 0.001 | 0.9692 ± 0.031 | 0.9284 ± 0.005 | 0.9406 ± 0.003 | 0.9606 ± 0.001 | 0.9469 ± 0.002 | 0.9996 ± 0.001 |
| | IS-U | 0.9101 ± 0.003 | 0.9160 ± 0.003 | 0.8566 ± 0.011 | **0.7985** ± 0.005 | 0.9078 ± 0.004 | **0.9966** ± 0.001 | 0.9067 ± 0.034 | 0.9344 ± 0.004 | 0.9408 ± 0.002 | 0.9612 ± 0.001 | 0.9475 ± 0.003 | 0.9993 ± 0.001 |
| | IS-Q | 0.9098 ± 0.004 | 0.9144 ± 0.003 | 0.8526 ± 0.012 | 0.7952 ± 0.006 | 0.9068 ± 0.005 | 0.9963 ± 0.000 | 0.9302 ± 0.030 | 0.9348 ± 0.005 | 0.9444 ± 0.003 | 0.9609 ± 0.001 | 0.9475 ± 0.003 | 0.9998 ± 0.000 |
| | IS-CART | 0.9113 ± 0.003 | 0.9173 ± 0.003 | 0.8528 ± 0.012 | 0.7951 ± 0.008 | 0.9088 ± 0.005 | 0.9965 ± 0.001 | 0.9343 ± 0.028 | 0.9329 ± 0.007 | 0.9453 ± 0.004 | 0.9615 ± 0.002 | 0.9479 ± 0.003 | **1.0000** ± 0.000 |
| | IS-LGBM | 0.9100 ± 0.004 | 0.9159 ± 0.001 | 0.8517 ± 0.011 | 0.7936 ± 0.008 | 0.9063 ± 0.006 | 0.9962 ± 0.000 | 0.9291 ± 0.033 | **0.9358** ± 0.002 | 0.9453 ± 0.004 | 0.9605 ± 0.001 | 0.9478 ± 0.002 | **1.0000** ± 0.000 |
| | IS-Grad-U | 0.9096 ± 0.004 | **0.9242** ± 0.003 | 0.8527 ± 0.011 | 0.7862 ± 0.012 | 0.9111 ± 0.006 | 0.9952 ± 0.000 | 0.7376 ± 0.098 | 0.9346 ± 0.005 | 0.9343 ± 0.005 | 0.9610 ± 0.002 | 0.9425 ± 0.003 | 0.9995 ± 0.001 |
| | MS-Grad-U | 0.9093 ± 0.003 | 0.9235 ± 0.003 | 0.8412 ± 0.014 | 0.7843 ± 0.009 | **0.9130** ± 0.005 | 0.9940 ± 0.001 | 0.6962 ± 0.101 | 0.9264 ± 0.005 | 0.9457 ± 0.002 | 0.9562 ± 0.001 | 0.9381 ± 0.003 | 0.9993 ± 0.001 |
| FTT | STD | 0.9152 ± 0.003 | 0.9294 ± 0.004 | 0.8540 ± 0.011 | 0.7909 ± 0.006 | 0.9215 ± 0.004 | 0.9952 ± 0.001 | **0.9715** ± 0.011 | 0.9234 ± 0.005 | 0.9073 ± 0.013 | 0.9663 ± 0.001 | 0.9324 ± 0.008 | **1.0000** ± 0.000 |
| | MinMax | 0.9147 ± 0.004 | 0.9293 ± 0.003 | 0.8565 ± 0.011 | 0.7952 ± 0.007 | 0.9229 ± 0.003 | 0.9938 ± 0.002 | 0.5766 ± 0.035 | 0.9257 ± 0.004 | 0.9181 ± 0.020 | 0.9666 ± 0.002 | 0.9250 ± 0.006 | 0.9989 ± 0.001 |
| | PLE | **0.9254** ± 0.003 | 0.9319 ± 0.003 | 0.8573 ± 0.006 | 0.7950 ± 0.007 | 0.9269 ± 0.004 | 0.9956 ± 0.001 | 0.9711 ± 0.009 | 0.9172 ± 0.004 | **0.9473** ± 0.003 | 0.9677 ± 0.002 | **0.9598** ± 0.006 | **1.0000** ± 0.000 |
| | BS-U | 0.9192 ± 0.004 | 0.9328 ± 0.005 | 0.8525 ± 0.011 | 0.7955 ± 0.007 | **0.9281** ± 0.004 | 0.9959 ± 0.001 | 0.7556 ± 0.087 | 0.9142 ± 0.002 | 0.9400 ± 0.005 | 0.9664 ± 0.003 | 0.9389 ± 0.002 | 0.9997 ± 0.001 |
| | BS-Q | 0.9160 ± 0.003 | 0.9308 ± 0.006 | 0.8530 ± 0.013 | 0.7947 ± 0.006 | 0.9224 ± 0.005 | 0.9954 ± 0.001 | 0.9436 ± 0.008 | 0.9199 ± 0.006 | 0.9441 ± 0.006 | 0.9660 ± 0.001 | 0.9433 ± 0.004 | **1.0000** ± 0.000 |
| | BS-CART | 0.9189 ± 0.003 | 0.9324 ± 0.002 | 0.8556 ± 0.011 | 0.7931 ± 0.006 | 0.9257 ± 0.005 | 0.9959 ± 0.001 | 0.9429 ± 0.009 | 0.9162 ± 0.005 | 0.9440 ± 0.006 | 0.9662 ± 0.001 | 0.9484 ± 0.004 | **1.0000** ± 0.000 |
| | BS-LGBM | 0.9180 ± 0.003 | 0.9319 ± 0.003 | 0.8556 ± 0.013 | 0.7951 ± 0.013 | 0.9231 ± 0.004 | 0.9958 ± 0.000 | 0.9404 ± 0.004 | 0.9160 ± 0.007 | 0.9428 ± 0.007 | 0.9669 ± 0.001 | 0.9435 ± 0.005 | **1.0000** ± 0.000 |
| | BS-Grad-U | 0.9205 ± 0.003 | 0.9297 ± 0.003 | 0.8572 ± 0.009 | 0.7946 ± 0.006 | 0.9266 ± 0.005 | 0.9954 ± 0.001 | 0.7986 ± 0.151 | 0.9250 ± 0.005 | 0.9445 ± 0.005 | 0.9674 ± 0.001 | 0.9338 ± 0.004 | 0.9986 ± 0.001 |
| | IS-U | 0.9176 ± 0.004 | 0.9334 ± 0.003 | 0.8572 ± 0.007 | 0.7974 ± 0.006 | 0.9279 ± 0.004 | 0.9959 ± 0.000 | 0.7335 ± 0.054 | 0.9213 ± 0.007 | 0.9436 ± 0.004 | 0.9676 ± 0.002 | 0.9453 ± 0.004 | 0.9996 ± 0.001 |
| | IS-Q | 0.9194 ± 0.003 | **0.9339** ± 0.003 | 0.8538 ± 0.005 | 0.7967 ± 0.006 | 0.9253 ± 0.005 | 0.9958 ± 0.000 | 0.8895 ± 0.048 | 0.9226 ± 0.006 | 0.9431 ± 0.005 | 0.9676 ± 0.002 | 0.9441 ± 0.004 | **1.0000** ± 0.000 |
| | IS-CART | 0.9203 ± 0.003 | 0.9323 ± 0.003 | 0.8567 ± 0.009 | 0.7956 ± 0.005 | 0.9264 ± 0.004 | **0.9961** ± 0.001 | 0.8665 ± 0.050 | 0.9245 ± 0.005 | 0.9453 ± 0.005 | 0.9679 ± 0.002 | 0.9455 ± 0.004 | **1.0000** ± 0.000 |
| | IS-LGBM | 0.9197 ± 0.003 | 0.9335 ± 0.002 | 0.8543 ± 0.013 | 0.7941 ± 0.009 | 0.9257 ± 0.004 | 0.9957 ± 0.001 | 0.8599 ± 0.045 | **0.9271** ± 0.004 | 0.9440 ± 0.004 | **0.9684** ± 0.001 | 0.9463 ± 0.001 | **1.0000** ± 0.000 |
| | IS-Grad-U | 0.9194 ± 0.002 | 0.9324 ± 0.004 | **0.8616** ± 0.009 | **0.7997** ± 0.009 | 0.9256 ± 0.004 | 0.9960 ± 0.001 | 0.8414 ± 0.122 | 0.9254 ± 0.006 | 0.9459 ± 0.003 | **0.9684** ± 0.001 | 0.9420 ± 0.003 | 0.9996 ± 0.001 |
| | MS-Grad-U | 0.9181 ± 0.003 | 0.9294 ± 0.006 | 0.8519 ± 0.010 | 0.7943 ± 0.007 | 0.9234 ± 0.004 | 0.9948 ± 0.001 | 0.8366 ± 0.081 | 0.9205 ± 0.004 | 0.9451 ± 0.005 | 0.9639 ± 0.001 | 0.9367 ± 0.004 | **1.0000** ± 0.000 |

Table 15: Classification results for $m = 15$. Mean AUC ($\uparrow$) ± standard deviation over 5-fold cross-validation. For multiclass datasets, AUC corresponds to weighted one-vs-rest ROC-AUC. Dataset and preprocessing abbreviations are given in Tables 4 and 3 respectively. Bold indicates the highest AUC for each dataset within each backbone.

| Backbone | Method | AD | BA | CH | FI | MA | AQ | EEG | GT | IP | LS | LT | SH |
|---|---|---|---|---|---|---|---|---|---|---|---|---|---|
| MLP | STD | 0.9017 ± 0.004 | 0.8907 ± 0.007 | 0.8425 ± 0.011 | 0.7871 ± 0.005 | 0.8857 ± 0.006 | 0.9934 ± 0.001 | 0.7758 ± 0.054 | 0.9125 ± 0.006 | 0.8999 ± 0.006 | 0.9548 ± 0.002 | 0.9199 ± 0.004 | 0.9994 ± 0.001 |
|  | MinMax | 0.8910 ± 0.004 | 0.8799 ± 0.005 | 0.7566 ± 0.017 | 0.7840 ± 0.004 | 0.8714 ± 0.006 | 0.9931 ± 0.001 | 0.5800 ± 0.013 | 0.8916 ± 0.007 | 0.8993 ± 0.007 | 0.9475 ± 0.002 | 0.8945 ± 0.004 | 0.9945 ± 0.002 |
|  | PLE | **0.9135** ± 0.005 | 0.9073 ± 0.005 | 0.8436 ± 0.009 | **0.7971** ± 0.009 | 0.9015 ± 0.006 | 0.9957 ± 0.000 | **0.9401** ± 0.004 | 0.9175 ± 0.003 | 0.9467 ± 0.003 | **0.9615** ± 0.002 | **0.9559** ± 0.003 | **1.0000** ± 0.000 |
|  | BS-U | 0.9077 ± 0.006 | 0.9054 ± 0.004 | 0.8431 ± 0.010 | 0.7961 ± 0.008 | 0.9003 ± 0.007 | 0.9954 ± 0.000 | 0.7457 ± 0.051 | 0.9117 ± 0.003 | 0.9435 ± 0.002 | 0.9584 ± 0.002 | 0.9440 ± 0.003 | 0.9994 ± 0.000 |
|  | BS-Q | 0.9092 ± 0.006 | 0.9043 ± 0.005 | 0.8422 ± 0.009 | 0.7959 ± 0.008 | 0.8983 ± 0.008 | 0.9948 ± 0.001 | 0.9237 ± 0.002 | 0.9091 ± 0.004 | 0.9434 ± 0.003 | 0.9596 ± 0.002 | 0.9469 ± 0.002 | **1.0000** ± 0.000 |
|  | BS-CART | 0.9100 ± 0.006 | 0.9017 ± 0.006 | 0.8416 ± 0.008 | 0.7962 ± 0.007 | 0.8964 ± 0.007 | 0.9948 ± 0.000 | 0.9154 ± 0.003 | 0.9118 ± 0.005 | 0.9428 ± 0.001 | 0.9606 ± 0.001 | 0.9501 ± 0.004 | 0.9994 ± 0.000 |
|  | BS-LGBM | 0.9082 ± 0.005 | 0.9038 ± 0.004 | 0.8424 ± 0.009 | 0.7944 ± 0.008 | 0.8973 ± 0.007 | 0.9945 ± 0.001 | 0.9202 ± 0.002 | 0.9109 ± 0.003 | 0.9440 ± 0.003 | 0.9598 ± 0.001 | 0.9472 ± 0.001 | **1.0000** ± 0.000 |
|  | BS-Grad-U | 0.9119 ± 0.006 | 0.9158 ± 0.004 | 0.8515 ± 0.012 | 0.7946 ± 0.006 | 0.9078 ± 0.006 | 0.9955 ± 0.001 | 0.8495 ± 0.031 | 0.9224 ± 0.004 | 0.9455 ± 0.004 | 0.9598 ± 0.001 | 0.9363 ± 0.004 | 0.9996 ± 0.000 |
|  | IS-U | 0.9059 ± 0.006 | 0.9060 ± 0.004 | 0.8452 ± 0.008 | 0.7966 ± 0.008 | 0.9002 ± 0.007 | 0.9959 ± 0.000 | 0.6470 ± 0.031 | 0.9173 ± 0.001 | 0.9426 ± 0.001 | 0.9583 ± 0.001 | 0.9396 ± 0.003 | 0.9983 ± 0.001 |
|  | IS-Q | 0.9064 ± 0.005 | 0.9060 ± 0.003 | 0.8441 ± 0.008 | 0.7963 ± 0.009 | 0.9019 ± 0.006 | 0.9960 ± 0.000 | 0.8332 ± 0.047 | 0.9233 ± 0.003 | 0.9433 ± 0.003 | 0.9588 ± 0.001 | 0.9443 ± 0.003 | 0.9997 ± 0.001 |
|  | IS-CART | 0.9077 ± 0.006 | 0.9054 ± 0.005 | 0.8438 ± 0.008 | 0.7967 ± 0.008 | 0.9005 ± 0.007 | **0.9961** ± 0.000 | 0.8248 ± 0.047 | 0.9198 ± 0.003 | 0.9432 ± 0.002 | 0.9587 ± 0.001 | 0.9440 ± 0.002 | 0.9999 ± 0.000 |
|  | IS-LGBM | 0.9070 ± 0.006 | 0.9057 ± 0.004 | 0.8434 ± 0.008 | 0.7965 ± 0.008 | 0.9017 ± 0.006 | 0.9960 ± 0.000 | 0.8351 ± 0.044 | 0.9226 ± 0.004 | 0.9434 ± 0.004 | 0.9588 ± 0.001 | 0.9444 ± 0.003 | 0.9997 ± 0.000 |
|  | IS-Grad-U | 0.9110 ± 0.004 | **0.9176** ± 0.004 | **0.8532** ± 0.011 | 0.7962 ± 0.010 | **0.9094** ± 0.005 | 0.9950 ± 0.001 | 0.7049 ± 0.069 | **0.9270** ± 0.005 | 0.9416 ± 0.003 | 0.9581 ± 0.001 | 0.9358 ± 0.003 | 0.9988 ± 0.000 |
|  | MS-Grad-U | 0.9120 ± 0.005 | 0.9102 ± 0.005 | 0.8369 ± 0.013 | 0.7890 ± 0.008 | 0.9006 ± 0.005 | 0.9947 ± 0.000 | 0.7887 ± 0.116 | 0.9227 ± 0.006 | **0.9476** ± 0.003 | 0.9513 ± 0.001 | 0.9368 ± 0.004 | 0.9998 ± 0.000 |
| RESNET | STD | 0.9054 ± 0.004 | 0.9008 ± 0.004 | **0.8537** ± 0.010 | 0.7904 ± 0.005 | 0.8939 ± 0.004 | 0.9953 ± 0.001 | 0.7914 ± 0.107 | 0.9230 ± 0.005 | 0.9152 ± 0.009 | 0.9617 ± 0.001 | 0.9315 ± 0.004 | 0.9988 ± 0.001 |
|  | MinMax | 0.9017 ± 0.004 | 0.8925 ± 0.003 | 0.8522 ± 0.015 | 0.7906 ± 0.004 | 0.8815 ± 0.007 | 0.9951 ± 0.001 | 0.8461 ± 0.057 | 0.9226 ± 0.004 | 0.9206 ± 0.014 | 0.9619 ± 0.001 | 0.9243 ± 0.005 | 0.9997 ± 0.000 |
|  | PLE | **0.9179** ± 0.005 | 0.9161 ± 0.006 | 0.8488 ± 0.012 | 0.7956 ± 0.011 | **0.9104** ± 0.005 | 0.9956 ± 0.001 | **0.9870** ± 0.007 | 0.9299 ± 0.004 | 0.9458 ± 0.004 | **0.9634** ± 0.002 | **0.9638** ± 0.002 | **1.0000** ± 0.000 |
|  | BS-U | 0.9126 ± 0.005 | 0.9136 ± 0.005 | 0.8522 ± 0.012 | 0.7912 ± 0.006 | 0.9076 ± 0.005 | 0.9948 ± 0.001 | 0.9679 ± 0.012 | 0.9222 ± 0.004 | 0.9461 ± 0.004 | 0.9599 ± 0.001 | 0.9545 ± 0.003 | 0.9999 ± 0.000 |
|  | BS-Q | 0.9130 ± 0.005 | 0.9116 ± 0.006 | 0.8508 ± 0.010 | 0.7900 ± 0.007 | 0.9055 ± 0.005 | 0.9942 ± 0.001 | 0.9441 ± 0.008 | 0.9136 ± 0.004 | 0.9439 ± 0.004 | 0.9606 ± 0.001 | 0.9529 ± 0.002 | **1.0000** ± 0.000 |
|  | BS-CART | 0.9140 ± 0.005 | 0.9105 ± 0.005 | 0.8500 ± 0.006 | 0.7895 ± 0.009 | 0.9038 ± 0.006 | 0.9941 ± 0.001 | 0.9408 ± 0.005 | 0.9165 ± 0.004 | 0.9433 ± 0.005 | 0.9619 ± 0.001 | 0.9560 ± 0.001 | **1.0000** ± 0.000 |
|  | BS-LGBM | 0.9134 ± 0.005 | 0.9106 ± 0.006 | 0.8486 ± 0.011 | 0.7876 ± 0.007 | 0.9056 ± 0.005 | 0.9939 ± 0.001 | 0.9430 ± 0.007 | 0.9135 ± 0.004 | 0.9449 ± 0.003 | 0.9610 ± 0.001 | 0.9509 ± 0.001 | **1.0000** ± 0.000 |
|  | BS-Grad-U | 0.9119 ± 0.006 | 0.9180 ± 0.005 | 0.8394 ± 0.016 | 0.7748 ± 0.007 | 0.9080 ± 0.003 | 0.9939 ± 0.001 | 0.9740 ± 0.014 | 0.9224 ± 0.003 | 0.9410 ± 0.003 | 0.9596 ± 0.001 | 0.9526 ± 0.003 | 0.9999 ± 0.000 |
|  | IS-U | 0.9108 ± 0.004 | 0.9167 ± 0.005 | 0.8499 ± 0.013 | 0.7965 ± 0.008 | 0.9101 ± 0.006 | 0.9962 ± 0.001 | 0.9043 ± 0.030 | 0.9291 ± 0.003 | 0.9411 ± 0.001 | 0.9603 ± 0.001 | 0.9499 ± 0.001 | 0.9996 ± 0.001 |
|  | IS-Q | 0.9111 ± 0.004 | 0.9151 ± 0.005 | 0.8481 ± 0.007 | **0.7968** ± 0.009 | 0.9095 ± 0.005 | **0.9964** ± 0.001 | 0.9403 ± 0.034 | 0.9328 ± 0.004 | 0.9431 ± 0.001 | 0.9608 ± 0.001 | 0.9514 ± 0.002 | **1.0000** ± 0.000 |
|  | IS-CART | 0.9124 ± 0.005 | 0.9143 ± 0.006 | 0.8507 ± 0.012 | 0.7961 ± 0.009 | 0.9080 ± 0.006 | 0.9961 ± 0.001 | 0.9326 ± 0.027 | 0.9313 ± 0.005 | 0.9447 ± 0.005 | 0.9608 ± 0.001 | 0.9518 ± 0.002 | **1.0000** ± 0.000 |
|  | IS-LGBM | 0.9116 ± 0.004 | 0.9159 ± 0.005 | 0.8492 ± 0.011 | 0.7968 ± 0.009 | 0.9095 ± 0.005 | 0.9961 ± 0.001 | 0.9378 ± 0.031 | **0.9342** ± 0.004 | 0.9424 ± 0.004 | 0.9610 ± 0.001 | 0.9512 ± 0.003 | 0.9999 ± 0.000 |
|  | IS-Grad-U | 0.9111 ± 0.006 | **0.9224** ± 0.004 | 0.8470 ± 0.013 | 0.7857 ± 0.010 | 0.9097 ± 0.004 | 0.9956 ± 0.001 | 0.7183 ± 0.091 | 0.9318 ± 0.004 | 0.9371 ± 0.003 | 0.9589 ± 0.001 | 0.9459 ± 0.003 | 0.9998 ± 0.000 |
|  | MS-Grad-U | 0.9110 ± 0.005 | 0.9129 ± 0.005 | 0.8360 ± 0.010 | 0.7724 ± 0.008 | 0.9046 ± 0.005 | 0.9930 ± 0.001 | 0.8299 ± 0.059 | 0.9215 ± 0.006 | **0.9470** ± 0.004 | 0.9495 ± 0.002 | 0.9453 ± 0.004 | 0.9993 ± 0.001 |
| FTT | STD | 0.9152 ± 0.003 | 0.9294 ± 0.004 | 0.8540 ± 0.011 | 0.7909 ± 0.006 | 0.9215 ± 0.004 | 0.9952 ± 0.001 | **0.9715** ± 0.011 | 0.9234 ± 0.005 | 0.9073 ± 0.013 | 0.9663 ± 0.001 | 0.9324 ± 0.008 | **1.0000** ± 0.000 |
|  | MinMax | 0.9147 ± 0.004 | 0.9293 ± 0.004 | 0.8565 ± 0.011 | 0.7952 ± 0.007 | 0.9229 ± 0.003 | 0.9938 ± 0.002 | 0.5766 ± 0.035 | **0.9257** ± 0.004 | 0.9181 ± 0.020 | 0.9666 ± 0.002 | 0.9250 ± 0.006 | 0.9989 ± 0.001 |
|  | PLE | **0.9243** ± 0.004 | 0.9314 ± 0.005 | 0.8539 ± 0.009 | 0.7965 ± 0.008 | 0.9246 ± 0.004 | 0.9958 ± 0.001 | **0.9715** ± 0.010 | 0.9198 ± 0.005 | **0.9469** ± 0.004 | **0.9695** ± 0.001 | **0.9625** ± 0.004 | **1.0000** ± 0.000 |
|  | BS-U | 0.9177 ± 0.004 | 0.9265 ± 0.008 | 0.8489 ± 0.008 | 0.7954 ± 0.007 | 0.9181 ± 0.005 | 0.9951 ± 0.001 | 0.8390 ± 0.051 | 0.9076 ± 0.002 | 0.9420 ± 0.005 | 0.9647 ± 0.004 | 0.9375 ± 0.004 | 0.9999 ± 0.000 |
|  | BS-Q | 0.9187 ± 0.003 | 0.9233 ± 0.005 | 0.8480 ± 0.010 | 0.7925 ± 0.009 | 0.9170 ± 0.007 | 0.9942 ± 0.001 | 0.8025 ± 0.020 | 0.9061 ± 0.004 | 0.9420 ± 0.006 | 0.9664 ± 0.006 | 0.9415 ± 0.004 | **1.0000** ± 0.000 |
|  | BS-CART | 0.9200 ± 0.004 | 0.9279 ± 0.007 | 0.8478 ± 0.006 | 0.7945 ± 0.009 | 0.9235 ± 0.003 | 0.9945 ± 0.001 | 0.8033 ± 0.012 | 0.9068 ± 0.003 | 0.9436 ± 0.003 | 0.9661 ± 0.002 | 0.9448 ± 0.007 | **1.0000** ± 0.000 |
|  | BS-LGBM | 0.9177 ± 0.004 | 0.9243 ± 0.005 | 0.8483 ± 0.008 | 0.7937 ± 0.009 | 0.9183 ± 0.007 | 0.9941 ± 0.001 | 0.8067 ± 0.024 | 0.9086 ± 0.005 | 0.9425 ± 0.004 | 0.9673 ± 0.001 | 0.9421 ± 0.002 | **1.0000** ± 0.000 |
|  | BS-Grad-U | 0.9208 ± 0.004 | 0.9292 ± 0.003 | 0.8504 ± 0.010 | 0.7962 ± 0.009 | 0.9244 ± 0.003 | 0.9935 ± 0.002 | 0.8885 ± 0.044 | 0.9096 ± 0.004 | 0.9452 ± 0.005 | 0.9665 ± 0.002 | 0.9528 ± 0.010 | 0.9986 ± 0.002 |
|  | IS-U | 0.9201 ± 0.004 | 0.9311 ± 0.005 | 0.8528 ± 0.012 | 0.7986 ± 0.008 | 0.9266 ± 0.004 | 0.9958 ± 0.001 | 0.8535 ± 0.022 | 0.9237 ± 0.005 | 0.9387 ± 0.007 | 0.9675 ± 0.002 | 0.9411 ± 0.004 | **1.0000** ± 0.000 |
|  | IS-Q | 0.9209 ± 0.004 | 0.9322 ± 0.005 | 0.8539 ± 0.009 | 0.7978 ± 0.008 | 0.9263 ± 0.004 | **0.9959** ± 0.001 | 0.8500 ± 0.062 | 0.9238 ± 0.007 | 0.9445 ± 0.005 | 0.9682 ± 0.001 | 0.9473 ± 0.002 | **1.0000** ± 0.000 |
|  | IS-CART | 0.9197 ± 0.004 | **0.9339** ± 0.006 | 0.8537 ± 0.011 | 0.7986 ± 0.008 | 0.9266 ± 0.004 | 0.9958 ± 0.001 | 0.8663 ± 0.021 | 0.9214 ± 0.009 | 0.9435 ± 0.005 | 0.9683 ± 0.001 | 0.9483 ± 0.005 | **1.0000** ± 0.000 |
|  | IS-LGBM | 0.9207 ± 0.004 | 0.9324 ± 0.005 | 0.8533 ± 0.007 | 0.7972 ± 0.008 | 0.9253 ± 0.004 | 0.9953 ± 0.001 | 0.8810 ± 0.066 | 0.9252 ± 0.007 | 0.9437 ± 0.006 | 0.9680 ± 0.002 | 0.9480 ± 0.002 | **1.0000** ± 0.000 |
|  | IS-Grad-U | 0.9208 ± 0.005 | 0.9326 ± 0.006 | **0.8566** ± 0.014 | **0.8001** ± 0.009 | **0.9275** ± 0.004 | 0.9956 ± 0.001 | 0.8770 ± 0.041 | 0.9237 ± 0.010 | 0.9460 ± 0.004 | 0.9675 ± 0.001 | 0.9533 ± 0.013 | **1.0000** ± 0.000 |
|  | MS-Grad-U | 0.9202 ± 0.005 | 0.9264 ± 0.005 | 0.8470 ± 0.008 | 0.7937 ± 0.006 | 0.9224 ± 0.005 | 0.9941 ± 0.000 | 0.9149 ± 0.034 | 0.9102 ± 0.007 | 0.9455 ± 0.004 | 0.9642 ± 0.001 | 0.9417 ± 0.005 | 0.9999 ± 0.000 |

Table 16: Classification results for $m = 30$. Mean AUC ($\uparrow$) $\pm$ standard deviation over 5-fold cross-validation. For multiclass datasets, AUC corresponds to weighted one-vs-rest ROC-AUC. Dataset and preprocessing abbreviations are given in Tables 4 and 3 respectively. Bold indicates the highest AUC for each dataset within each backbone.

| Method | m=5 | 10 | 15 | 20 | 25 | 30 | 35 | 40 | 45 | 50 |
|---|---|---|---|---|---|---|---|---|---|---|
| Std | | | | | | $0.0663 \pm 0.0075$ | | | | |
| MinMax | | | | | | $0.0725 \pm 0.0100$ | | | | |
| $PLE_{adp}^{50}$ | | | | | | $0.0474 \pm 0.0015$ | | | | |
| PLE | $0.0625 \pm 0.0020$ | $0.0592 \pm 0.0029$ | $0.0505 \pm 0.0024$ | $0.0491 \pm 0.0011$ | $0.0491 \pm 0.0010$ | $0.0485 \pm 0.0012$ | $0.0473 \pm 0.0012$ | $0.0476 \pm 0.0012$ | $0.0473 \pm 0.0010$ | $0.0472 \pm 0.0014$ |
| BS-U | $0.0499 \pm 0.0012$ | $0.0474 \pm 0.0015$ | $0.0467 \pm 0.0019$ | $0.0463 \pm 0.0013$ | $0.0460 \pm 0.0020$ | $0.0465 \pm 0.0018$ | $0.0459 \pm 0.0013$ | $0.0463 \pm 0.0012$ | $0.0460 \pm 0.0013$ | $0.0462 \pm 0.0016$ |
| BS-Q | $0.0510 \pm 0.0012$ | $0.0479 \pm 0.0017$ | $0.0477 \pm 0.0017$ | $0.0474 \pm 0.0013$ | $0.0472 \pm 0.0018$ | $0.0473 \pm 0.0016$ | $0.0472 \pm 0.0014$ | $0.0476 \pm 0.0014$ | $0.0474 \pm 0.0016$ | $0.0478 \pm 0.0016$ |
| BS-CART | $0.0507 \pm 0.0006$ | $0.0473 \pm 0.0010$ | $0.0465 \pm 0.0016$ | $0.0461 \pm 0.0018$ | $0.0461 \pm 0.0019$ | $0.0456 \pm 0.0014$ | $0.0457 \pm 0.0013$ | $0.0457 \pm 0.0010$ | $0.0459 \pm 0.0013$ | $0.0460 \pm 0.0014$ |
| BS-LGBM | $0.0504 \pm 0.0013$ | $0.0484 \pm 0.0017$ | $0.0471 \pm 0.0018$ | $0.0473 \pm 0.0019$ | $0.0474 \pm 0.0019$ | $0.0472 \pm 0.0014$ | $0.0472 \pm 0.0015$ | $0.0473 \pm 0.0013$ | $0.0474 \pm 0.0012$ | $0.0474 \pm 0.0014$ |
| BS-Grad-U | $0.0505 \pm 0.0015$ | $0.0478 \pm 0.0010$ | $0.0470 \pm 0.0018$ | $0.0465 \pm 0.0016$ | $0.0467 \pm 0.0019$ | $0.0462 \pm 0.0017$ | $0.0463 \pm 0.0014$ | $0.0462 \pm 0.0013$ | $0.0463 \pm 0.0014$ | $0.0464 \pm 0.0016$ |
| IS-U | $0.0509 \pm 0.0019$ | $0.0499 \pm 0.0023$ | $0.0483 \pm 0.0013$ | $0.0480 \pm 0.0020$ | $0.0471 \pm 0.0009$ | $0.0478 \pm 0.0021$ | $0.0474 \pm 0.0015$ | $0.0472 \pm 0.0018$ | $0.0471 \pm 0.0018$ | $0.0471 \pm 0.0018$ |
| IS-Q | $0.0515 \pm 0.0017$ | $0.0515 \pm 0.0018$ | $0.0502 \pm 0.0012$ | $0.0494 \pm 0.0017$ | $0.0501 \pm 0.0015$ | $0.0499 \pm 0.0018$ | $0.0498 \pm 0.0014$ | $0.0498 \pm 0.0017$ | $0.0499 \pm 0.0016$ | $0.0500 \pm 0.0016$ |
| IS-CART | $0.0515 \pm 0.0020$ | $0.0500 \pm 0.0015$ | $0.0485 \pm 0.0016$ | $0.0475 \pm 0.0012$ | $0.0471 \pm 0.0010$ | $0.0471 \pm 0.0014$ | $0.0468 \pm 0.0010$ | $0.0466 \pm 0.0014$ | $0.0467 \pm 0.0015$ | $0.0468 \pm 0.0016$ |
| IS-LGBM | $0.0510 \pm 0.0022$ | $0.0510 \pm 0.0021$ | $0.0497 \pm 0.0019$ | $0.0495 \pm 0.0018$ | $0.0491 \pm 0.0020$ | $0.0490 \pm 0.0027$ | $0.0488 \pm 0.0020$ | $0.0489 \pm 0.0021$ | $0.0491 \pm 0.0020$ | $0.0493 \pm 0.0019$ |
| IS-Grad-U | $0.0528 \pm 0.0015$ | $0.0509 \pm 0.0018$ | $0.0494 \pm 0.0013$ | $0.0486 \pm 0.0020$ | $0.0492 \pm 0.0018$ | $0.0481 \pm 0.0018$ | $0.0482 \pm 0.0017$ | $0.0481 \pm 0.0018$ | $0.0483 \pm 0.0017$ | $0.0485 \pm 0.0016$ |
| MS-U | $0.0507 \pm 0.0028$ | $0.0528 \pm 0.0046$ | $0.0563 \pm 0.0052$ | $0.0526 \pm 0.0053$ | $0.0567 \pm 0.0088$ | $0.0614 \pm 0.0022$ | $0.0592 \pm 0.0031$ | $0.0566 \pm 0.0060$ | $0.0559 \pm 0.0067$ | $0.0561 \pm 0.0054$ |
| MS-Q | $0.0511 \pm 0.0020$ | $0.0520 \pm 0.0017$ | $0.0520 \pm 0.0029$ | $0.0513 \pm 0.0048$ | $0.0512 \pm 0.0059$ | $0.0536 \pm 0.0092$ | $0.0530 \pm 0.0033$ | $0.0521 \pm 0.0030$ | $0.0519 \pm 0.0030$ | $0.0519 \pm 0.0028$ |
| MS-CART | $0.0521 \pm 0.0021$ | $0.0525 \pm 0.0005$ | $0.0564 \pm 0.0041$ | $0.0552 \pm 0.0066$ | $0.0611 \pm 0.0025$ | $0.0600 \pm 0.0042$ | $0.0608 \pm 0.0052$ | $0.0593 \pm 0.0048$ | $0.0594 \pm 0.0042$ | $0.0595 \pm 0.0038$ |
| MS-LGBM | $0.0513 \pm 0.0023$ | $0.0554 \pm 0.0057$ | $0.0505 \pm 0.0016$ | $0.0519 \pm 0.0029$ | $0.0483 \pm 0.0023$ | $0.0534 \pm 0.0080$ | $0.0535 \pm 0.0077$ | $0.0514 \pm 0.0022$ | $0.0517 \pm 0.0013$ | $0.0524 \pm 0.0010$ |
| MS-Grad-U | $0.0534 \pm 0.0032$ | $0.0494 \pm 0.0024$ | $0.0487 \pm 0.0020$ | $0.0473 \pm 0.0021$ | $0.0479 \pm 0.0017$ | $0.0469 \pm 0.0016$ | $0.0473 \pm 0.0020$ | $0.0472 \pm 0.0020$ | $0.0474 \pm 0.0027$ | $0.0478 \pm 0.0022$ |

Table 17: Sensitivity to encoding resolution on the synthetic regression task. NRMSE (mean $\pm$ std over 5 seeds; lower is better) for varying encoding resolution $m \in \{5, 10, 15, 20, 25, 30, 35, 40, 45, 50\}$, corresponding to the number of bins or basis functions. Preprocessing abbreviations are given in Table 3.

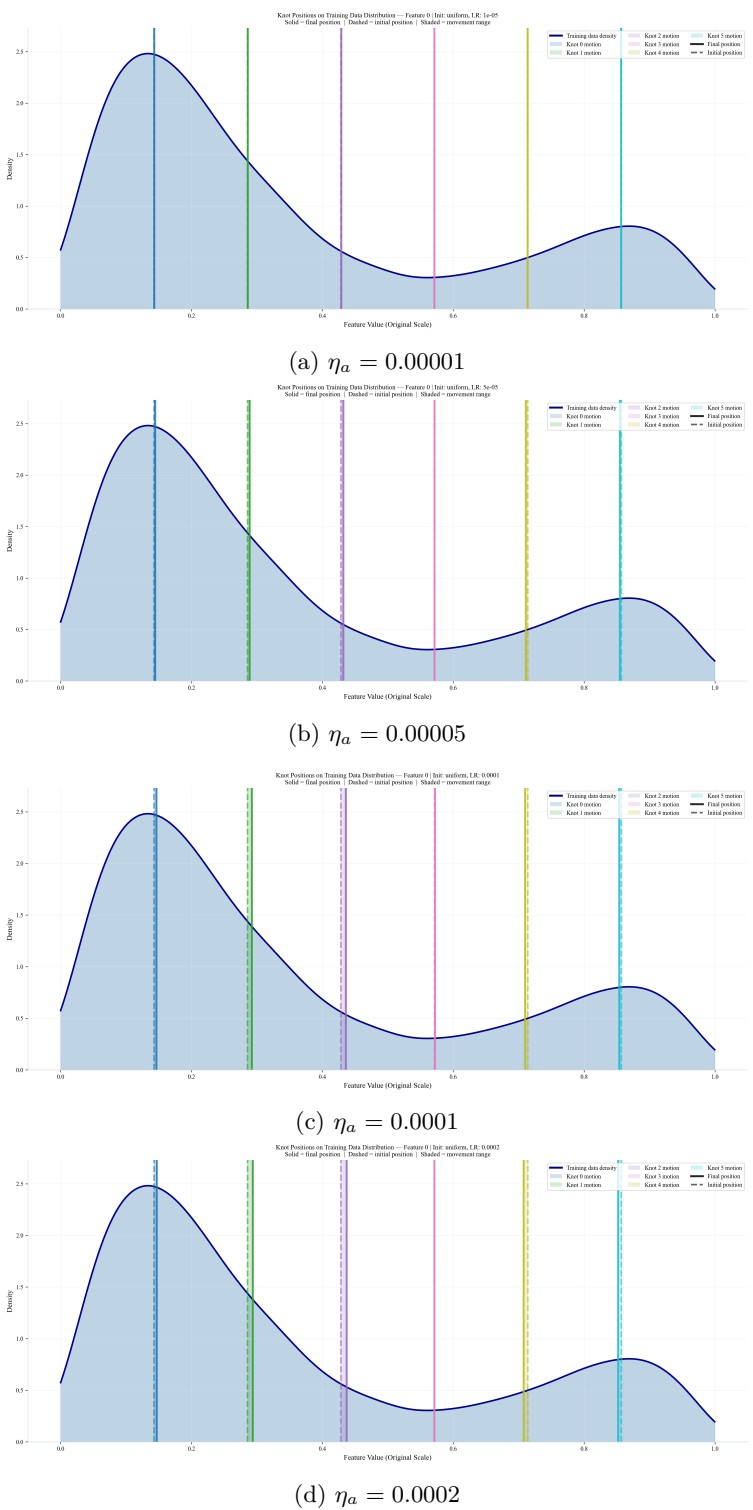

(a) $\eta_a = 0.00001$

(b) $\eta_a = 0.00005$

(c) $\eta_a = 0.0001$

(d) $\eta_a = 0.0002$

Figure 10: Effect of knot learning rate on knot relocation during training. Learned knot trajectories for the informative feature under the learnable-knot B-spline variant BS-Grad-U, shown for increasing knot learning rates $\eta_a \in \{0.00001, 0.00005, 0.0001, 0.0002\}$. All panels use the same synthetic regression setup and training configuration, differing only in the knot learning rate. Smaller values of $\eta_a$ lead to limited knot movement, whereas larger values produce more pronounced relocation during training. Details of BS-Grad-U and the appendix setup are provided in Appendix K.1.

