# OpenReview forum: "From Uniform to Learned Knots: A Study of Spline-Based Numerical Encodings for Tabular Deep Learning"
_TMLR — Under review for TMLR_

### Review · Reviewer_EAjv · 2026-05-06

**Summary Of Contributions:**

Inputting numeric data into a neural network is surprisingly non-trivial. Often, it is helpful to use some sort of basis expansion, where a single float becomes a multi-dimensional vector. This paper provides a very systematic assessment of different encoding schemes, ranging from simple approaches, to approaches based on splines, to spline approaches where the knot locations are learned.

**Audience:**

Yes

**Audience Explanation:**

This paper tackles a practical problem that deserves more interest in the community. It is well executed and could definitely help inform practitioners' choices.

**Claims And Evidence:**

Yes

**Claims Explanation:**

The evaluation is extremely systematic and precise. Care is given so that methods are compared on equal terms in terms of hyper-parameters, feature dimensionality, etc. The analysis is rigorous in terms of statistical methods and spans a large number of datasets. There are also a number of interesting ablations and visualizations. I particularly liked figure 5.

**Requested Changes:**

The paper provides a really systematic study, but I felt that it didn't provide a clear enough set of recommendations for how to actually go about selecting a given approach for a given problem. Of course, one approach is to do a massive sweep over encoding schemes with cross validation, but this is hard to set up. Some sort of recipe for how to choose an approach a-priori would be helpful, even if this recommendation has flaws.


Are there natural ways to extend these findings to multi-dimensional inputs? Instead of transforming each feature independently, you can imagine something that transforms them jointly (e.g. 2 features become a grid of features). What approaches do you think would work best?

There is a lot of overlap between the content in this paper and positional encodings that are used in transformers. Can you discuss some of the similarities/differences?

---

> ### Author Response · Authors · 2026-07-03
> **Answer to Reviewer EAjv**
>
> Dear reviewer,
>
> Thank you for your careful review and constructive suggestions. We have revised the manuscript in response to your comments, and we believe the changes have improved the clarity, completeness, and practical usefulness of the manuscript.
>
> > The paper provides a really systematic study, but I felt that it did not provide a clear enough set of recommendations for how to actually go about selecting a given approach for a given problem. Of course, one approach is to do a massive sweep over encoding schemes with cross validation, but this is hard to set up. Some sort of recipe for how to choose an approach a priori would be helpful, even if this recommendation has flaws.
>
> - To address this, we added a practical recommendation table at the end of Section 4.5. The recommendations are summarized in Table 2 and are based on task, backbone, and computational budget.
> - For classification, we recommend starting with PLE, since it is the strongest default across output sizes and backbones. For regression, we recommend starting with B splines using uniform or quantile knots, and adding target aware knots or I splines when a larger sweep is feasible. For MLP and ResNet style models, we recommend explicit encodings, starting with small output sizes such as $m \approx 7$ to $10$. For FT-Transformer and other high capacity or feature tokenizing models, we recommend starting with Std or MinMax, and adding PLE or fixed knot splines only if needed. For learnable knot variants, we recommend BS-Grad-U as the best efficiency performance tradeoff, while MS-Grad-U and IS-Grad-U are treated as higher cost options.
> - We also note that M-splines are not recommended as the default because of stability and efficiency concerns.
>
> > Are there natural ways to extend these findings to multi dimensional inputs? Instead of transforming each feature independently, you can imagine something that transforms them jointly, for example 2 features become a grid of features. What approaches do you think would work best?
>
> - We agree that this is a natural extension. The current study focuses on feature wise numerical encodings, where each numerical feature is transformed independently before being passed to the backbone. We now state this limitation explicitly in Section 7.
> - A natural extension would be to encode selected low order interactions directly, for example through tensor product or grid based spline encodings. For two features $x_j$ and $x_k$, one possible joint basis is
>
> $$
> b_{j,\ell}^{(p)}(x_j) \, b_{k,r}^{(p)}(x_k)
> $$
>
> where the two basis functions are defined using the feature specific knot sequences $\tau_j$ and $\tau_k$. This would allow interactions between continuous features to be represented at the encoding level rather than only through the downstream backbone.
> - Such constructions can become expensive quickly, since a pairwise tensor product yields $m_j m_k$ basis functions, or $m^2$ when both features use the same output size. Therefore, sparse interaction selection or low rank parameterizations are more practical than applying joint encodings to all feature combinations.
> - We added this discussion to Section 7 as part of the limitations and future work.
>
> > There is a lot of overlap between the content in this paper and positional encodings that are used in transformers. Can you discuss some of the similarities and differences?
>
> - Thank you for this suggestion. We added a new discussion on the relation to positional encodings in Section 2 and clarified the feature wise encoding perspective in Section 3.
> - The similarity is that PLE, spline encodings, sinusoidal positional encodings, and Fourier feature maps all map scalar inputs to vector representations before the main model.
> - The main difference is what the scalar represents. In sequence models, the scalar usually denotes position, so the encoding is designed to expose order, distance, or relative position within a common notion of sequence location. In tabular learning, the scalar is a feature value whose scale, distribution, and meaning can vary across columns.
> - This distinction is central to our setting. Our spline based encodings are feature wise and use feature specific knot sequences, so they represent continuous tabular values through localized basis expansions rather than encoding a universal notion of position.
> - We believe this added discussion clarifies that positional encodings provide a relevant point of comparison, while the motivation and inductive bias of tabular numerical encodings are different.

---

### Review · Reviewer_UAv8 · 2026-06-26

**Summary Of Contributions:**

The paper studies the impact of using different ways of numerical encodings in tabular deep learning. The paper conducts different experiments under different settings, different hyperparameters, and different datasets. The paper also gives some suggestion on how to conduct encoding processing in different tasks and different settings.

**Audience:**

Yes

**Audience Explanation:**

Tabular deep learning is an important field. The preprocessing part has been rarely discussed and studied, but it plays an important role in the training. The paper thoroughly studied these preprocessing methods and gives some helpful suggestions.

**Claims And Evidence:**

Yes

**Claims Explanation:**

The paper conducts a thorough studies on different encoding methods B-splines, M-splines, and integrated splines (I-splines), under
uniform, quantile-based, target-aware, and learnable-knot (gradient-based) placemene on different models,  MLP, ResNet, and FT-Transformer and different tasks (both regression and classification). The paper also gives some good suggestions on different settings applied.

**Requested Changes:**

1. As a non-expert in the tabular deep learning field, I feel Section 3 is difficult to follow and hard to read. I think there are some notation problems where the vector, such as $\pi_j$, needs to be bold to show it is a vector rather than a number. The notations are kind of messy, and there are too many symbols.
2. I have a difficult time interpreting all figures in the papers. I am not sure whether it is better to present as a table for critical difference (CD).
3. I feel there is no enough introduction on the problem of numerial encoding. For example, how is the numerical encoding different from other encoding? why is it more difficult? What's the intuition between every encoding scheme?

---

> ### Author Response · Authors · 2026-07-03
> **Answer to Reviewer UAv8**
>
> Dear Reviewer,
>
> Thank you for your careful review and constructive feedback. We have revised the manuscript in response to your comments, and we believe the changes have improved the clarity, completeness, and practical usefulness of the manuscript.
>
> > As a non-expert in the tabular deep learning field, I feel Section 3 is difficult to follow and hard to read. I think there are some notation problems where the vector needs to be bold to show it is a vector rather than a number. The notations are kind of messy, and there are too many symbols.
>
> We agree that the original methodology section was notation heavy.
> - We revised Section 3 to introduce numerical encoding more gradually as a feature wise scalar to vector mapping. We now distinguish more clearly between scalar feature values, numerical feature vectors, feature wise encoding vectors, concatenated encoded representations, internal knots, and full knot sequences.
> - To make the notation easier to follow, we also added a notation table in Appendix C.1. This table summarizes the main symbols used in the methodology and spline definitions, including $x_{i,j}$, $\phi_j$, $\Phi$, $\kappa_j$, $\tau_j$, and the learnable knot parameters.
> - We also cleaned up the notation in the target aware and learnable knot sections to avoid unnecessary ambiguity.
>
> > I have a difficult time interpreting all figures in the papers. I am not sure whether it is better to present as a table for critical difference (CD).
>
> Thank you for pointing this out. We kept the CD diagrams because they compactly show both the average rank ordering and the statistically indistinguishable groups under the post hoc test. However, we agree that the numerical rank values are easier to inspect in table form.
> - To address this, we added average rank tables for the CD diagrams. Appendix I.1 (Table 9) now reports the average ranks used for the regression CD diagrams, and Appendix I.2 (Table 10) reports the corresponding average ranks for classification. These tables complement Figures 2 and 3 and make the CD results easier to interpret.
>
> > I feel there is no enough introduction on the problem of numerical encoding. For example, how is the numerical encoding different from other encoding? why is it more difficult? What's the intuition between every encoding scheme?
>
> Thank you for pointing this out. We agree that the motivation and intuition behind numerical encoding should be explained more clearly.
> - We revised Section 3 to explain numerical encoding as a feature wise representation problem and to distinguish it from categorical and positional encoding. Categorical encodings represent discrete identifiers, while positional encodings represent positions or coordinates with a common meaning across inputs. In contrast, numerical tabular features are measured quantities whose scale, distribution, and meaning vary across columns. This makes numerical encoding a feature specific modeling choice.
> - We also added Figure 1 to give an intuitive schematic of the main encoding families. The figure shows how one numerical value is represented by Std, MinMax, PLE, and B-, M-, and I-spline encodings. In the revised text, we explain that Std and MinMax keep a single rescaled scalar, PLE represents the value through piecewise linear bin activations, and spline encodings represent the value through evaluations of basis functions over a feature specific knot sequence.